# Quantum Simulation Logic, Oracles, and the Quantum Advantage

**DOI:** 10.3390/e21080800

**Published:** 2019-08-15

**Authors:** Niklas Johansson, Jan-Åke Larsson

**Affiliations:** Institutionen för Systemteknik, Linköpings Universitet, 581 83 Linköping, Sweden

**Keywords:** simulation framework, quantum query complexity, quantum algorithms

## Abstract

Query complexity is a common tool for comparing quantum and classical computation, and it has produced many examples of how quantum algorithms differ from classical ones. Here we investigate in detail the role that oracles play for the advantage of quantum algorithms. We do so by using a simulation framework, Quantum Simulation Logic (QSL), to construct oracles and algorithms that solve some problems with the same success probability and number of queries as the quantum algorithms. The framework can be simulated using only classical resources at a constant overhead as compared to the quantum resources used in quantum computation. Our results clarify the assumptions made and the conditions needed when using quantum oracles. Using the same assumptions on oracles within the simulation framework we show that for some specific algorithms, such as the Deutsch-Jozsa and Simon’s algorithms, there simply is no advantage in terms of query complexity. This does not detract from the fact that quantum query complexity provides examples of how a quantum computer can be expected to behave, which in turn has proved useful for finding new quantum algorithms outside of the oracle paradigm, where the most prominent example is Shor’s algorithm for integer factorization.

## Contents


**1 Introduction**
 **3**   1.1 Quantum Resources 3   1.2 Previous Results Using QSL 4   1.3 Structure of the Present Paper 5
**2 Preliminaries**
 **5**   2.1 Turing Machines 5   2.2 Oracle Turing Machines and Oracle Notions 6   2.3 Quantum Computation 8
**3 Quantum Simulation Logic**
 **11**   3.1  Elementary Systems 12     3.1.1 States 12     3.1.2 Transformations 14     3.1.3 No Universal Spin-1/2 Inverter 17     3.1.4 Measurement 18     3.1.5 Preparation 18     3.1.6 Non-Commutativity of Measurements 18     3.1.7 QKD—BB84 18   3.2  Pairs of Elementary Systems 20     3.2.1 Transformations 20     3.2.2 Entanglement 21     3.2.3 Remote Steering 23     3.2.4 Anticorrelation in Spin-Measurements of the Singlet 24     3.2.5 No-Cloning 24     3.2.6 Interference 25     3.2.7 Measurements 27     3.2.8 Superdense Coding 28   3.3  Higher Number of Elementary Systems 29     3.3.1 Teleportation 30     3.3.2 Transformations 31   3.4  Properties and Relations to Other Theories 32     3.4.1 The Relation to Stabilizer Quantum Mechanics, Locality and Contextuality 32     3.4.2 QSL Extends the State Space of Spekkens’ Model 33     3.4.3 QSL Is an Example of a Generalized Probability Theory 34
**4 The Bernstein-Vazirani Problem**
 **35**   4.1  Problem Formulation 35   4.2  Classical Algorithm 35   4.3  Quantum Algorithm 36   4.4  QSL Simulation 36
**5 The Deutsch-Jozsa Problem**
 **38**   5.1  Problem Formulation 38   5.2  Deterministic and Probabilistic Algorithms 38   5.3  Quantum Algorithm 39   5.4  The Problem for Small Input 40   5.5  QSL Simulation Guaranteed a Constant or Balanced Function 41   5.6  QSL Simulation Accepting Arbitrary Boolean Functions 42   5.7  Query Complexity 45
**6 Oracles as a Comparison**
 **45**   6.1  The Additional Structure and Constraints 45   6.2  Is the Black Box Black? 46   6.3  Assumptions in the Use of Oracles 47   6.4  Systematic Phase Errors 47   6.5  Starting with Something Else Than Access to an Oracle 50
**7 Grover’s Algorithm**
 **50**   7.1  Problem Formulation 51   7.2  One-Shot Grover 51   7.3  The *n*-Toffoli 53   7.4  A Scaling Algorithm 55   7.5  Comparison with a 3-qubit Experiment 56   7.6  Application to Ciphers 56
**8 Simon’s Algorithm**
 **57**   8.1  Problem Formulation 57   8.2  Probabilistic Solution 57   8.3  Quantum Algorithm 58   8.4  QSL Simulation 59   8.5  Adding the Function Output to the Target Modulo 2 61   8.6  A Deterministic Algorithm for SIMON’S Problem 62   8.7  Application to Symmetric Ciphers 64
**9 Shor’s Algorithm Factoring 15**
 **65**
**10  Conclusions**
 **69**
**A   Constant and Balanced Functions for Three Bits of Input**
 **70**
**B   Error Probability for Different Constructions of the Majority Function**
 **72**
**References**



## 1. Introduction

Quantum computers are said to be more powerful than classical computers. This is usually quantified in the following way: a quantum computer needs a smaller number of function calls of some mathematical function under investigation than a classical computer would. Without going into too much detail in this first section, this paper poses the question: what gives the quantum computer the advantage, what is the central property or resource that gives the speed-up?

To answer this question we construct a simulation framework, Quantum Simulation Logic (QSL), that contains one (and only one) property that a quantum computer has, but a classical computer does not: an extra degree of freedom of each bit that the computation is performed on. Each bit then has two degrees of freedom, similar to the two standard conjugate physical degrees of freedom of a mechanical system, position, and momentum. In a quantum computer, the appropriate conjugate degrees of freedom are those of “computation” ($|0\rangle$ and $|1\rangle$), and “phase” ($|0\rangle +|1\rangle$ and $|0\rangle -|1\rangle$), and consequently each QSL-bit is equipped with two degrees of freedom, computation and phase.

We continue to show in that for some problems, a classical system equipped with an extra phase degree of freedom (QSL), needs the same number of function calls of the mathematical function under investigation, as a quantum computer would. For some other problems, QSL does not reach full quantum performance but still has an advantage compared to classical computers without this extra degree of freedom. The full details of the correspondence are contained in the paper.

We do not claim that QSL reproduces the speed-up of generic quantum computation, but it does reproduce the speed-up in certain situations. These results put into question whether the formal tool often used to measure speed-up, the oracle model, really can produce conclusive evidence for a separation between quantum and classical computation. More importantly, the results show that the root cause (resource) for the quantum speed-up is the ability to store, process, and retrieve information in an additional information-carrying degree of freedom in the physical system being used as an information carrier. More on that in what follows.

### 1.1. Quantum Resources

Earlier attempts to understand the resources needed for quantum computation has centered on other properties of quantum mechanics. Some properties that have been proposed as resources for quantum computation are interference [1], entanglement [2], nonlocality [3], contextuality [4,5,6], and even coherence [7,8]. In some of these and subsequent work, a resource is a property used to enable the computational speed-up, usually described as something consumed by the computation, and in others it is thought of as a static resource: a property present in one computational model but absent in another. We should point out that perhaps different algorithms make use of different resources, or different combinations of resources, which further motivates resource studies in the context of specific algorithms.

From our perspective contextuality is clearly a top contender. If contextuality is the intrinsic property that gives quantum computers their advantage one would expect to find evidence of this even in simple algorithms. Our first attempt to study this started with the Deutsch-Jozsa algorithm [9,10] that determines whether a given function is constant or balanced. This was the first proposed quantum algorithm that showed an exponential speed-up compared to the best-known classical algorithm, although this speed-up only holds when requiring the solution to be deterministic.

There are existing results on resources for this algorithm, for example, Collins et al. [11] shows that for one and two qubits there is one implementation where entanglement is not needed. While this does not rule out implementations where entanglement is present, it does show that entanglement is not *strictly* needed for the algorithm to work. The question remains if contextuality is needed. Notable is that for one- and two-qubit input, the oracles can be implemented with a stabilizer circuit, and while stabilizer circuits can be efficiently simulated [12], they can also produce phenomena such as non-local and, in particular, contextual correlations. At this point some tool is needed that enables three-qubit inputs, which also allows comparison between contextual and non-contextual behavior.

### 1.2. Previous Results Using QSL

A framework that is closely related to the stabilizer sub-theory is Spekkens’ model [13] which contains similar transformations, but does not produce non-local and contextual correlations. Interestingly, for the Deutsch-Jozsa algorithm, Spekkens’ model gives the same behavior as the stabilizer sub-theory of quantum mechanics for one- and two-qubit input functions.

Our question at this point was simply: what needs to be added to Spekkens’ model to enable three-qubit inputs (or larger). To our surprise, what was needed was to add a new gate to Spekkens’ model [14], without allowing for non-local correlations and without adding contextual behavior. This new gate enables all balanced functions on an arbitrarily large input, not only a subset thereof. In our paper, we show that for each problem instance there exists a realization in this new extended model, so that all balanced and constant functions at any size of the input can be reproduced. In addition, we show that the standard Deutsch–Jozsa quantum algorithm, using the gates of the model, solves the problem with a single query.

Since non-local and contextual correlations are manifestly absent from the framework, the possibility of stating and solving each problem instance within the model, rules out these properties from being enabling properties. As before, this does not rule out implementations where entanglement or contextuality is present, but shows that they are not *strictly* needed. Furthermore, the model can be efficiently simulated on a classical probabilistic Turing machine, i.e., the algorithm is efficiently simulatable in a classical probabilistic Turing machine—with constant overhead—meaning that there is no speed-up when comparing to the algorithm run on a quantum Turing machine.

Having established this framework, we turned to a second algorithm, Simon’s algorithm [15,16] that finds the generator of a hidden order-two subgroup. This algorithm is usually portrayed as the posterchild for quantum *exponential* speed-up, and is regarded as stronger evidence since the speed-up remains even if the solution is accepted with a bounded error probability. Also, here our framework reproduces the behavior of Simon’s quantum algorithm [14]. Thus, contextuality is not needed, and the algorithm is efficiently simulatable in a classical probabilistic Turing machine. Both algorithms assume access to the function through a black-box oracle, meaning that the solver only has access to the function’s input and output, not the internal structure of the oracle.

The framework used in [14] is limited and puts hard structural constraints on the setup. In this paper, we extend the framework, allowing for a less strict setup resulting in an approximation that gives predictions closer to those of quantum theory. This builds further on the main idea of [14] that the resources needed is the ability *to choose* to store, process, and retrieve information from an additional information-carrying degree of freedom of the physical system.

### 1.3. Structure of the Present Paper

The above observations motivate a study of more algorithms, and the underlying structure of the framework, and this is the focus of the present paper. We begin with some preliminaries (Section 2) and a thorough introduction to our framework (Section 3). We then start with a simple oracle problem called the Bernstein-Vazirani problem (Section 4), for which our model completely reproduces the results of the quantum algorithm. We also show that our approach works even when Deutsch-Jozsa is considered both as a promise problem and as a decision problem (see Section 5). For Grover Search there is a speed-up, but not as much as in the quantum case (Section 7), and also for Simon’s problem our model completely reproduces the results of the quantum algorithm (Section 8).

A more pressing question that these results point to, is whether the oracle model really can produce conclusive evidence for a separation between quantum and classical computation. We tend to believe it cannot, and definitely not for Deutsch-Jozsa and Simon’s problems, see the discussion in Section 6. However, quantum algorithms that produce bounds for quantum query complexity do provide us with examples of how a quantum computer could be expected to behave, and inspire us to invent new approaches to non-relativized computational problems. Shor’s algorithm [17,18] is one such example.

We do point out that there is no principal reason prohibiting our model (or a similar one) to have an effect outside of the oracle paradigm, and that our model is best compared with a construction suffering from systematic errors. In this spirit, we compare our framework with a current state-of-the-art implementation of Shor’s algorithm (Section 9).

## 2. Preliminaries

In this section, we go through some relevant theory helpful to understand the content of this paper, namely the concepts of Turing machines and oracles, and also a brief introduction to quantum computation.

### 2.1. Turing Machines

The concepts of Probabilistic Turing Machines (PTMs) and Quantum Turing Machines (QTMs) are essential here. While it is possible to use the formal notion of a Turing Machine that manipulates classical or quantum symbols on a strip of tape according to a table of rules, the comparison is made simpler by using the presentation of Bernstein and Vazirani [19,20].

In their physics-like view of a PTM, they describe its time evolution as a sequence of probability distributions. At each time-step, the distribution describes the likelihood of the machine’s configuration, and the probability assigned to any specific configuration is a real number in [0,1], to within a precision of $2^{-m}$ for some integer *m*. With a machine able to do coin-flips, these numbers are reachable with a deterministic algorithm in time polynomial in *m*. The probability distribution over all configurations at time *t* is represented by a vector ${\vec{v}}_{t}$. The transition function taking the machine into the next configuration can be thought of as a stochastic matrix *M* whose rows and columns index configurations, entries are probabilities, and the columns sum to one.

Bernstein and Vazirani [20] point out that stochastic matrices obtained from probabilistic TM are finitely specified and map each configuration by making only local changes to it, but even so, the support of the probability distribution can be exponential in the running time of the machine. Observing the PTM (or part of it) at a time-step will yield a configuration (or partial configuration) sampled from the probability distribution, and the distribution is updated conditioned on the value observed, i.e., sampling shrinks the support of the distribution to only cover the observed configuration. This update does not change the future behavior of the machine, because the involved probabilities are additive. It is, therefore, only necessary to keep track of which configuration the machine is in, at each time-step. So, even though the distribution may have a support that grows exponentially with running time, it is only necessary to keep track of a constant amount of information to trace the behavior of the machine as time progresses.

A similar physics-like view of a QTM assigns complex numbers called amplitudes, instead of probabilities, to the possible configurations. The resulting vector of complex amplitudes describes the QTM’s quantum state at each time-step, as a linear combination of configurations known as a superposition, and the probability of observing a specific configuration is given by the absolute square of its amplitude [21]. Furthermore, observation of a measurement outcome enforces the state of the machine to be updated to be consistent with the outcome according to *Lüders rule* [22]. The quantum time evolution needs to keep track of the amplitude and phase of all the configurations because the involved quantities are not only added to each other; the components may cancel each other because of the phase. This phenomenon is known as interference, and is needed to reproduce the quantum behavior. Observation would restrict the configuration so that the interference is prohibited and therefore, in a QTM, observation may disturb its state and its later behavior. The result is that the possible exponential growth of the superposition support needs to be retained to reproduce the complete quantum behavior. In other words, it may be necessary to keep track of an exponentially increasing amount of information to trace the behavior of the machine as time progresses.

More details and the correspondence with the formal Turing Machine model can be found in [19,20]. In what follows the time evolution will be divided into chunks known as gates as we will be using the circuit model of quantum computation. This has been shown to be polynomially equivalent to the QTM model by Yao [23].

### 2.2. Oracle Turing Machines and Oracle Notions

A brief deviation into the standard definition of an oracle TM is needed, before we translate it into the physics-like description. A formal definition can be found in Arora and Barak [24], but here we will adopt the more suitable description by Bennett et al. [25]. An oracle TM has a special *query tape* and two distinguished internal states: a pre-query state and a post-query state. A query is executed whenever the machine enters the pre-query state, and causes the oracle *f* to evaluate the query string present on the query tape (if no query is present the oracle performs a no-op). The query string should be written in the form x||b, i.e., the input to the oracle *x* concatenated with a target bit *b*, commonly initialized to 0, onto which the oracle’s answer is added modulo two. (Except for the query tape and change of internal state from pre- to post-query state, other parts of the oracle TM do not change during the query.) The target bit is strictly speaking not needed for a classical oracle TM, since the answer could be overwritten onto the space used for the query string, but it can be added without loss of generality. More importantly, the target bit enables reversion of the oracle, sometimes known as “uncomputation” [26], since the input equals *x* and the target bit is already equal to f(x), calling the oracle again will reset the target bit to 0. Extending this to several target bits is simple and can be done at a linear cost corresponding to one evaluation per target bit, most presentations condense this to a single evaluation per target bit-string.

Informally [25] an oracle is a device that lets the machine evaluate some function at unit cost. Effectively, using an oracle the question becomes: *if we could compute this function efficiently, what else could we then compute?* With this in mind, oracles can be thought of as a tool used to calculate a lower bound on the resource requirement.

Reformulating this in the already mentioned physics-like description, we describe the evaluation of the oracle as a change in a single time-step, so that the evaluation has unit cost. We will later implement this physics-like description in terms of quantum or classical reversible gates that specify the map in each time-step. In this circuit model, an oracle constitutes a part of the circuit that counts as one single gate when analyzing its resource requirements. Normally, a circuit implementation of a function *f* uses a query register from which the input is read, and an answer register where the result is added, bitwise modulo two (see Figure 1). The query register remains unchanged by the gate; this enables reversibility of the circuit implementation even for non-bijective functions, similar to the oracle TM model above. A simple extension of the argument of Yao [23] can be used to show that this model is polynomially equivalent with the oracle TM model, taking into account that each output bit needs to be handled by a separate oracle in the formal model.

In *oracle query complexity* the relevant quantity is the number of calls an algorithm makes to the oracle. Each call is counted as one operation, or one unit of time, and this is the measure that will be used throughout this paper. As mentioned above, most of the problems that we will consider are *decision* and *promise problems*. The solver of a promise problem is guaranteed that the function is one of a few possible types, and the problem is to determine which it is.

It is conventional in computational complexity to consider the oracle as a black box. Sometimes however, and especially in the related field of cryptography, a wider definition is used to better capture the problem. In the context of analyzing the security of a cryptographic algorithm, an attack model or security model needs to be specified, stating assumptions on the power, knowledge, and access to the system available to an adversary. Chow et al. [27] explains that in the black-box model the adversary is restricted to observe input and output from the cryptographic algorithm, in contrast to the white-box model where he or she is assumed to have complete access to both the specific software implementation, and to the running environment of the algorithm. A gray-box model is a mixture between these two extremes, and applicable whenever the adversary has access to some internal details of the algorithm, but not all.

As an example, consider that we know (are promised) the mapping of a cryptographic function and the problem is to determine an input parameter (a key) based on the output of the function. If access is given to the function in the form of a black box, then the problem is usually intractable. However, if the function runs on some hardware, then a more realistic model is that we can monitor its power consumption. This additional information may open up for a *side-channel* attack known as power analysis; information about the key leaks out through the power consumption, and retrieving the key becomes tractable (see e.g., [28]). Side channels naturally appear when the protocol is implemented in a physical system. This is an example of a gray-box model since it only requires having an incomplete description of the machinery, we only need to have knowledge about the side effect resulting from the physical implementation, and not the full description as in the white-box model.

Another example where these notions fit well is computational modeling and system identification [29], where the goal is instead to model or identify a process. In the white-box setting, we have complete knowledge about the process, and a model can be built from first principles. In the black-box setting we have no prior knowledge about the underlying process, and to our help is only the statistics generated by the input-output process. A gray-box model corresponds to something in between, when we have partial knowledge about the process.

In the circuit model the different black- and non-black-box models will translate into the following:
The white-box model assumes that we know everything that we can know about implementation of the oracle, i.e., we know the specific circuit that performs the function, and even how the computing machinery is built and operates.The black-box model assumes that we know nothing about the implementation of the oracles. We can only access inputs to and outputs from the circuit.A gray-box is some specified mixture of black and white, for instance, it could be that we have more knowledge about how the function is implemented, but not complete knowledge.

### 2.3. Quantum Computation

A QTM computer is a machine that operates on quantum bits, or qubits, rather than bits. Qubits are the elementary information carriers in quantum theory. They are two-level systems that can be parameterized by two complex numbers *a* and *b* through the expression
(1)$$a|0\rangle +b|1\rangle ,$$
where $|0\rangle$ and $|1\rangle$ are the two orthogonal eigenstates to the Pauli-Z operator,
(2)$$Z|0\rangle =|0\rangle ,\;Z|1\rangle =-|1\rangle .$$
These states are usually referred to as the computational basis states.

We will use two kinds of operations on these qubits, *reversible transformations* and *measurements*. *Transformations* relate to the classical bit operations and are described by unitary operators
(3)$$AA^{†}=A^{†}A=I,$$
where (${}^{†}$) is the Hermitian conjugate, and *I* the identity operator. These operators are also referred to as quantum gates. *Measurements* are used to retrieve information from these systems, and of particular use are projective measurements. In such a measurement the state is projected onto an eigenstate of an observable (a Hermitian operator), and the information retrieved by an observer from such a measurement is the eigenvalue corresponding to that eigenstate. The probability of obtaining a specific outcome from a specific observable depends on the state before measurement, and is given by the *Born rule* [21]. The probability is given by the absolute square of the component (amplitude) in the direction of the eigenstate related to the outcome. As an example, the probability of finding the qubit represented by expression (Equation 1) in state $|0\rangle$, from measuring the *Z* observable, is ${\left|a\right|}^{2}$. Furthermore, the state of the physical system is changed in a projective measurement to the eigenstate in question (or a vector in the eigenspace, in the degenerate case). The state is projected onto the eigenspace, hence the term projective measurement. The requirement that the probabilities of the different outcomes sums to one translates into normalization of the state so that ${\left|\psi \right|}^{2}=1$.

There exists a third operation: *preparation*, that can be thought of as measurement on an unknown state followed by a unitary transformation that depends on the measurement outcome, such that the output eigenstate is rotated to the desired state. For a more thorough explanation of the primitives (and their generalizations that we do not treat here), see Nielsen and Chuang [30].

An important concept in quantum information theory is that of *mutually unbiased bases*. Take two sets of states $\left\{|e_{i}\rangle \right\}$ and $\left\{|f_{j}\rangle \right\}$, both being bases for the quantum system. These two bases are said to be mutually unbiased if any pair of two states formed between the sets satisfy the condition
(4)$$|\langle e_{i}|f_{j}\rangle {|}^{2}=\frac{1}{d},$$
where *d* is the dimension of the system. Importantly, the left-hand side is constant, independent of bases and states, so that information retrieved from a projective measurement along one basis is completely unrelated to the information retrieved from a projective measurement in the other basis [31].

As an example, the bases spanned by the eigenstates of the Pauli operators X,Y and *Z* are 2-dimensional and mutually unbiased.

(5)$${\left\{|0\rangle ,|1\rangle \right\}}_{Z},\;{\left\{|+\rangle ,|-\rangle \right\}}_{X},\;{\left\{|+i\rangle ,|-i\rangle \right\}}_{Y}$$

A transition between the computational basis and the *phase basis*, spanned by the eigenstates of *X*, can be performed by the Hadamard gate $H=H^{†}$ by
(6)$$H|0\rangle =|+\rangle ,\;H|1\rangle =|-\rangle .$$
It also transforms the Pauli operators according to
(7)HZH=X,HXH=Z,HYH=-Y.

Composition of systems of several qubits uses the tensor product, for example
(8)$${|\psi \rangle }_{AB}={|\phi \rangle }_{A}⊗{|\varphi \rangle }_{B}={|\phi \rangle }_{A}{|\varphi \rangle }_{B},$$
where the subscripts *A* and *B* indicates that the states describe two subsystems, but these are often dropped from the notation. The dimension of the tensor product vector space is the product of the constituent spaces, and therefore, computations involving *n* qubits will have states described by a $2^{n}$-dimensional Hilbert space. States that can be written on the form (Equation 8) are called product states. The tensor product creates a Hilbert space (here, finite-dimensional complex vector space) of all linear combinations of the possible separable states. Some of these linear combinations cannot be written as a product state as in Equation (Equation 8), and these are called entangled states.

The most well-known two-qubit gate is the quantum *CNOT* gate. A *CNOT* is a controlled-*X* operation; it applies *X* to the target qubit conditioned on the control qubit. The graphical presentation is a dot on the control qubit, connected with a control line to the target gate, here an *X*. For some examples of gate arrangements, see Figure 2. With $|c\rangle$ as control and $|t\rangle$ as target it has the effect
(9)$$CNOT|c\rangle |t\rangle =|c\rangle |t⊕c\rangle ,$$
where “⊕” is the exclusive-OR, or addition modulo 2. The effect of the *CNOT* on the four computational basis states is
(10)$$\begin{array}{c} |00\rangle \to |00\rangle \\ |01\rangle \to |01\rangle \\ |10\rangle \to |11\rangle \\ |11\rangle \to |10\rangle . \end{array}$$
This is the effect as seen from the computational basis. From the point of view of the phase basis, transforming the operator via Hadamard transforms, we obtain a different map (see Figure 2A), and it is easy to verify that for example
(11)$$\begin{array}{cc} |+-\rangle = & \frac{1}{2}|00\rangle -\frac{1}{2}|01\rangle +\frac{1}{2}|10\rangle -\frac{1}{2}|11\rangle \overset{CNOT}{\to }\frac{1}{2}|00\rangle -\frac{1}{2}|01\rangle +\frac{1}{2}|11\rangle -\frac{1}{2}|10\rangle =|--\rangle . \end{array}$$
The effect of the *CNOT* on the four phase-basis states is
(12)$$\begin{array}{c} |++\rangle \to |++\rangle \\ |+-\rangle \to |--\rangle \\ |-+\rangle \to |-+\rangle \\ |--\rangle \to |+-\rangle , \end{array}$$
that is,
(13)$$H^{⊗2}\;CNOT\;H^{⊗2}|c\rangle |t\rangle =|c⊕t\rangle |t\rangle .$$
This effect in the phase basis is sometimes called phase kickback [32].

A controlled phase flip (Controlled-*Z* or CZ) and a qubit swap (*SWAP*) that corresponds to classically swapping the qubits [33], can also be constructed directly from the *CNOT* (see Figure 2A). The CZ and *SWAP* behaviors correspond to small modifications of Equations (Equation 10) and (Equation 12).

Toffoli and Fredkin gates are controlled versions of the CNOT and SWAP, respectively. A Toffoli gives the map
(14)$$\mathrm{Toffoli}|c\rangle |c^{\prime }\rangle |t\rangle =|c\rangle |c^{\prime }\rangle |t⊕cc^{\prime }\rangle ,$$
while a Fredkin gives
(15)$$\mathrm{Fredkin}|c\rangle |t\rangle |t^{\prime }\rangle =|c\rangle |t⊕c(t⊕t^{\prime })\rangle |t^{\prime }⊕c\left(t⊕t^{\prime }\right)\rangle ,$$

A Toffoli gate with *n* controlling qubits is called an *n*-Toffoli. Also, inverted (white) controls enables the gate if the control system is in state $|0\rangle$, and can be implemented by applying an *X* before and after the control.

Since functions in quantum computation are constructed from unitary transformations, any function implemented needs to be reversible. The standard approach is to use the circuit model wherein the computation is described by a reversible circuit, and then replace the classical reversible gates by their quantum equivalents. Then, the circuit constitutes a unitary operation with the same mapping of the computational basis states as the classical analogue of a reversible circuit operating on bits. In contrast to the classical reversible function, there may be additional information to retrieve from the output of the query register. There is at least one additional information-carrying degree of freedom that we can choose to retrieve information from, instead of retrieving the result of the computation. Of course, in classical reversible logic, we cannot expect to retrieve any additional information about the computation from the query register after such a transformation, see Figure 3.

A quantum oracle is an oracle assumed to be a specific unitary transformation acting on qubits (or other quantum systems) rather than on classical bits. (For a formal definition in the QTM model, see Bennett et al. [25].) Such an oracle is a unitary transformation, so it is necessarily a reversible map from qubits to qubits. A quantum oracle is usually described by a classical reversible logical function encoded into the computational basis of a unitary operator. Having access to such an oracle a quantum computer can exploit the ability to sample from some distributions related to the function, rather than just having access to the function itself. This will be further discussed in Section 6.

## 3. Quantum Simulation Logic

In what follows we will use QSL, which extends an earlier model for simulation of quantum mechanical systems known as Spekkens’ toy model [13]. That model views the quantum state as a state of the observer’s knowledge, an *epistemic state*, represented as probability distributions over a set of *ontic states*, and the ontic state is hidden away from the observer. The word hidden is key here, because the model restricts the knowledge an observer can have about the ontic state of the system; *the knowledge balance principle* states that the amount of knowledge an observer has about the system is at most equally large as the amount of knowledge an observer lacks about the system. This implies that there is a restriction to the allowed probability distributions that are used to describe states in the model.

Spekkens’ model associates the ontic states of a qubit system with the four points of the corresponding phase space. QSL indexes these four points using two bits of information, so that the ontic state is represented through two classical bits, details will follow below. Since the main part of the paper is about computation, the words ontic and epistemic will not be used below, but those acquainted with Spekkens’ model may find it useful to remember that the basic bit-values in the QSL model are associated with Spekkens’ ontic states, and a probabilistic combination of different bit-values correspond to Spekkens’ epistemic states, or indeed, correspond directly to quantum states of the simulated quantum system.

In this section, we will see several examples of how QSL is similar to quantum theory. Some, but not all, of these examples can be stated directly in Spekkens’ toy model, and a few are even present in Ref. [13]. We include these to show how easy it is to follow the time evolution of a QSL system, protocol, or algorithm, as compared to the standard representation of Spekkens’ model, and to illustrate and get a feeling of how the QSL framework works. One should be aware that QSL extends beyond Spekkens’ toy model (through the Toffoli as described below), so some properties of Spekkens’ model do not hold in QSL, but QSL can in turn reproduce phenomena not reproducible in Spekkens’ model.

### 3.1. Elementary Systems

Each elementary QSL system is constructed from two classical information carriers that each can carry one bit of information. Yet, preparation and measurement only allow for storage of one bit of information or retrieval of one bit of information, while the other is destroyed through randomization, just as in Spekkens’ model. This gives the following description.

#### 3.1.1. States

The state of an elementary system is represented by a tuple
(16)$$(x_{0},p_{0}),$$
where $x_{0}$ and $p_{0}$ represent the bit-values of the two classical carriers.

Preparing an elementary system to have a definite value in the $x_{0}$ bit corresponds to preparing a quantum system in the computational basis state $|x_{0}\rangle$, an eigenstate of the *Z* observable. For this reason, $x_{0}$ will be referred to as the *computational bit*. Since the elementary system only can carry one bit of information, the second bit $p_{0}$ is in this case necessarily described by an equally weighted random variable R∈{0,1}. In other words, to simulate the eigenstates of the *Z* observable we have
(17)$$\left(0,R\right)∼|0\rangle ,\;\left(1,R\right)∼|1\rangle .$$
Similarly, preparing an elementary system to have a definite value of the $p_{0}$ bit corresponds to preparing a quantum system in a phase-basis state,
(18)$$\left(R,0\right)∼|+\rangle =\frac{|0\rangle +|1\rangle }{\sqrt{2}},\;\left(R,1\right)∼|-\rangle =\frac{|0\rangle -|1\rangle }{\sqrt{2}}.$$
These give us a simulation of eigenstates of the *X* observable. For this reason, $p_{0}$ will be referred to as the *phase bit*. Also here, the other bit, in this case the computational bit, must be random.

Please note that the $|1\rangle$ state is associated with $x_{0}=1$ and the -1 eigenvalue of the *Z* observable, and similarly, the $|-\rangle$ state is associated with $p_{0}=1$ and the -1 eigenvalue of the *X* observable. In what follows, the *X* observable will be used interchangeably with the projection $\left(I-X\right)/2=|-\rangle \langle -|$, differing only in eigenvalues while retaining eigenvectors.

This brings us to the first analogy with quantum theory, namely the *uncertainty principle*. An observer cannot simultaneously make a value assignment to the complementary observables *X* and *Z*, and having perfect knowledge about one implies having none about the other. This is true in both QSL and in quantum theory.

The information can also be stored in the correlation between the computational bit and phase bit. They are then either correlated or anti-correlated, and this simulates the eigenstates of the *Y* observable.
(19)$$\left(R,R\right)∼\frac{|0\rangle +i|1\rangle }{\sqrt{2}},\;\left(R,R⊕1\right)∼\frac{|0\rangle -i|1\rangle }{\sqrt{2}}$$
Also here, each value of the bit XOR is associated with the eigenspaces of the (I-Y)/2 observable.

These six are all the pure qubit states that we can simulate with a single elementary QSL system. In addition, we can represent the maximally mixed state that encodes that we have no knowledge at all about the system. The natural way to do that is to have both the computational and phase bit represented by two independent random variables.
(20)$$\left(R_{x},R_{p}\right)∼\frac{I}{2}$$
These seven QSL states are equivalent to Spekkens’ epistemic states [13]: the six pure states, and the maximally mixed state.

An elementary QSL system has a sample space of four discrete points, corresponding to the four ontic states in Spekkens’ model. A description with four boxes □□□□ appears in [13], such that
(21)$$\begin{array}{cccc} (x_{0},p_{0})=\; & \underset{0}{□}\underset{1}{□}\underset{2}{□}\underset{3}{□}∼|\psi \rangle , & (R_{x},R_{p})=\; & ▪▪▪▪∼\frac{I}{2},\\ (0,R)=\; & ▪▪□□∼|0\rangle , & (1,R)=\; & □□▪▪∼|1\rangle ,\\ (R,0)=\; & ▪□▪□∼|+\rangle , & (R,1)=\; & □▪□▪∼|-\rangle ,\\ (R,R)=\; & ▪□□▪∼|+i\rangle , & (R,R⊕1)=\; & □▪▪□∼|-i\rangle . \end{array}$$
Please note that associating the XOR value to the eigenspace of the (I-Y)/2 observable leads to a different sign convention in the last row, than that used in Ref. [13]. With $p_{0}$ as the least significant bit (LSB) they have the canonical labeling
(22)$$\begin{array}{cc} (x_{0},p_{0})= & \underset{0}{□}\underset{1}{□}\underset{2}{□}\underset{3}{□}\\ (0,0)= & 0\\ (0,1)= & 1\\ (1,0)= & 2\\ (1,1)= & 3. \end{array}$$

Perhaps the most used method of calculating the similarity between two quantum states $|\psi \rangle$ and $|\phi \rangle$ is the absolute square of their inner product, $|\langle \psi |\phi \rangle {|}^{2}$. This is called the *statistical overlap* or *transition probability* [34], and take on values between 0 and 1. If the states are parallel, then $|\langle \psi |\phi \rangle {|}^{2}=1$, and if they are orthogonal $|\langle \psi |\phi \rangle {|}^{2}=0$.

In QSL this quantity is the standard statistical overlap, or the complement of the *Kolmogorov distance*
(23)$$F^{2}\left(P,Q\right)=1-\delta \left(P,Q\right)=1-\frac{1}{2}\sum_{x\in \Omega }\left|P\left(x\right)-Q\left(x\right)\right|,$$
where *P* and *Q* are distributions describing the states. States with disjoint support are perfectly distinguishable in the same way as orthogonal states are perfectly distinguishable, while states with overlapping support are not.

Another popular measure of similarity is the *fidelity*, which in QSL is given as the square root of the statistical overlap,
(24)$$F\left(P,Q\right)=\sqrt{1-\delta (P,Q)}.$$
The quantum fidelity between two mixed states ρ and σ is defined in [30] to be
(25)$$F\left(\rho ,\sigma \right)=\mathrm{Tr}\left(\sqrt{\rho^{1/2}\sigma \rho^{1/2}}\right),$$
where Tr denotes the trace of the operator. If ρ is pure this simplifies to $\sqrt{\langle \rho |\sigma |\rho \rangle }$, and if both states are pure to $|\langle \rho |\sigma \rangle |$ (for a derivation, see [35]). For an elementary system the fidelity between QSL states is completely equivalent with that between quantum states. As an example, the fidelity between (0,R) and $\left(0,R^{\prime }\right)$ is 1, between (0,R) and $\left(1,R^{\prime }\right)$ is 0, and between (0,R) and any other state is $1/\sqrt{2}$ (including the maximally mixed).

Please note that the fidelity measure in QSL is not the Bhattacharyya coefficient
(26)$$B\left(P,Q\right)=\sum_{x\in \Omega }\sqrt{P(x)Q(x)}$$
that is usually taken as the classical analogue to quantum fidelity [36].

The quantum states of a qubit have a geometrical representation called the *Bloch sphere*. It is a unit ball where all pairs of antipodal points on the surface correspond to orthogonal pure states, and inside the surface resides the mixed states.

The QSL states in relation to the Bloch sphere is shown in Figure 4. Their positions are motivated by the relations of Equations (Equation 17)–(Equation 19) to mutually unbiased pairs of orthogonal quantum states. The states in each pair have disjoint support, and each state has a statistical overlap $F^{2}=1/2$ with any state in any other pair; they form mutually unbiased partitions of the state space.

#### 3.1.2. Transformations

Unitary transformations are reversible maps on quantum states. It is, therefore, natural to simulate these with operations from *classical reversible logic* on the bits composing a QSL state.

In identifying which transformations that correspond to which quantum gates, we start with the *X* transformation. This operation inverts the states of the computational basis and leaves the phase basis unchanged.
(27)$$X|0\rangle =|1\rangle ,\;X|1\rangle =|0\rangle ,\;X|\pm \rangle =|\pm \rangle$$
Therefore, the corresponding QSL gate flips the computational bit and leaves the phase bit unchanged (see Figure 5),
(28)$$X\left(x_{0},p_{0}\right)=\left(x_{0}⊕1,p_{0}\right)$$
or as the permutation (02)(13) in the cyclic notation.

In the same way, the *Z* gate inverts the phase basis and leaves the computational basis unchanged.
(29)$$Z|0\rangle =|0\rangle ,\;Z|1\rangle =|1\rangle ,\;Z|\pm \rangle =|\mp \rangle .$$
Hence, the QSL Z-gate flips the phase bit instead of the computational bit
(30)$$Z\left(x_{0},p_{0}\right)=\left(x_{0},p_{0}⊕1\right)$$
or (01)(23).

In QSL the states that correspond to eigenstates of *Y* can be singled out as those that preserve the parity between the computational and phase bit, $x_{0}⊕p_{0}$. The only way a transformation will preserve that while also inverting the computational and phase basis, is to flip both bits.
(31)$$Y\left(x_{0},p_{0}\right)=\left(x_{0}⊕1,p_{0}⊕1\right)$$
This corresponds to the permutation (03)(12). Figure 5 shows a graphical representation of these transformations. Blue line segments represent the computational bit and red represents the phase bit.

As in quantum theory these transformations uphold the identities
(32)$$X^{2}=Y^{2}=Z^{2}=I,$$
but composition of these gates also shows a difference between QSL and quantum theory. The identity
(33)XZ=-iY
is not upheld, but instead
(34)XZ=Y.

Another useful transformation is the Hadamard gate. This quantum gate obeys
(35)HZH=X,andHXH=Z
while also being an involution, $H^{2}=I$. In QSL the mapping that simulates this is the one that swaps the computational and phase bit.
(36)$$H\left(x_{0},p_{0}\right)=\left(p_{0},x_{0}\right).$$
This permutation, which in cyclic notation is given by (12), is identified as the analogue of the Hadamard gate in [37], and indeed,
(37)$$H^{2}=I,\;HZH=X,\;\mathrm{and}\;HXH=Z.$$
Please note that the analogy is not complete, because in quantum theory we have
(38)HYH=-Y,
while in QSL
(39)HYH=Y.

The last single qubit gate that we describe is the *S*-gate. This is the square root of the Z-gate,
(40)$$S^{2}=Z,$$
which has an order of four. By defining it as
(41)$$S\left(x_{0},p_{0}\right)=\left(x_{0}⊕1,p_{0}⊕x_{0}\right),$$
which is equivalent with the permutation (0213), and with
(42)$$S^{-1}\left(x_{0},p_{0}\right)=\left(x_{0}⊕1,p_{0}⊕x_{0}⊕1\right),$$
we see that Equation (Equation 40) and the quantum identities
(43)$$SXS^{†}=Y\;\mathrm{and}\;SZS^{†}=Z$$
are obeyed since
(44)$$S^{2}=Z,\;SXS^{-1}=Y,\;\mathrm{and}\;SZS^{-1}=Z,$$
but again, while we expect
(45)$$SYS^{†}=-X$$
QSL instead gives
(46)$$SYS^{-1}=X.$$

Figure 6 shows a graphical representation of the simulation of the Hadamard, *S*, and $S^{†}$ gates.

All of these transformations are generated by H and S (${Z=S}^{2}$, X=HZH, Y=XZ, $S^{-1}=XSX$), which also generate the group of transformations that Spekkens allows for in his model, the symmetric group on four symbols $S_{4}$ [38]. This is because S gives the 4-cycle (0213) and H gives the 2-cycle (12) of adjacent elements in that 4-cycle.

Another way to view this [13] is through the Bloch representation. Figure 7 shows the effect of X, Z, H, and S, on the six QSL states. We can see that the X and Z transformations have the expected effect, considering that in the Bloch sphere representation they can be viewed as $180^{\circ }$ rotations around the $\hat{x}$ and $\hat{z}$ directions respectively (see Figure 7A,B). However, the expected effect of the Hadamard gate is to also flip the states along $\hat{y}$, but this is not the case in QSL (see Figure 7C). Finally, the quantum *S*-gate correspond to a $90^{\circ }$ plane rotation of the Bloch sphere, while the QSL simulation also flips the states along $\hat{z}$ (see Figure 7C).

Spekkens recognizes this and shows that some transformations, such as those we relate to *H* and *S* gates, correspond to antiunitary maps in the Bloch sphere representation. Skotiniotis et al. [39] connected the subgroup of 12 even permutations $A_{4}$ to unitary maps, while the odd permutations correspond to antiunitary maps.

It is surprising that these faulty identities, or systematic errors, exist at the level of simulating qubits, but that the simulation can reproduce many phenomena even when the system size grows. There will be many examples of this in the rest of this paper.

#### 3.1.3. No Universal Spin-1/2 Inverter

A universal state inverter is a transformation that takes an arbitrary state ρ and produces an orthogonal state $\rho^{⊥}$. Such a device is not allowed by quantum theory [40]. For a pure qubit state the operation inverting the spin is given by a composition of *X*, *Z*, and complex conjugation $|\psi^{⊥}\rangle =XZ|\psi^{*}\rangle$ [41]. Here $|\psi^{*}\rangle$ represents the complex conjugated state. In the Bloch sphere representation the spin flip operation is such that
(47)$$\rho =\frac{I+\vec{r}\;\vec{\sigma }}{2}\to \rho^{⊥}=\frac{I-\vec{r}\;\vec{\sigma }}{2},$$
where $\vec{r}$ is the Bloch vector and $\vec{\sigma }=X\hat{x}+Y\hat{y}+Z\hat{z}$ is the Pauli vector. In general
(48)$$\rho^{⊥}=XZ\rho^{*}ZX,$$
where *Z* flips the sign of the $\hat{x}$-component in the Bloch vector, *X* flips the sign of the $\hat{z}$-component, and conjugation flips the sign of the $\hat{y}$-component. This is an antiunitary map and cannot be implemented with the dynamics of a quantum system.

In QSL, because of the existence of transformations that correspond to antiunitary maps in the Bloch sphere representation, one may be tempted to think that a universal spin-1/2 inverter could be possible. This is not the case, because a device that takes an arbitrary QSL state and produces the disjoint state must obey the following three conditions.

Map the eigenstates of X to each other. This is done by flipping the phase bit while doing nothing to the computational bit.Map the eigenstates of Z to each other. This is done by flipping the computational bit while doing nothing to the phase bit.Map the eigenstates of Y to each other. This requires changing the parity between the computational and phase bit.

The third condition cannot be obeyed without violating the other two.

#### 3.1.4. Measurement

A measurement in QSL consists of information retrieval followed by a state update to ensure that only one bit of information can be known about an elementary system. This is completely equivalent to Spekkens’ model, and simulates measurement disturbance, or the update of a quantum state after a measurement (sometimes called collapse). The specifics are as follows.

A projective measurement of Z in QSL returns the computational bit and randomizes the phase bit, while measuring the X observable instead reads the phase bit and randomizes the computational bit. Remember that the phase bit value 0 is associated with a positive relative phase between the computational states, and the phase bit value 1 is associated with a negative relative phase. Finally, measuring Y returns the parity of the computational and phase bit and then randomizes both the computational and phase bits while preserving parity, that is, if measuring Y yields $x_{0}⊕p_{0}=0$, then the system is updated randomly to either (0,0) or (1,1), and if $x_{0}⊕p_{0}=1$, the system is updated randomly to either (0,1) or (1,0).

This ensures repeatability of measurements, i.e., if a projective measurement is repeated, the second measurement will yield the same outcome as the first. It also ensures measurement disturbance, prohibiting that the bit-values not measured can be predicted after the measurement. The connection with mutual unbiasedness is clear. Information retrieved from one of these measurements is unrelated to the information retrieved from the other. That is, we can retrieve either the computational bit, the phase bit, or the parity between them, never two of them, or indeed three.

#### 3.1.5. Preparation

To prepare a system that simulates $|0\rangle$ or $|1\rangle$, we create a system with the computational bit $x_{0}=0$ or $x_{0}=1$ respectively, and the phase bit is chosen randomly. In the same way we can prepare QSL states simulating $|+\rangle ,|-\rangle ,|+i\rangle$ and $|-j\rangle$ by distributing $p_{0}$ and $x_{0}$ according to the respective distribution.

Preparation by measurement can also be performed, by taking a maximally mixed state (the state of *no knowledge*, with both computational and the phase bit randomized), measuring some observable to obtain information on the state, and then transforming the now known state into the desired state.

#### 3.1.6. Non-Commutativity of Measurements

In quantum theory the order of measurements performed is of importance to their outcomes. Consider for instance that we have prepared a quantum state $|0\rangle$ and then we measure the observables *X* and *Z*. If *Z* is measured first it will produce the outcome 0 with certainty, and then *X* will randomly yield the outcome 0 (for $|+\rangle$) or 1 (for $|-\rangle$) with equal probability. If the order of the measurements is reversed, both measurements of *X* and *Z* will instead produce independent random outcomes.

In QSL the outcomes are, for this particular protocol, in complete compliance with quantum theory. To simulate the above procedure, we start by preparing the QSL state
(49)(0,R).
Measuring Z will return the computational bit which will always be 0, and randomize the already random phase bit, so that the state changes from (0,R) to $\left(0,R^{\prime }\right)$. Measurement of X will then return the new random value $R^{\prime }$ of the phase bit and randomize the computational bit. If we instead start by measuring X the outcome will be the random value *R* while the computational bit is randomized so that the state changes from (0,R) to $\left(R^{\prime \prime },R\right)$. A subsequent measurement of *Z* will retrieve $R^{\prime \prime }$.

#### 3.1.7. QKD—BB84

Having built up the model for simulating single qubit systems, let us now consider a cryptographic protocol that only requires a single qubit system. In 1984, Bennett and Brassard [42] presented a protocol for distributing a cryptographic key, with the property that an eavesdropper can in principle be detected, and then the compromised key can be discarded.

The setup is shown in Figure 8 and the protocol proceeds as follows. In one round of the protocol Alice wants to send a random bit *b* to Bob. Alice encodes this bit in a one of two bases, {$|0\rangle$,$|1\rangle$} or {$|+\rangle$,$|-\rangle$}. The bit-value b=0 is encoded in either $|0\rangle$ or $|+\rangle$, and b=1 into $|1\rangle$ or $|-\rangle$. Alice chooses the encoding randomly, and Bob chooses to perform a measurement in one of these two bases, also randomly. If Bob chooses the same encoding as Alice, then he receives the bit *b*, otherwise he retrieves 0 or 1 with equal probability.

This procedure is now repeated several times, and after a sufficiently long sequence they stop and share the encoding they used. Both Alice and Bob can now deduce in which rounds of the protocol their encoding were different, and discard data from these rounds. This process is known as sifting. The remaining data of Alice is now equal to that of Bob (in the ideal case); they each have a copy of the same random bit-string.

If someone (usually denoted Eve) is trying to eavesdrop during this process, the measurement disturbance will introduce noise in the key. Eve is assumed to have no knowledge about the random encoding used by Alice, so Eve’s best strategy is to guess the encoding at random. If Eve’s guess is correct, the quantum state is unaffected by her measurement, but if Eve’s guess is wrong, she only gets random data and her measurement device outputs a state in the wrong basis. For example, if Alice sends the state $|0\rangle$, and Eve picks the phase basis $\left\{|+\rangle ,|-\rangle \right\}$, then Eve’s measurement outcome *e* is random, and her measurement will update the quantum state into
(50)$$\frac{|0\rangle +{\left(-1\right)}^{e}|1\rangle }{\sqrt{2}}.$$
When Bob then measures in the computational basis there is a 50% chance of creating a bit-error in the key.

An eavesdropper can now be detected if Alice and Bob sacrifice part of the key and compare it between themselves. If the error rate is too high (non-zero in the ideal case), the whole key should be discarded because Eve has been eavesdropping. If the error rate is sufficiently low (zero in the ideal case), they can be confident that no one have listened in on their communication. In a real system with noise and lost qubits this is more complicated, and there are also ways to handle limited eavesdropping, but here we will restrict ourselves to the ideal case.

The simulation of the protocol in QSL can be exemplified as follows. If Alice chooses the computational basis state, her bit *b* is encoded into the QSL state
(51)(b,R).
If she instead chooses the phase basis, her bit *b* is encoded in the QSL state
(52)(R,b).
If Bob chooses to measure in the same basis as Alice has used for encoding, he will retrieve the bit *b*, otherwise he will retrieve the random value *R*, which will then be discarded in the sifting step.

If Eve is eavesdropping, there is a 50% chance that she chooses a different encoding than Alice, in which case she retrieves the random value *R*, and through the measurement disturbance mechanism, she also randomizes the bit-value *b*. In this case, Bob will receive a random bit-value (independent of *b*), irrespective of in which encoding he measures. This reproduces the quantum behavior of BB84. In other words, the BB84 protocol including the described attack, known as the intercept-resend attack, can be faithfully simulated in QSL.

It is important to note that the security fails if there exists a measurement that retrieves information without disturbing the system. In QSL we have included a restriction on the allowed measurements to simulate measurement disturbance. In an actual implementation of QSL, using physical bits, this restriction could be ignored by an adversary, and a measurement that does not disturb the system be performed. In quantum theory there is no such measurement [43], but if an actual implementation of BB84 does not use an ideal qubit there may be such a measurement. A well-known example is when failing to make a single-photon source so that in each round, the information is encoded onto several photons. In this case, an eavesdropper could split off two of the many photons, and measure separately on them, in the two possible encodings. This would reveal the information present in one of the encodings (Eve will learn which is correct in the sifting phase), while not disturbing the remaining photons at all, so that no noise is present in the generated key. This attack, known as the photon-number-splitting attack, would be equivalent to an adversary ignoring the restriction of QSL.

### 3.2. Pairs of Elementary Systems

In this section, we are at most considering pairs of qubits, i.e., 2×2-dimensional systems.

QSL systems compose under the Cartesian product, and for two elementary systems we write
(53)$$\left(x_{1},p_{1}\right)\times \left(x_{0},p_{0}\right)=\left(x_{1},p_{1}\right)\;\left(x_{0},p_{0}\right),$$
the latter notation for brevity, which we could equivalently write
(54)$$\left(x,p\right)=\left(2x_{1}+x_{0},2p_{1}+p_{0}\right)$$
so that *x* and *p* can take the values {0,1,2,3}. This gives a pair of QSL systems a sample space of 16 discrete points.

States that cannot be described as convex combinations of other states corresponds to pure states, and the maximally mixed state is still represented by a uniform distribution over all 16 points.

#### 3.2.1. Transformations

For pairs of elementary systems, we also have gates corresponding to two-qubit gates. The relations in Figure 2 motivates the following construction of a QSL analogue of the quantum CNOT, and one can verify that this construction is equivalent with the construction in [13].
(55)$$\mathrm{CNOT}\left(x_{1},p_{1}\right)\;\left(x_{0},p_{0}\right)=\left(x_{1},p_{1}⊕p_{0}\right)\;\left(x_{0}⊕x_{1},p_{0}\right),$$
see Figure 9A for a graphical representation.

Controlled-Z (CZ) and *SWAP* follow directly from the identities
(56)CZ=(I⊗H)CNOT(I⊗H)
(57)SWAP=CNOT(H⊗H)CNOT(H⊗H)CNOT
(see Figure 9B,C). This implies the QSL maps
(58)$$\mathrm{CZ}\left(x_{1},p_{1}\right)\;\left(x_{0},p_{0}\right)=\left(x_{1},p_{1}⊕x_{0}\right)\;\left(x_{0},p_{0}⊕x_{1}\right)$$
and
(59)$$\mathrm{SWAP}\left(x_{1},p_{1}\right)\;\left(x_{0},p_{0}\right)=\left(x_{0},p_{0}\right)\;\left(x_{1},p_{1}\right).$$

#### 3.2.2. Entanglement

The QSL analogue to product states are states where the information about one subsystem is independent of the information about the other subsystem. In contrast, in entangled states the information is stored in the correlations between the pair of QSL systems.

In quantum theory, a measure of the amount of entanglement in a bipartite system is the entropy of entanglement. This is obtained by calculating the von Neumann entropy of the reduced density operator $\rho_{A}={\mathrm{tr}}_{B}\left(\rho_{AB}\right)$, where $\rho_{AB}$ is the state of the composite system A⊗B, and ${\mathrm{tr}}_{B}$ is the partial trace taken over the subspace related to system *B*. As an example, consider the Bell state $1/\sqrt{2}\left(|01\rangle +|10\rangle \right)$ with density operator
(60)$$\rho =\frac{\left|01\rangle \langle 01\right|+\left|01\rangle \langle 10\right|+\left|10\rangle \langle 01\right|+\left|10\rangle \langle 10\right|}{2}.$$
The reduced density operator for system *A* is the maximally mixed state,
(61)$$\begin{array}{cc} \rho_{A}= & \frac{\left|0\rangle \langle 0\right|\langle 1|1\rangle +\left|0\rangle \langle 1\right|\langle 1|0\rangle +\left|1\rangle \langle 0\right|\langle 0|1\rangle +\left|1\rangle \langle 1\right|\langle 0|0\rangle }{2}=\frac{\left|0\rangle \langle 0\right|+\left|1\rangle \langle 1\right|}{2}=\frac{I}{2}. \end{array}$$

This makes the entropy of entanglement equal to 1, and means that even though we have a pure bipartite state (of entropy 0) we obtain a maximally mixed state (of entropy 1) for the subsystem *A*. The entropy of entanglement is symmetric in its arguments, so the uncertainty about the state of subsystem *A* equals that of subsystem *B*, $S\left(\rho_{A}\right)=S\left(\rho_{B}\right)$. Therefore, we have maximal information about the whole system but no information about the individual parts. Thus, in a sense all information is stored in the correlations of the pair. States with this property are called maximally entangled. The Bell states
(62)$$|\Psi^{\pm }\rangle =\frac{|00\rangle \pm |11\rangle }{\sqrt{2}}\;\mathrm{and}\;|\Phi^{\pm }\rangle =\frac{|01\rangle \pm |10\rangle }{\sqrt{2}},$$
are maximally entangled states. These can be generated from the following recipe (see Figure 10).
Prepare a pair of qubits in the state $|a\rangle |b\rangle$, where ab is 00 for $|\Psi^{+}\rangle$, 01 for $|\Phi^{+}\rangle$, 10 for $|\Psi^{-}\rangle$, and 11 for $|\Phi^{-}\rangle$.Apply a Hadamard gate to the first qubit.Apply a *CNOT* with the first qubit as control and the second as target.

We can use this recipe also in QSL to produce the states simulating the Bell states in Equation (Equation 62). As an example, the state simulating $\Phi^{+}$ is
(63)$$\begin{array}{ccc} \left(0,R\right)\;\left(1,R^{\prime }\right) & \overset{H\times I}{\to } & \left(R,0\right)\;\left(1,R^{\prime }\right)\\ & \overset{\mathrm{CNOT}}{\to } & \left(R,R^{\prime }\right)\;\left(\bar{R},R^{\prime }\right). \end{array}$$
For the whole state there are 2 bits of uncertainty, and is therefore a state of maximal knowledge since it is composed by two elementary subsystems. Viewing one of the individual subsystems, and ignoring the other, there is also two bits of uncertainty and therefore, viewed individually, they are maximally mixed.

The four QSL states related to Bell states are
(64)$$\begin{array}{c} |\Psi^{+}\rangle ∼\left(R,R^{\prime }\right)\;\left(R,R^{\prime }\right)\\ |\Psi^{-}\rangle ∼\left(R,\bar{R^{\prime }}\right)\;\left(R,R^{\prime }\right)\\ |\Phi^{+}\rangle ∼\left(R,R^{\prime }\right)\;\left(\bar{R},R^{\prime }\right)\\ |\Phi^{-}\rangle ∼\left(R,\bar{R^{\prime }}\right)\;\left(\bar{R},R^{\prime }\right), \end{array}$$
where *R* and $R^{\prime }$ are independent randomly generated bits. These states are equivalent with those used by Spekkens, but represented differently. Spekkens represents the state of a pair of elementary systems by their joint probability distribution. As an example,
(65)$$\begin{array}{c} □□□▪\\ □□▪□\\ □▪□□\\ ▪□□□ \end{array}$$
shows the joint probability distribution corresponding to $|\Psi^{+}\rangle$. The marginal probability distributions for each elementary system are uniform, corresponding to maximally mixed states; in their marginal distributions there is no information about the state.

#### 3.2.3. Remote Steering

Suppose that Alice and Bob each possess one qubit of a pair in the state
(66)$$\frac{|00\rangle +|11\rangle }{\sqrt{2}}=\frac{|++\rangle +|--\rangle }{\sqrt{2}}.$$
This is sometimes expressed as Alice and Bob sharing an entangled state. If Bob measures his qubit in the computational basis $\left\{|0\rangle ,|1\rangle \right\}$, then he obtains the outcome for $|0\rangle$ and $|1\rangle$ with equal probability. If he obtains $|0\rangle$, the state updates according to
(67)$$\frac{|00\rangle +|11\rangle }{\sqrt{2}}\to |00\rangle$$
and if he gets $|1\rangle$ it updates according to
(68)$$\frac{|00\rangle +|11\rangle }{\sqrt{2}}\to |11\rangle .$$
In both cases he knows the state of Alice’s qubit, but if Bob instead chooses to measure in the phase basis, he will still get outcomes for $\left\{|+\rangle ,|-\rangle \right\}$ at random and with equal probability. If the outcome is $|+\rangle$ or $|-\rangle$ the state updates to
(69)$$\frac{|00\rangle +|11\rangle }{\sqrt{2}}\to |++\rangle \;\mathrm{or}\;\frac{|00\rangle +|11\rangle }{\sqrt{2}}\to |--\rangle$$
respectively. In either case Bob will learn the state of Alice’s system, but he also notes that it will be different if he chose to measure in the computational or phase basis. This phenomenon, that Bob apparently can influence Alice’s state by the choice of his measurement basis, is known as remote steering.

In QSL this is simulated as follows. Alice and Bob each get one elementary system from a pair initiated in the state
(70)$$|\Psi^{+}\rangle ∼\left(R,R^{\prime }\right)\;\left(R,R^{\prime }\right).$$

If Bob measures Z, he learns the value of *R*, and thus also the value of the computational bit in Alice’s system. If *R* was 1 (there is a 50% probability of that being the case) the state updates to
(71)$$\left(1,R^{\prime \prime }\right)\;\left(1,R^{\prime }\right),$$
and accordingly, if the outcome is 0, here $R^{\prime \prime }$ is a new i.i.d. random bit generated by the measurement. Note that this also terminates the last bit of correlation between the two systems so that there is no correlation between them after the measurement. If Bob instead measures X, he learns the value of $R^{\prime }$, and thus the value of the phase bit in Alice’s system. If the outcome is 1 the state updates to
(72)$$\left(R^{\prime \prime },1\right)\;\left(R,1\right).$$
and accordingly, if the outcome is 0. In QSL, and in Spekkens’ model, the measurement choice of Bob is not influencing Alice’s system—the measurement choice is influencing Bob’s knowledge about Alice’s system.

#### 3.2.4. Anticorrelation in Spin-Measurements of the Singlet

Another closely related example that instead does not work in Spekkens’ model or QSL is complete anticorrelation within the singlet. If a single-system spin measurement is performed in any direction, $\vec{r}\;\vec{\sigma }$, on both qubits in the singlet state
(73)$$|\Phi^{-}\rangle =\frac{|01\rangle -|10\rangle }{\sqrt{2}},$$
they will always output opposite values. The state simulating the singlet is
(74)$$\left(R,\bar{R^{\prime }}\right)\;\left(\bar{R},R^{\prime }\right),$$
where the overline denotes the Boolean complement. We can see that measuring Z,X and Y on both systems will return, in order
(75)$$\begin{array}{cc} R\; & \mathrm{and}\;\bar{R}\\ \bar{R^{\prime }}\; & \mathrm{and}\;R^{\prime }\\ R⊕\bar{R^{\prime }}\; & \mathrm{and}\;\bar{R}⊕R^{\prime }. \end{array}$$
The first two measurements Z and X will produce opposite outcomes, but the last, Y, will produce equal outcomes (see the last line of expression (Equation 75)). Thus, even though we are restricted to only three measurements, QSL does *not* reproduce this phenomenon.

There are three more examples such as this; one for each Bell state, with different relations between the correlations that can be seen from measurements of *X*, *Y*, and *Z* on both qubits. In QSL and Spekkens’ however, measurement of Y will always show opposite correlations as those in quantum theory, just as in the above example (see further [13]).

#### 3.2.5. No-Cloning

No-cloning is the no-go theorem stating that there exists no unitary transformation that takes an arbitrary state $|\psi \rangle$ and an auxiliary qubit $|a\rangle$, and returns both systems in the state $|\psi \rangle$ [44]
(76)$$|\psi \rangle |a\rangle \to |\psi \rangle |\psi \rangle .$$
A unitary transformation preserves the inner product, so if we take two quantum states $|\psi \rangle$ and $|\psi^{\prime }\rangle$ and apply a hypothetical cloner,
(77)$$\begin{array}{cc} |\psi \rangle |a\rangle & \to \;|\psi \rangle |\psi \rangle \\ |\psi^{\prime }\rangle |a\rangle & \to \;|\psi^{\prime }\rangle |\psi^{\prime }\rangle . \end{array}$$
Requiring the inner product to be preserved we have
(78)$$\langle \psi |\psi^{\prime }\rangle \langle a|a\rangle ={\left(\langle \psi |\psi^{\prime }\rangle \right)}^{2}.$$
Since $\langle a|a\rangle =1$ this implies that the norm of these two scalars is equal,
(79)$$|\langle \psi |\psi^{\prime }\rangle \left|\;=\;\right|\langle \psi |\psi^{\prime }\rangle {|}^{2},$$
which only happens if the states are parallel or orthogonal. Therefore, there exists no single unitary that performs the task for an arbitrary state.

In QSL we have two elementary systems in pure states T,W, and an auxiliary system in a pure state *A*. The transformation that clones these states is
(80)$$\begin{array}{cc} T\times A & \to \;T\times T\\ W\times A & \to \;W\times W. \end{array}$$

For this to be reversible, the number of elements in the joint support of T×A and W×A needs to be preserved by the transformation, because if the support grows or shrinks it is not a bijection. Before the transformation, the number of elements in the joint support is
(81)$$\begin{array}{cc} |S(T\; & \times A)\cup S(W\times A)|=|S(T\times A)|+|S(W\times A)|-|S(T\times A\cap W\times A)|\\ & =|S(T)||S(A)|+|S(W)||S(A)|-|S(T\times A\cap W\times A)|=8-|S(T\times A\cap W\times A)|\\ & =8-|S((T\cap W)\times A)|=8-|S(T\cap W)||S(A)|=8-2|S(T\cap W)|, \end{array}$$
where we have used that the support of a pure state has 2 elements, i.e., |S(A)|=2. The number of elements in the joint support after the transformation is
(82)$$\begin{array}{cc} |S(T\; & \times T)\cup S(W\times W)|=|S(T\times T)|+|S(W\times W)|-|S(T\times T\cap W\times W)|\\ & ={\left|S\left(T\right)\right|}^{2}+{\left|S\left(W\right)\right|}^{2}-\left|S\left(T\times T\cap W\times W\right)\right|=8-\left|S\left(T\times T\cap W\times W\right)\right|\\ & =8-|S(T\cap W\times T\cap W)|=8-|S(T\cap W)||S(T\cap W)|. \end{array}$$
This gives us the condition (compare with Equation (Equation 79))
(83)$$2|S\left(T\cap W\right)|={\left|S\left(T\cap W\right)\right|}^{2},$$
but this is only fulfilled when *T* and *W* are disjoint (orthogonal) or completely overlap (parallel). Therefore, in a setup with two QSL systems, there is no single QSL transformation that performs the task for an arbitrary state.

#### 3.2.6. Interference

Examples of interference does not require a bipartite system, but it makes for a good example of how it is used in quantum information processing.

Consider the protocol in Figure 11. Prepare a two-qubit system in the state $|01\rangle$, and apply a Hadamard gate to each of them
(84)$$|01\rangle \overset{H⊗H}{\to }\frac{|0\rangle \left(|0\rangle -|1\rangle \right)+|1\rangle \left(|0\rangle -|1\rangle \right)}{2}.$$
Then a unitary operation that encodes a reversible function in the computational basis is applied.
(85)$$\frac{|0\rangle (|f(0)\rangle -|\bar{f(0)}\rangle )+|1\rangle (|f(1)\rangle -|\bar{f(1)}\rangle )}{2}.$$
Even though the unitary encoding the function is used only once, information about more than a single function values (in this case two) appears in the state. This phenomenon is sometimes called quantum parallelism. The state in expression (Equation 85) also equates to
(86)$$\frac{{\left(-1\right)}^{f(0)}|0\rangle |-\rangle +{\left(-1\right)}^{f(1)}|1\rangle |-\rangle }{\sqrt{2}},$$
and after the final Hadamard it becomes
(87)$$\frac{{\left(-1\right)}^{f(0)}|+\rangle |-\rangle +{\left(-1\right)}^{f(1)}|-\rangle |-\rangle }{\sqrt{2}}.$$
Measuring the observable $\left|0\rangle \langle 0\right|⊗I$, i.e., testing whether the first qubit is still in the initial state $|0\rangle$, will test positive with probability
(88)$${\left|\frac{{\left(-1\right)}^{f(0)}+{\left(-1\right)}^{f(1)}}{2}\right|}^{2}.$$
Therefore, if f(0)≠f(1) amplitudes will interfere destructively, and the test will be positive with probability zero. If f(0)=f(1) they interfere constructively, and the test will be positive with unit probability. This is known as Deutsch algorithm [9], and considered by many the starting point of quantum algorithm research.

There are four functions over 1 bit: f(x)=0,1,x, and x⊕1. The canonical way of constructing those with reversible logic follows. The first two, f(x)=0 and f(x)=1, are constructed by doing nothing at all respectively applying a NOT gate to the output. For the last two, f(x)=x and f(x)=x⊕1, an additional *CNOT* is applied between the input and output. The quantum unitary $U_{f}$ would then be realized in four ways, as the identity, an *X* gate on the second qubit only, a *CNOT*, or a *CNOT* followed by an *X* gate on the second qubit.

In QSL, the protocol proceeds as follows. Prepare two systems simulating $|01\rangle$, and apply to each system
(89)$$\left(0,R\right)\;\left(1,R^{\prime }\right)\overset{H\times H}{\to }\left(R,0\right)\;\left(R^{\prime },1\right).$$
The effect of the above discussed construction of $U_{f}$ in QSL will map the function from the first to the second system over the computational bits. It will also add the phase of the second system into the first system if the *CNOT* is present (that is, if f(0)⊕f(1)=1). This simulates the phenomenon sometimes called phase kickback [32]. For the other two functions (when f(0)⊕f(1)=0) there will be no phase kickback. Therefore, not knowing which function that is implemented, the natural way of describing the effect of the phase kickback is to use the parity of the function. This gives the map
(90)$$\left(x,p\right)\;\left(y,z\right)\to \left(x,p⊕z(f(0)⊕f(1)\;)\right)\;\left(y⊕f(x),z\right).$$
Applying the function to the result of the map in expression (Equation 89) we get
(91)$$\left(R,0\right)\;\left(R^{\prime },1\right)\overset{U_{f}}{\to }\left(R,f(0)⊕f(1)\right)\;\left(R^{\prime }⊕f\left(R\right),1\right),$$
and then applying the last transformation H⊗I gives
(92)$$\left(f(0)⊕f(1),R\right)\;\left(R^{\prime }⊕f\left(R\right),1\right).$$
Measuring the observable $\left|0\rangle \langle 0\right|⊗I$ corresponds to testing whether the computational bit of the first system is zero or not. From the state in expression (Equation 92) this translates to testing
(93)$$f\left(0\right)⊕f\left(1\right)\overset{?}{=}0,$$
which is true only when f(0) and f(1) are equal, and false when they are not, just as in quantum theory. Figure 12 shows the explicit instance of the Deutsch algorithm in QSL when f(x)=x.

This is a faithful simulation of a protocol showing interference and quantum parallelism. Even in QSL there is an apparent effect of information from both function values being added to the result, while only one query is made (see the QSL map $U_{f}$ in Equation (Equation 90)). Now, whether a quantum-computer access all function values with only one query, or whether the equations only describe the structure of additional information that can be retrieved, is an unsettled philosophical debate. For QSL the latter is certainly the case, but we should also stress that QSL is *not* quantum theory. Yet, since one process faithfully simulates the other, these processes are operationally equivalent, and in that spirit a machinery running this particular QSL recipe should not be distinguished from one running the quantum recipe.

We need to stress that in QSL we do *not* calculate the parity of the function f(0)⊕f(1) by accessing the function twice, and then have the oracle signal that information. The expression using the parity of the function just turns out to be a good description for the additional information available when we do not know which of the four functions that was applied. With the constructions of f(x)=x and f(x)=x⊕1, the phase bit of the first system is flipped, but not for the constructions for f(x)=0 and f(x)=1, i.e., the output depends on the choice of function made in the construction.

To further clarify, with “operationally equivalent” we mean that processes, or theories are equivalent when only judged by their input/output behavior. Therefore, a machine running Deutsch algorithm in QSL is operationally equivalent to the corresponding process on a quantum machine, but comparing QSL as a theory against quantum theory is not. In fact, we have already seen examples that QSL is operationally different, for instance in the correlations seen in the outcomes from Pauli measurement over Bell states. However, our main goal here is not to fuel the philosophical debate, nor to perfectly simulate the whole of quantum theory, but only to simulate it accurate enough to solve the computational problem.

#### 3.2.7. Measurements

For one elementary system we have defined three measurements that we relate to the Pauli observables. We also have six observables for the orthogonal projections onto their eigenstates, as exemplified in the previous section, and we can ask whether the computational bit is 0 or not (corresponding to the observable $\left|0\rangle \langle 0\right|$), rather than asking if it is 0 or 1 (Corresponding to the observable *Z*). When measuring a single elementary system there is no distinction between asking if it is 0 or not, or 0 or 1, but for higher-dimensional systems there is a distinction. For instance, say that the dimension is four and that we measure the observable $\left|0\rangle \langle 0\right|$, with the outcome that the system is not in $|0\rangle$, we cannot infer whether it is $|1\rangle$, $|2\rangle$, or $|3\rangle$. In QSL a measurement of $\left|0\rangle \langle 0\right|$ corresponds to asking the system if all the computational bits are zero or not. A measurement that distinguishes between $|0\rangle$, $|1\rangle$, $|2\rangle$, or $|3\rangle$ corresponds to asking the system whether the computational bits encode 0, 1, 2, or 3 respectively.

For systems composed of two elementary systems, we have seen that single system measurements act locally. In the Pauli group of observables there are also joint measurements of the kind $\sigma_{i}⊗\sigma_{j}$, and in QSL these are simulated as follows.

A measurement of the Z×Z observable returns the correlation between the computational bits of both systems, $x_{1}⊕x_{0}$, and redistributes according to the outcome. That is, if $x_{1}⊕x_{0}=0$ the computational bits of both systems will be random but completely correlated after the measurement, and if $x_{1}⊕x_{0}=1$ they will be random but anti-correlated. A measurement represented by X×X does the same, but for the phase bits. The procedure is similar for Z×X, for the computational bit of the first system and with the phase bit of the second, and so on.

These joint measurements retrieve 1 bit of information even though the whole system is composed of two elementary systems, and we should therefore be able to retrieve two bits, implying that the joint measurements are non-maximally informative measurements. In quantum theory this relates to when an observable has degenerate eigenvalues. For instance, the observable quantity X⊗X can take on two different values, and knowing this value gives us 1 bit of information.

A Bell-state measurement is a measurement that distinguish the four Bell states. This is a maximally informative measurement since it takes the maximally mixed state into a pure state (one of the Bell states). In QSL, an analogue to this measurement returns the correlation between the two computational bits *and* between the two phase bits. An example of this will be given in Section 3.2.8.

This is not to be confused with measurements whose outcome violate Bell inequalities—measurements included in a Bell test. For instance the CHSH inequality has a classical bound of 2, and quantum theory can violate this to a maximal value of $2\sqrt{2}$, called the Tsirelson bound. An experiment that maximally violates the inequality can be constructed from measuring the observables
(94)XorZ
on one qubit, and
(95)$$\frac{Z-X}{\sqrt{2}}\;\mathrm{or}\;\frac{-Z-X}{\sqrt{2}}$$
on the other, starting with a system in the singlet state. For a complete description see Nielsen and Chuang [30]. In Hilbert space the observables in expression (Equation 95) has eigenbases that are offset with $22.5^{\circ }$ from the eigenbases of those in expression (Equation 94). In QSL we do not have measurements with this relationship between each other.

Therefore, in QSL we have maximally entangled states, but cannot violate Bell inequalities. Obtaining a violation from QSL, which is a local realist model, would contradict Bell’s theorem [3].

Another phenomenon that QSL cannot reproduce is the contextual correlations in the Peres–Mermin square, which is an explicit example of the Kochen–Spekker theorem [4]. This was shown by Pusey [37], not in the QSL representation, but with his stabilizer representation of Spekkens’ model. However, Kleinmann et al. [5] initiated work on extending Spekkens’ model to reproduce the contextual correlations that can be seen from Pauli group measurements. Their work was later concluded by Harrysson [45]. It is also interesting to note that correlations seen from measurement sequences in the Peres–Mermin square have an efficient simulation according to the Gottesman–Knill theorem [12,46].

#### 3.2.8. Superdense Coding

Superdense coding is the name of a protocol that in a sense allows one party, Alice, to convey two bits of information $m_{1},m_{0}$ to the other party, Bob, by only interacting with one qubit. It is, however, a two-qubit protocol as shown by Figure 13.

First create the state $|\Psi^{+}\rangle$, give one of the two qubits to Alice and the other qubit to Bob. Depending on which of the four messages that Alice wants to send to Bob, she applies *I*, *X*, *Z*, or *Y* to her qubit, creating
(96)$$\begin{array}{cc} I⊗I\frac{1}{\sqrt{2}}\left(|00\rangle +|11\rangle \right) & =\frac{1}{\sqrt{2}}\left(|00\rangle +|11\rangle \right)\\ X⊗I\frac{1}{\sqrt{2}}\left(|00\rangle +|11\rangle \right) & =\frac{1}{\sqrt{2}}\left(|10\rangle +|01\rangle \right)\\ Z⊗I\frac{1}{\sqrt{2}}\left(|00\rangle +|11\rangle \right) & =\frac{1}{\sqrt{2}}\left(|00\rangle -|11\rangle \right)\\ Y⊗I\frac{1}{\sqrt{2}}\left(|00\rangle +|11\rangle \right) & =\frac{1}{\sqrt{2}}\left(|10\rangle -|01\rangle \right). \end{array}$$
She then sends her qubit to Bob. A Bell-state measurement will allow Bob to perfectly distinguish the state of the pair and deduce Alice’s message.

In QSL the state simulating $|\Psi^{+}\rangle$ is $\left(R,R^{\prime }\right)\;\left(R,R^{\prime }\right)$, and when Alice applies I,X,Z or Y the result is
(97)$$\begin{array}{cc} I\times I\;\left(R,R^{\prime }\right)\;\left(R,R^{\prime }\right) & =\;\left(R,R^{\prime }\right)\;\left(R,R^{\prime }\right),\\ X\times I\;\left(R,R^{\prime }\right)\;\left(R,R^{\prime }\right) & =\;\left(R,\bar{R^{\prime }}\right)\;\left(R,R^{\prime }\right),\\ Z\times I\;\left(R,R^{\prime }\right)\;\left(R,R^{\prime }\right) & =\;\left(\bar{R},R^{\prime }\right)\;\left(R,R^{\prime }\right),\;\mathrm{or}\\ Y\times I\;\left(R,R^{\prime }\right)\;\left(R,R^{\prime }\right) & =\;\left(\bar{R},\bar{R^{\prime }}\right)\;\left(R,R^{\prime }\right) \end{array}$$
respectively. As previously stated, a Bell-state measurement returns the XOR (the correlation) between the two computational bits and the two phase bits. This measurement on the above states will yield 00, 01, 10, or 11, respectively. Thus, we also have superdense coding in QSL.

The reason that Alice can convey a 2-bit message to Bob by sending him one elementary QSL system is that each QSL system contains 2 bits. It is true that Bob cannot access both bit-values, since he is restricted to only retrieving 1 bit of information from each system. He can retrieve both bits of the message only because there is a second system, initially highly correlated with the first, and where the correlations are manipulated by Alice.

### 3.3. Higher Number of Elementary Systems

Going to higher number of elementary systems, QSL departs from Spekkens’ toy model to become a less restrictive theory. Systems still compose under the Cartesian product, and the following notation is used
(98)$$\begin{array}{cc} \left(x_{n-1},p_{n-1}\right) & \;\ldots \;\left(x_{1},p_{1}\right)\;\left(x_{0},p_{0}\right)\\ & =\left(\sum 2^{i}x_{i},\sum 2^{i}p_{i}\right)=\left(x,p\right). \end{array}$$
Sometimes we will use a separation between different registers with the notation $\left(x,p\right)\;\left(x^{\prime },p^{\prime }\right)$, where *x*, *p*, $x^{\prime }$ and $p^{\prime }$ will be integers modulo a power-of-two.

#### 3.3.1. Teleportation

Before we introduce the new transformations, let us take one more example with more than two qubits.

Teleportation is the name of a protocol where given some pre-shared entanglement, Bob can recreate a qubit state $|\psi \rangle$ made available to Alice, from information from Alice on the result of a Bell-state measurement performed by her.

The protocol is shown in Figure 14, and proceeds as follows. Alice and Bob share a qubit pair in the Bell state $|\Psi^{+}\rangle$ created by the *H* and a *CNOT* gates at the left of Figure 14,
(99)$$|00\rangle |\psi \rangle \to \frac{|00\rangle +|11\rangle }{\sqrt{2}}|\psi \rangle .$$
Alice then correlates the state $|\psi \rangle =a|0\rangle +b|1\rangle$ that she wants to teleport to Bob, with her part of the pair. This is done using a Bell-state measurement built from the next *CNOT* and *H* gates.
(100)$$\begin{array}{c} \frac{a\left(|00\rangle +|11\rangle \right)|0\rangle +b\left(|00\rangle +|11\rangle \right)|1\rangle }{\sqrt{2}}\to \frac{a\left(|00\rangle +|11\rangle \right)|0\rangle +b\left(|01\rangle +|10\rangle \right)|1\rangle }{\sqrt{2}}. \end{array}$$
After the Hadamard, the state can be written as
(101)$$\begin{array}{c} \frac{\left(a|0\rangle +b|1\rangle \right)|00\rangle +\left(a|0\rangle -b|1\rangle \right)|01\rangle }{\sqrt{2}}+\frac{\left(a|1\rangle +b|0\rangle \right)|10\rangle +\left(a|1\rangle -b|0\rangle \right)|11\rangle }{\sqrt{2}}. \end{array}$$
Alice measures her two qubits and sends the result to Bob. We see from expression (Equation 101) that if the least significant bit that Bob receives is set, then he needs to apply *Z* to his qubit, and if the most significant is set apply an *X*. Doing so, he retrieves the state $a|0\rangle +b|1\rangle =|\psi \rangle$.

The same protocol (Figure 14) with QSL gives
(102)$$\begin{array}{ccc} \left(0,R\right)\;\left(0,R^{\prime }\right)\;\left(b_{x},b_{p}\right) & \overset{H\times I\times I}{\to } & \left(R,0\right)\;\left(0,R^{\prime }\right)\;\left(b_{x},b_{p}\right)\\ & \overset{\mathrm{CNOT}\times I}{\to } & \left(R,R^{\prime }\right)\;\left(R,R^{\prime }\right)\;\left(b_{x},b_{p}\right)\\ & \overset{I\times \mathrm{CNOT}}{\to } & \left(R,R^{\prime }\right)\;\left(R⊕b_{x},R^{\prime }\right)\;\left(b_{x},b_{p}⊕R^{\prime }\right)\\ & \overset{I\times I\times H}{\to } & \left(R,R^{\prime }\right)\;\left(R⊕b_{x},R^{\prime }\right)\;\left(b_{p}⊕R^{\prime },b_{x}\right). \end{array}$$
By the two measurements Alice now retrieves the values of $R⊕b_{x}$ and $R^{\prime }⊕b_{p}$, and sends them to Bob. If the first bit is set he performs an X, i.e., adds $R⊕b_{x}$ to the computational bit modulo 2. If the latter is set he performs a Z, i.e., adds $R^{\prime }⊕b_{p}$ to the phase bit, also modulo 2. The state he ends up with is
(103)$$\left(R⊕\left(R⊕b_{x}\right),R^{\prime }⊕\left(R^{\prime }⊕b_{p}\right)\right)=\left(b_{x},b_{p}\right),$$
which is the state Alice was provided with. Thus, Alice and Bob cooperate to “teleport” the state of Alice’s input system to Bob. Just as in the quantum protocol, Alice does not at any point retrieve any information on the state provided to her. That is, Alice and Bob cooperate to update the state of Bob’s system to correspond to the state of Alice’s input system, independent of what the state of that input system is, just as in the quantum protocol.

In classical communication this is not described as teleportation, but rather as encryption and decryption through the One-Time-Pad, where the random numbers *R* and $R^{\prime }$ act as secret shared key between Alice and Bob. Therefore, in QSL (or Spekkens’ toy model), the teleportation protocol is equivalent to (two uses of) the One-Time-Pad.

We should note that quantum teleportation goes beyond this, since we are restricted to simulate a finite subset of quantum states being teleported, rather than the continuum of states in quantum theory.

#### 3.3.2. Transformations

Here we introduce a transformation unavailable in Spekkens’ model, and as we will see it take us out to a less restricted model. The QSL–Toffoli construction is shown in Figure 15A, and produces the mapping
(104)$$\begin{array}{c} \left(x_{2},p_{2}\right)\\ \left(x_{1},p_{1}\right)\\ \left(x_{0},p_{0}\right) \end{array}\;\to \;\begin{array}{c} \left(x_{2},p_{2}⊕p_{0}x_{1}\right)\\ \left(x_{1},p_{1}⊕p_{0}x_{2}\right)\\ \left(x_{0}⊕x_{2}x_{1},p_{0}\right). \end{array}$$
It is constructed in this way to uphold quantum gate identities such as those in Figure 16A. and Figure 16B. If the input of one of the two control qubits is initiated in $|1\rangle$, the effect is that of a *CNOT* over two other systems. Also, if the target qubit is initiated in $|-\rangle$, the effect is that of a CZ over the two control qubits.

There is another identity using two *CNOT*s to produce a controlled-SWAP, called Fredkin gate (see Figure 16C). The QSL analogue of this is shown in Figure 15B. and produces the mapping
(105)$$\begin{array}{c} \left(x_{2},p_{2}\right)\\ \left(x_{1},p_{1}\right)\\ \left(x_{0},p_{0}\right) \end{array}\to \begin{array}{c} \left(x_{2},p_{2}⊕\left(x_{1}⊕x_{0}\right)\left(p_{1}⊕p_{0}\right)\right)\\ \left(x_{1}⊕x_{2}\left(x_{1}⊕x_{0}\right),p_{1}⊕x_{2}\left(p_{1}⊕p_{0}\right)\right)\\ \left(x_{0}⊕x_{2}\left(x_{1}⊕x_{0}\right),p_{0}⊕x_{2}\left(p_{1}⊕p_{0}\right)\right). \end{array}$$

### 3.4. Properties and Relations to Other Theories

Simulated quantum phenomena in QSL have an efficient simulation on a classical probabilistic Turing machine. It uses two classical bits for each elementary system, and all QSL gates are constructed from a constant number of classical reversible gates. We therefore have the following simple lemma.
**Lemma** **1.**Any quantum circuits, constructed from gates also present in QSL, have a classical simulation that requires at most a constant overhead in resources. It can in turn be simulated in polynomial time, in the size of the circuit, on a classical probabilistic Turing machine.

#### 3.4.1. The Relation to Stabilizer Quantum Mechanics, Locality and Contextuality

Another theory that is computationally tractable is the *stabilizer sub-theory*. As the name suggests it is a sub-theory of quantum theory where one is restricted to transformations from the Clifford group (generated by the Hadamard, phase-gate, and CNOT), and Pauli group measurements. Being restricted to these operations, one can only reach the stabilizer states. Classical controls are also allowed. This sub-theory has an efficient classical simulation, and this is shown by the Gottesman–Knill theorem [12]. An in-depth discussion of its resource requirements can be found in Aaronson and Gottesman [46].

It turns out that Spekkens’ model is closely related to the stabilizer sub-theory [47]. Pusey [37] showed that the number of states in Spekkens’ model is the same as the stabilizer states, and it has operations that are similar to the generators of the Clifford group. However, it is not a restricted version of quantum theory—as we saw in Section 3.1 there are transformations that correspond to antiunitary transformations.

QSL contains Spekkens’ model and is therefore not a restricted version of quantum theory. It is per construction completely local, since all information propagates through local interactions, while the stabilizer sub-theory is not. At this point, it is important to note that both QSL and Spekkens’ model are both also non-contextual, in contrast to stabilizer QM. Consider an observable *A* that is jointly measurable with observables *B* and *C*, where *B* and *C* might not be jointly measurable. Then, we can measure *A* together with *B*, or *A* together with *C*. In quantum theory, *A* would commute with both *B* and *C*. Even so, any attempt to assign values to the outcomes from measuring *A*, *B*, and *C* can force the value of *A* to depend on the *context* of it being measured together with *B* or with *C*, resulting in a contextual model [4].

**Definition** **1.**
*A non-contextual model is a model where measurement outcomes do not depend on the context of the measurement.*


Using the above definition, we arrive at the following theorem.

**Theorem** **1.**
*QSL is a non-contextual model.*


**Proof.** Per construction, QSL has a simultaneous value assignment to all observable quantities, and these values do not change dependent on the measurement or measurement context that we choose to use to retrieve them. Thus, measurements outcomes in QSL does not depend on the context of the measurement. □

#### 3.4.2. QSL Extends the State Space of Spekkens’ Model

QSL also allows for a strictly larger set of states than Spekkens’ model and the stabilizer sub-theory. To see why, consider the construction in Figure 17 that produces a simulation of the GHZ-state
(106)$$\frac{|000\rangle +|111\rangle }{\sqrt{2}}.$$
If the least significant system is measured in the computational basis, the value of *x* is retrieved. If the second system is measured in the phase basis, the value of y⊕xz is retrieved. This information is enough to completely specify the state of the most significant system, i.e., we know the state of both the computational and phase bit, resulting in zero uncertainty.

We can repeat this scheme retrieving the basic bit-values of a fourth system while reusing the two least significant systems as auxiliaries. By induction, with *n* repetitions we can learn the basic bit-values of *n* QSL systems, using only two auxiliary systems. Using the available reversible transformations, or permutations, we can create any value of the basic bit-values we desire. The output of this procedure is not a valid (epistemic) state in Spekkens’ model, and this has consequences for the set of pure states in QSL, because pure states are now states where both bit-values are known.

This also enables a larger mixed-state space. It is true that Spekkens’ original model [13] does not allow for general mixtures of states since knowledge is defined in terms of partitions of the phase space corresponding to the support of the probability distributions used, and not the distributions themselves. For example, the only single-system mixed state allowed in Ref. [13] is the completely mixed state, see Equation (Equation 21). However, it is simple to create any classical mixture of the six available pure states, corresponding to the octahedron in Figure 18. For example, to create a classical mixture of two pure states with probability *p* for one of them, all that is needed is several auxiliary systems that is linear in the number of bits of *p*, all in the completely mixed state. Measurement of each system gives a fair coin toss, and simple binary classical comparison with the number *p* gives a binary output which is 1 with probability *p*. This can then be used to choose what state to prepare, resulting in the desired classical mixture of the two states. The addition in QSL of the QSL–Toffoli enables mixtures within the larger tetrahedron, but note that the states outside the octahedron does not correspond to quantum states, at least not in a simple manner.

Compare this to how with stabilizer quantum theory we start with quantum theory and restrict it to a finite set of states, transformations, and measurement. Adding the quantum Toffoli to stabilizer quantum theory we asymptotically recover quantum theory. For one and two elementary systems, QSL and Spekkens’ theory are equivalent, and can be derived by imposing restrictions on a classical statistical theory over the phase space described by $Z_{2}^{2n}$ (see [47]). Adding the QSL–Toffoli to Spekkens’ model we can reach the ontic states and any mixtures of these. Thus, asymptotically we recover the unrestricted classical theory over $Z_{2}^{2n}$.

#### 3.4.3. QSL Is an Example of a Generalized Probability Theory

Common for all these theories, quantum theory, the stabilizer sub-theory, Spekkens’ model, and QSL is that they are *generalized probabilistic theories* (GPTs). That is, if we define a probabilistic theory *D* aiming to describe the effects of measurement outcomes from these theories, it is necessarily non-Kolmogorovian. To see why let us consider X=1 and Z=1 as events in *D*. The observables do not commute in any of the four theories and are not simultaneously measurable. This means that {X=1}∩{Z=1} is not well-defined, and therefore, *D* does not form a σ-algebra as required by Kolmogorov’s third axiom. For another and more comprehensive account of this see Kleinmann [48].

A σ-algebra is a collection of sets closed under the operations of complement, and countable intersection and union [49]. If these sets—connected to events—are measurable, they can be used to form a logic. With the complement, intersection, and union as the negation, conjunction, and disjunction respectively (NOT, AND, and OR), we recover a Boolean algebra [38]; the logic underpinning classical probability theory. However, since measurable quantities in the four theories do not form a σ-algebra—some quantities are not simultaneously measurable—they cannot form a Boolean algebra. Systems that are described by these theories do not obey classical logic.

A hands-on example of non-classical logic, closely related to Spekkens’ model and QSL, is owing to Cohen [50]. He considers a firefly in a box which is either lit up or not. The box can be viewed from the front side, and from one of the adjacent sides. On the front side of the box there are two windows, one to the right and one to the left. Also, to the side of the box there are two windows, one to the right and one to the left. Now, if the firefly is lit up, a single observer (without any depth perception) looking into either the front or side windows will only be able to tell whether the firefly was on the left or right side, or close to or far from the front side of the box. We can think of the box as divided into four partitions and a single observer can only distinguish between two of them at a time. Cohen then writes
“If we believe that [this] is the best possible characterization of the firefly system, then we believe our firefly in a box is a non-classical physical system, because [looking at both front and side windows] cannot be performed simultaneously. If we believe that we can [look into both front and side windows] simultaneously, perhaps by positioning two observers, one at each window, [...] then we believe our system is classical and [that it has a better characterization].”—Cohen [50]

The connections to Spekkens’ model and QSL are clear. The four partitions of the box relate to Spekkens’ four ontic states, and we can relate the two windows to the two classical bits, the computational and phase bit, of a QSL system. Please note that if the firefly does not light up, the observer has no idea as to where it is, and the system is in the maximally mixed state. Cohen shows that the system is classical by finding a refined description, just like Spekkens’ model and QSL is constructed in a way permitting a finer description, but without that finer description, all three operate per definition according to a non-classical logic.

To summarize, Spekkens’ and QSL are not restricted quantum theories, such as the stabilizer sub-theory, they are restricted *classical* theories. They are generalized probabilistic theories and systems behave according to a non-classical logic, because of the restriction that turn them into generalized probabilistic theories.

We will now switch our attention from protocols describing quantum phenomena, to protocols for solving computational problems. First out is the Bernstein-Vazirani problem, which builds on an important primitive in quantum computation known as Fourier sampling.

## 4. The Bernstein-Vazirani Problem

In 1993 Ethan Bernstein and Umesh Vazirani [19,20] devised the first decision problem that showed a significant oracle separation between the quantum and classical (probabilistic) computational models. In other words, they provided an oracle algorithm separating the computational complexity class **BPP** from the matching class for quantum machines—**BQP**. **BPP** contains all decision problems that can be solved in polynomial time with an error probability bounded away from 1/2 on a probabilistic Turing machine, while **BQP** contains all decision problems that can be solved in polynomial time with an error probability bounded away from 1/2 on a quantum Turing machine [20].

### 4.1. Problem Formulation

The problem they solved is called Recursive Fourier Sampling, but here we will only consider the base problem that we call the Bernstein-Vazirani problem, where there is less than the above-mentioned advantage, see below. Consider that we are given access to an oracle computing a Boolean function, $f:{\left\{0,1\right\}}^{n}\to \left\{0,1\right\}$, promised to be of the particular linear form f(x)=s·x, where $x,s\in {\left\{0,1\right\}}^{n}$. The task is now to find the “secret” string *s*.

### 4.2. Classical Algorithm

In the case where the oracle only gives us access to the function values, we query the function for all inputs *x* with a Hamming weight of 1, i.e., all inputs
(107)$${\tilde{x}}_{i}=\left\{\begin{array}{cc} x_{j}=1, & \mathrm{for}\;j=i\\ 0, & \mathrm{otherwise}, \end{array}\right.$$
where j∈{1,2,…,n} is the bit-index. Then the function will answer with one new bit of information about *s* for each query, $f\left(\tilde{x_{i}}\right)=s_{i}$. After *n* queries we have retrieved *s*.

### 4.3. Quantum Algorithm

If instead the oracle is given as a unitary transformation, adding the result of the query to the query-qubit in the computational basis, specifically $|x\rangle |y\rangle \to |x\rangle |y⊕f(x)\rangle$, then a quantum algorithm can solve the problem with a single query (see Figure 19 and Algorithm 1).

**Algorithm 1:** Bernstein and Vazirani [19,20]  Proceed with the following steps. Prepare an *n*-qubit query register and an additional answer qubit in the state $|0^{n}\rangle |1\rangle$.Apply the Walsh-Hadamard transform to the query register and the output qubit.Apply the oracle.Apply the Walsh-Hadamard transform to the query register.Measure the query register in the computational basis.  The measurement in the last step will reveal the secret string *s*.

From step 1 and 2 we have
(108)$$|0^{n}\rangle |1\rangle \overset{H^{⊗n}⊗H}{\to }\frac{1}{\sqrt{2^{n}}}\sum_{x}|x\rangle |-\rangle .$$
From here on, if nothing else is stated, summation indexes run over the variable’s whole domain. Applying the oracle (step 3) we get
(109)$$\begin{array}{c} \;\frac{1}{\sqrt{2^{n+1}}}\sum_{x}|x\rangle \left(|0⊕f(x)\rangle -|1⊕f(x)\rangle \right)=\frac{1}{\sqrt{2^{n+1}}}\sum_{x}{\left(-1\right)}^{f(x)}|x\rangle \left(|0\rangle -|1\rangle \right)\;, \end{array}$$
and in step 4 the inverse Hadamard transform on the query register gives
(110)$$\begin{array}{cc} \frac{1}{2^{n}\sqrt{2}}\sum_{x,z}{\left(-1\right)}^{f(x)-x\cdot z}|z\rangle \left(|0\rangle -|1\rangle \right)= & \frac{1}{2^{n}\sqrt{2}}\sum_{x,z}{\left(-1\right)}^{(s-z)\cdot x}|z\rangle \left(|0\rangle -|1\rangle \right)=\frac{1}{\sqrt{2}}|s\rangle \left(|0\rangle -|1\rangle \right), \end{array}$$
where the last identity is given by evaluating the sum over *x* to obtain $2^{n}$ if z=s and zero otherwise. By measuring the query register (step 5) we retrieve *s* by calling the quantum oracle only once. This is compared to the linear number of times in the case when we only have access to the function output.

### 4.4. QSL Simulation

To simulate this with QSL the first thing we need is to define the oracle. We know that it is promised to perform f(x)=x·s over the computational basis, and such oracle can be constructed by only using CNOT gates. With their targets at the answer qubit, and the controls on the qubits in the query register that corresponds to where the bit-values $s_{i}$ in the secret string are set. This will produce the desired function. Observe that this allows us to construct the whole algorithm using only Clifford group operations, and it is well-known that there is an efficient classical simulation via the Gottesman–Knill theorem. However, the Stabilizer sub-theory is a non-local contextual model, while QSL is local and non-contextual. Therefore, if this construction works in QSL we can rule out these from being required properties.

First, prepare *n* QSL-bits for the query register all in the (0,R) state, and one QSL-bit for the answer register initiated to (1,R)
(111)$$\left(0^{n},R\right)\;\left(1,R^{\prime }\right)$$
where *R* is a uniformly distributed random bit string, and $R^{\prime }$ is a uniformly distributed random bit. Applying the QSL gate simulating the Hadamard to all QSL-bits gives
(112)$$\left(R,0^{n}\right)\;\left(R^{\prime },1\right)\;.$$

In general, simulating the above construction of the oracle will result in the following map
(113)$$\left(x,p\right)\;\left(a,b\right)\to \left(x,p⊕bs\right)\;\left(a⊕f(x),b\right).$$

For the state in expression (Equation 112) the oracle will add f(R) (modulo 2) to the computational bit of the answer register. Since the phase bit *b* of the answer register is set, the oracle will also have the effect of flipping the phase bits of every QSL system that acts as a control in the query register. Since the phase bits in the query register are all zero, flipping each phase bit for which $s_{i}=1$. This will induce *s* into the phase, giving the state
(114)$$\left(R,s\right)\;\left(R^{\prime }⊕f\left(R\right),1\right).$$
The Walsh-Hadamard transform again swaps all computational and phase bits of the query register
(115)$$\left(s,R\right)\;\left(R^{\prime }⊕f\left(R\right),1\right).$$
Now, measuring the computational bits of the query register will reveal the secret string *s*, at the cost of only one oracle call. Figure 20 shows an example for when the secret string is s=(1011).

This is a special case of the more general procedure known as Fourier Sampling, where all Boolean functions are allowed, i.e., we are not restricted to functions of the form f(x)=x·s. In Fourier Sampling the outcomes of a computational basis measurement is weighted by the absolute square of the Fourier (Walsh-Hadamard) coefficients. In the quantum algorithm, the Fourier-basis contains the Fourier transform of the map, giving direct access to sampling from that distribution. For a classical reversible oracle that only gives access to the function values, the values of the Fourier transform are not directly accessible. In other words, having access to the function encoded in a unitary operator enables us to sample from some distributions related to the function, not only the function itself. To sample from these distributions only having access to the function output might be hard. A discussion about this in relation to the Bernstein-Vazirani problem can be found in [51]. The Deutsch-Jozsa problem in the next section is also a special case of Fourier Sampling.

## 5. The Deutsch-Jozsa Problem

The Deutsch-Jozsa algorithm [10] was the first oracle algorithm that suggested there could be a substantial advantage of doing information processing on quantum systems. It is a generalization of the Deutsch algorithm (see Section 3.2.6), and have been used in many experimental demonstrations of quantum computing. For a detailed account see [52] and citations therein.

### 5.1. Problem Formulation

We are going to use two different problem formulations, the first is due to Cleve et al. [32].

**Definition** **2.**
*Consider that you are given access to an oracle encoding a Boolean function, guaranteed to be either constant or balanced. The problem is to determine whether the function is constant or balanced.*


A *constant* Boolean function is one that always returns 1, or always returns 0. While a *balanced* function returns an equal number of 1s and 0s, i.e., the string of values for all $2^{n}$ possible input values will have a Hamming weight of $2^{n-1}$. To distinguish this string of all output states $\left(f\left(2^{n}-1\right)\ldots f\left(1\right)f\left(0\right)\right)$, that completely characterize the function, from the bit-string that we usually call output, we will call it a *function string*.

The second problem formulation we are going to use is the original formulation by Deutsch and Jozsa [10].

**Definition** **3.**
*Consider that you are given access to an oracle implementing a Boolean function. The problem is to determine whether*
*(i)* 
*the function is **not** constant, or*
*(ii)* 
*the function is **not** balanced.*



Here the function is not guaranteed to be of one or the other kind, it can be any Boolean function, and one of these statements can always be found true. This is therefore a decision problem in contrast to the promise problem in Definition 2.

For completeness, we will go through the algorithms for solving this problem having access to an oracle implementing the function over bits, qubits, and then QSL-bits. We will also consider both definitions of the problem.

### 5.2. Deterministic and Probabilistic Algorithms

If we get access to an oracle that computes the function $f:{\left\{0,1\right\}}^{n}\to \left\{0,1\right\}$, then we can query the function $2^{n-1}+1$ times and decide both problems. If all these outputs are 0 (or 1) then it cannot be balanced, and we have solved the problem described in Definition 3. The problem as described in Definition 2 is also solved by the same algorithm, the only difference is the promise.

This algorithm solves the problem (according to both definitions) with certainty, but with several queries exponential in *n*. This is viewed as evidence that, relative to the oracle, these problems are not in P (the class of problems solvable in polynomial time).

However, if we only require the algorithm to return the solution with an error probability bounded away from 1/2, they can be solved with only a few (constant number of) queries. An algorithm that fails with at most probability 1/4 is to query the function three times, and answer constant (or not balanced in the decision problem) if all three outputs are equal, otherwise balanced (or not constant in the decision problem). The error probability can be calculated as follows.

Since balanced functions have an equal amount of 1s and 0s in the function string, choosing inputs at random we will see a 0 or a 1 at the output with equal probability. Therefore, if the function is balanced, and we query the function three times, the output (000) or (111) both occur with probability 1/8. Thus, there is then a 1/4 probability of wrongfully guess the function to be constant (not balanced), and a corresponding success probability of 3/4, independent of the problem size. If the function is constant (not balanced), the algorithm will succeed with unit probability. For the explicit analysis see [10]. This also shows that relative to the same oracle, this problem is in **BPP**, according to both definitions.

### 5.3. Quantum Algorithm

Here we instead are given access to a quantum oracle, namely one that implements the function as a unitary over the computational basis. Specifically,
(116)$$U_{f}|x\rangle |y\rangle =|x\rangle |y⊕f(x)\rangle .$$
The algorithm can be found as Algorithm 2 below (see also Figure 21).    

**Algorithm 2:** Deutsch and Jozsa [10]  Proceed with the following steps. Prepare an *n*-qubit query register in the state $|0\rangle$, and an output qubit in $|1\rangle$.Apply the Walsh-Hadamard transform to the query register and the output qubit.Apply the oracle.Apply the Walsh-Hadamard transform to the query register.Test if the output state of the query register is $|0\rangle$ by a measurement of the observable $\left|0\rangle \langle 0\right|⊗I_{2}$.  If the test is positive, output “constant,” otherwise “balanced”.

In the algorithm, step 1 and 2 result in
(117)$$|0^{n}\rangle |1\rangle \overset{H^{⊗n}⊗H}{\to }\frac{1}{\sqrt{2^{n}}}\sum_{x}|x\rangle |-\rangle .$$
The oracle (step 3) transforms this into
(118)$$\begin{array}{c} \frac{1}{\sqrt{2^{n+1}}}\sum_{x}|x\rangle (|f(x)\rangle -|\bar{f(x)}\rangle )=\frac{1}{\sqrt{2^{n}}}\sum_{x}{\left(-1\right)}^{f(x)}|x\rangle |-\rangle . \end{array}$$
After the final Walsh-Hadamard transform of the query register (step 4) this becomes
(119)$$\frac{1}{2^{n}}\sum_{x,z}{\left(-1\right)}^{f(x)+x\cdot z}|z\rangle |-\rangle .$$
The query register part of this state is measured to find the value of the Hermitian projector $\left|0\rangle \langle 0\right|$, which can be seen as a test of whether the system is in the state $|0\rangle$, in other words, it tests whether “the state of the query register is unchanged after applying the circuit”. This test will be positive with probability
(120)$${\left|\frac{1}{2^{n}}\sum_{x}{\left(-1\right)}^{f(x)}\right|}^{2}=\left\{\begin{array}{c} 1,\;\mathrm{if}\;f(x)\;\mathrm{is}\;\mathrm{constant}\\ 0,\;\mathrm{if}\;f(x)\;\mathrm{is}\;\mathrm{balanced}, \end{array}\right.$$
and that allows us to solve the problem with only one query, showing that relative to the oracle this problem according to both definitions is in the complexity class **EQP** (problems exactly solvable on a quantum machine in Polynomial time) [19].

### 5.4. The Problem for Small Input

It is well-known that for a small number of input qubits the quantum algorithm admits an efficient classical simulation. Collins et al. [11] pointed out that using the Deutsch-Jozsa algorithm for a meaningful test of quantum computation requires the number of input qubits to be strictly larger than two. Similar findings have been made in [53,54].

The observation of Collins et al. [11] was that the algorithm was completely independent of the answer register, which therefore can be omitted. The important part of the construction is that it produces the correct phase imprint in the query register. This is sometimes referred to as a phase oracle. Please observe that this kind of oracle does not allow for retrieving function values, and cannot be used to employ the regular solution, but only makes the quantum algorithm available. Furthermore, having access to a unitary implementing a Boolean function, a phase oracle can efficiently be constructed [20].

We can summarize these results simply by the fact that all balanced and constant functions over one and two bits have oracles that only use Clifford group operations (see Figure 22) and therefore admit a classical simulation in ether QSL [14] or the stabilizer sub-theory.

For three qubits of input, and not constructing phase oracles, all 72 balanced and constant function can be implemented in a reversible circuit using only one Toffoli gate and five *CNOT*s (for the explicit implementations see Appendix A). Mapping these circuits into the corresponding QSL-circuits, we obtain an implementation of an oracle for each of the 72 functions, and using these to simulate the Deutsch-Jozsa algorithm, QSL solves the problem with unit probability.

In general, there are two constant functions and
(121)2n2n-1≥2n2n-12n-1=22n-1
balanced functions, because of the simple bound
(122)nk≥nkk.
This makes it intractable to construct, or even to index, all oracles explicitly. Thus, the oracle paradigm is unavoidable in this problem setting. This will be discussed further in Section 6.

### 5.5. QSL Simulation Guaranteed a Constant or Balanced Function

To approach the problem with QSL we need to determine the effect of an oracle from QSL-bits onto QSL-bits. We will start by specifying a valid implementation of a quantum oracle, and then simply map all the quantum gates of the implementation into QSL gates. This will allow us to obtain an expression for the effect of the QSL oracle, and show that there exists an oracle corresponding to a simulation of a valid quantum oracle. We then go on to show that if we are given access to this oracle that is sufficient to solve the problem with a single query.

The implementation of the quantum circuit that we are going to use is shown in Figure 23, and employs two Toffoli gates and two boxes representing a permutation $\pi \in S_{2^{n}}$ of the computational basis states. These can be constructed from NOT, *CNOT*, and Toffoli gates, and at most one auxiliary bit [55].

There is one parameter $|b\rangle$ that determines what type of function the circuit implements. If $|b_{0}\rangle =|1\rangle$ the function is balanced and if $|b_{0}\rangle =|0\rangle$ it is constant. If the function is constant, the value is $f\left(x\right)=b_{1}$. The idea is to use a simple construction for one particular balanced function, and then use a permutation of the possible values *x* to make all balanced functions available. If $|b_{0}\rangle =|1\rangle$ and the permutation is the identity, the Toffoli connected to $|b_{0}\rangle$ will give the balanced function $f\left(x\right)=x_{n-1}$ If the permutation is any other than the identity permutation, the input values are permuted before the function is calculated and written to the answer register. This makes all balanced functions available in the used reversible circuitry. The inverse permutation is used on the query register to uncompute the permutation, assuring that we get the map $|x\rangle |y\rangle \to |x\rangle |y⊕f(x)\rangle$.

We start by analyzing the effect of the permutation box mapped into QSL. Its effect can be separated into two parts, one is the effect on the computational bits and one on the phase bits. The effect on the computational bit-string is the same as the effect of the quantum gate $\pi_{f}$ on the computational basis. The effect over the phase bit-string will be another permutation, and since any computational basis permutation can be built from *X*, *CNOT*, and Toffoli gates, the corresponding QSL gates can be used to build a QSL permutation In general, if the construction uses Toffoli gates, the exact permutation on the phase bit-string will depend not only on which computational basis-string permutation is used, but also on the actual value of the computational bits. Let us call this permutation $\pi_{f,x}$. As we shall see, it does not matter precisely which permutation is realized, only that the resulting map is invertible. The corresponding map to a computational basis permutation $\pi_{f}$ in QSL will be
(123)$$\left(x,p\right)\to \left(\pi_{f}\left(x\right),\pi_{f,x}\left(p\right)\right),$$
and the effect of the inverse on the latter state, after a possible change of the phase bits, will be
(124)$$\left(\pi_{f}\left(x\right),p^{\prime }\right)\to (x,\pi_{f,x}^{-1}\left(p^{\prime }\right)).$$
Then, the overall effect of the circuit in Figure 23 is
(125)$$\begin{array}{ccc} \left(x,p\right)\;\left(y,r\right) & \overset{\pi_{f}}{\to } & \left(\pi_{f}\left(x\right),\pi_{f,x}\left(p\right)\right)\;\left(y,r\right)\\ & \overset{T_{a}}{\to } & \left(\pi_{f}\left(x\right),\pi_{f,x}\left(p\right)+b_{0}r\delta_{n-1}\right)\;\left(y⊕b_{0}\pi_{f}{\left(x\right)}_{n-1},r\right)\\ & \overset{T_{b}}{\to } & \left(\pi_{f}\left(x\right),\pi_{f,x}\left(p\right)+b_{0}r\delta_{n-1}\right)\;\left(y⊕b_{0}\pi_{f}{\left(x\right)}_{n-1}⊕b_{1},r\right)\\ & = & \left(\pi_{f}\left(x\right),\pi_{f,x}\left(p\right)+b_{0}r\delta_{n-1}\right)\;\left(y⊕f(x),r\right)\\ & \overset{\pi_{f}^{-1}}{\to } & \left(x,\pi_{f,x}^{-1}\left(\pi_{f,x}\left(p\right)+b_{0}r\delta_{n-1}\right)\right)\;\left(y⊕f(x),r\right), \end{array}$$
where $\pi_{f}{\left(x\right)}_{n-1}$ denotes the most significant bit of the input vector after it has gone through the permutation, and $\delta_{n-1}$ is the bit-vector for which the most significant bit is 1 and all others 0.

If we now get oracle access to this function we can use it to query for particular function values, by preparing the registers in the states corresponding to $|x\rangle |0\rangle$ and applying the oracle. We can also use it to determine whether the function is constant or balanced by running the Deutsch-Jozsa algorithm. This gives
(126)$$\begin{array}{ccc} \left(0,X\right)\;\left(1,Y\right) & \overset{H^{\times (n+1)}}{\to } & \left(X,0)\;(Y,1\right)\\ & \overset{U_{f}}{\to } & \left(X,\pi_{f,X}^{-1}(\pi_{f,X}\left(0\right)+b_{0}\delta_{n-1})\right)\;\left(Y⊕f\left(X\right),1\right)\\ & \overset{H^{\times n}}{\to } & \left(\pi_{f,X}^{-1}\left(\pi_{f,X}\left(0\right)+b_{0}\delta_{n-1}\right),X\right)\;\left(Y⊕f\left(X\right),1\right). \end{array}$$
If the function is constant, then $b_{0}=0$ and measurement of the query register will return $\pi_{f,X}^{-1}\left(\pi_{f,X}\left(0\right)\right)=0$. If the function is balanced, then the measurement returns $\pi_{f,X}^{-1}\left(\pi_{f,X}\left(0\right)+\delta_{n-1}\right)\neq 0$, which is part from zero since $\pi_{f,X}^{-1}$ is injective. Please note that the detailed behavior of the permutation $\pi_{f,X}$ is not important since knowing that it is invertible is enough. We have proven the following theorem.

**Theorem** **2.**
*There is an efficient QSL algorithm that solves the*
Deutsch-Jozsa
*problem by a single query to the oracle. This algorithm has an efficient classical simulation on a PTM.*


That the simulation is efficient follows from the fact that the QSL circuit uses several gates polynomial in the input size, relative to the oracle, and from Lemma 1. The QSL algorithm gives the same answers as the quantum algorithm: the zero bit-string if the function is constant, and a non-zero bit-string if the function is balanced. It is thus a faithful simulation of the quantum algorithm, in other words, these procedures are operationally the same.

### 5.6. QSL Simulation Accepting Arbitrary Boolean Functions

To approach the less strict problem formulation (Definition 3) we need a construction that can produce all Boolean functions. To do that we have chosen an implementation using a comparator. A comparator is a device that compares two values, and we have chosen to compare x+a and $2^{n}$, and output 1 if $x+a\geq 2^{n}$ and 0 otherwise. This is an example of a function f(x) that output 1 for *a* of the possible inputs, and 0 otherwise, for each $a\in \{0,\ldots ,2^{n}-1\}$. A generic such function can be generated by using the computational basis permutation presented in Section 5.5.

Such a comparator can be built in reversible logic by adapting a construction for a ripple-carry adder, known as a Cuccaro adder [56,57]. The Cuccaro adder uses four reversible logic registers $\left(c_{\mathrm{in}},x,a,z\right)$, where *x* and *a* are input-strings of equal length, and $c_{\mathrm{in}}$, and *z* are single-bit registers for carry-in and carry-out respectively. The mapping is
(127)$$\left(c_{\mathrm{in}},x,a,z\right)\to \left(c_{\mathrm{in}},x,x+a,z⊕c_{\mathrm{out}}\right),$$
but since we want the above comparator, we note that $c_{\mathrm{out}}$ is set if and only if $x+a\geq 2^{n}$ (generating a carry). By using the initial value z=0 and uncomputing the ripple-carry-adder intermediate values instead of completing the subtraction (see [57]), we obtain the map
(128)$$\left(0,x,a,0\right)\to \left(0,x,a,x+a\geq 2^{n}\right).$$
An illustrative example of a comparator built using majority-function building blocks (see Figure 24) to compare two 4-bit integers is shown in Figure 25.

To enable the function that is constant 1 one possible solution is to extend the comparator so that it works for $0\leq a\leq 2^{n}$. Inverting the output if the additional bit $a_{n}$ equals 1 gives the constant function 0 for the value a=0 and the constant function 1 for the value $a=2^{n}$. The complete construction of the circuit is shown in Figure 26.

To verify that this gives the correct answer in the Deutsch-Jozsa algorithm, we need to check the phase kickback of the gate array. For a given bit-index, the MAJ and ${\mathrm{MAJ}}^{-1}$ gates reverse the phase transformation as well as the computational basis transformation. The one difference is the possible phase kickback $⊕k_{i}$ on the $a_{i}$ register, between the two gates, from the i+1st step of the chain or from the oracle target register if i+1=n. The effect of the three gates in the final ${\mathrm{MAJ}}^{-1}$ gate is
(129)$$\begin{array}{cc} \left(c_{i}⊕a_{i},\cdot \right)\;\left(x_{i}⊕a_{i},\cdot \right)\;\left(c_{i+1},\cdot ⊕k_{i}\right) & \to \left(\cdot ,\cdot ⊕k_{i}\left(x_{i}⊕a_{i}\right)\right)\;\left(\cdot ,\cdot ⊕k_{i}\left(c_{i}⊕a_{i}\right)\right)\;\left(\cdot ,\cdot ⊕k_{i}\right)\\ & \to \left(\cdot ,\cdot ⊕k_{i}\left(x_{i}⊕a_{i}\right)\right)\;\left(\cdot ,\cdot ⊕k_{i}\left(c_{i}⊕a_{i}\right)\right)\;\\ & \;\times (\cdot ,\cdot ⊕k_{i}⊕k_{i}\left(x_{i}⊕a_{i}\right))\\ & \to \left(\cdot ,\cdot ⊕k_{i}\left(x_{i}⊕a_{i}\right)\right)\;\left(\cdot ,\cdot ⊕k_{i}\left(c_{i}⊕a_{i}\right)\right)\;\\ & \;\times (\cdot ,\cdot ⊕k_{i}⊕k_{i}\left(x_{i}⊕c_{i}\right)) \end{array}$$
There are two cases we need to check. If *f* is constant then a=0 or $a=2^{n}$, and if *f* is balanced then $a=2^{n-1}$. In both cases $c_{i}=a_{i}=0$ for 0≤i≤n-2, making $k_{i}\left(c_{i}⊕a_{i}\right)=0$, so the phase kickback is 0 for all the query register bits except the most significant bit. For the most significant bit, it is still the case that $c_{n-1}=0$ so that the phase kickback is $k_{n-1}\left(c_{n-1}⊕a_{n-1}\right)=k_{n-1}a_{n-1}$. The Deutsch-Jozsa algorithm now gives
(130)$$\begin{array}{ccc} \left(0,X\right)\;\left(1,Y\right) & \overset{H^{\times (n+1)}}{\to } & \left(X,0)\;(Y,1\right)\\ & \overset{U_{f}}{\to } & \left(X,\pi_{f,X}^{-1}(\pi_{f,X}\left(0\right)+a_{n-1}\delta_{n-1})\right)\;\left(Y⊕f\left(X\right),1\right)\\ & \overset{H^{\times n}}{\to } & \left(\pi_{f,X}^{-1}\left(\pi_{f,X}\left(0\right)+a_{n-1}\delta_{n-1}\right),X\right)\;\left(Y⊕f\left(X\right),1\right). \end{array}$$
This is exactly the same behavior as in the previous section with $a=2^{n-1}b$. In fact, $a=2^{n-1}b$ makes the present oracle perform the exact same map as the one in the previous section, both in the computational and phase basis. The same analysis as in Section 5.5 now follows, and the measurement will reveal that the function was *not balanced* ($a_{n-1}=0$ gives zero output) or *not constant* ($a_{n-1}=1$ gives non-zero output). We have the following theorem.

**Theorem** **3.**
*There is an efficient QSL algorithm that solves the **original***
Deutsch-Jozsa
*problem by a single query to the oracle. This algorithm has an efficient simulation on a classical probabilistic Turing machine.*


That the simulation is efficient relative to the oracle follows from Lemma 1.

### 5.7. Query Complexity

The standard analysis of the Deutsch-Jozsa problem states that if we only have access to the function $f:{\left\{0,1\right\}}^{n}\to \left\{0,1\right\}$, then it cannot be solved with unit probability using less than an exponential number of queries. This is usually taken as evidence that the problem is not in **P**. More carefully put, this is evidence that relative to this particular oracle, the problem is not in **P**. If we accept a solution with a bounded error probability then, relative to the same oracle, the problem is in **BPP** [10].

Quantum computation allows us to solve the problem using only O(1) queries, so that relative to a quantum oracle the problem is in **BQP**. However, the quantum oracle is different, since the function is encoded in a much richer framework—as a transformation of the computational basis of a quantum system composed of qubits. Using this richer framework, we can choose to extract additional information not available in the regular query model, and it is this information that allows us to solve the problem using only O(1) queries. Relative to this *richer* oracle, the problem is in **BQP**.

Please note that relative to an oracle in QSL, the Deutsch-Jozsa problem can be solved with only O(1) queries. In addition, both the oracle and the Deutsch-Jozsa algorithm in QSL can be efficiently simulated on a classical probabilistic Turing machine, so that relative to an oracle in QSL, the Deutsch-Jozsa problem is in **BPP**. In this particular case, there is no quantum advantage in terms of query complexity.

Furthermore, the ideal quantum algorithm solves the Deutsch-Jozsa problem with unit probability, so a more precise characterization is to put the problem into **EQP**. The QSL algorithm also provides the solution with unit probability, and in fact it provides it deterministically. It is even the case that the randomness present in the framework never contributes to the computation in the Deutsch-Jozsa algorithm, i.e., the random values can be replaced with a fixed value without interfering with the result. Combining this with Lemma 1, we see that relative to the QSL-oracles, the problem is in **P**.

## 6. Oracles as a Comparison

The fact that the quantum query model possesses *much richer* quantum oracles than standard classical binary-input binary-output function oracles, puts doubt in the standard comparison between the two. The richness of the quantum oracle really calls for a comparison with a correspondingly rich classical framework in which oracles are able to encode the function in one subpart of the system, akin to the computational basis, and encode some additional function in another subpart of the system, akin to the phase basis. Any comparison between classical and quantum query complexity should take place between these two richer frameworks: one quantum, and one classical that allows these richer oracles. QSL has exactly the required properties, and can in turn be efficiently simulated on a classical probabilistic Turing machine. There are several points to make here.

### 6.1. The Additional Structure and Constraints

Only having access to an oracle that computes a function from bits to bits is insufficient for quantum computation. So, what are the conditions?

One condition is that the oracle needs to accept qubits as inputs. This is sound but not sufficient, since there are many quantum operations that produce the correct function map, but will not enable quantum computation. One example is an implementation of the function over a completely phase-mixing channel.

A more precise condition is that the function needs to be implemented as a unitary operator, but even this is not enough. The function needs to be implemented as a reversible function with all auxiliary bits cleared and the query register restored. Even more important, the unitary operator also needs to preserve the relative phases $c_{x,y}$
(131)$$U_{f}\left(\sum_{x,y}c_{x,y}|x\rangle |y\rangle \right)=\sum_{x,y}c_{x,y}|x\rangle |y⊕f(x)\rangle$$

There are many unitary implementations encoding the function in the computational basis for which the algorithm does not work. One example is the unitary given by
(132)$$U_{f}^{\prime }\left(\sum_{x,y}c_{x,y}|x\rangle |y\rangle \right)=\sum_{x,y}c_{x,y}{\left(-1\right)}^{f(x)}|x\rangle |y⊕f(x)\rangle$$
for which the Deutsch-Jozsa algorithm will answer “constant”—for all Boolean functions—with unit probability [14]. Another example is the unitary
(133)$$U_{f}^{\prime \prime }\left(\sum_{x,y}c_{x,y}|x\rangle |y\rangle \right)=\sum_{x,y}c_{x,y}{\left(-1\right)}^{x_{0}+f\left(x\right)}|x\rangle |y⊕f(x)\rangle ,$$
for which the Deutsch-Jozsa algorithm will answer “balanced”—for all Boolean functions—with unit probability. There are exponentially many unitaries of this kind. Still, for all these examples, the function map is available, and the oracle can still be used to solve the problem using the classical query algorithm.

A similar observation has been made by Machta [58]. In fact, most unitary implementations of *f* will not work for the algorithm since, in general, we can have an addition of ${\left(-1\right)}^{g(x,y)}$ to the relative phases, where g(x,y) is an arbitrary function not necessarily related to *f*.

Underneath the compact requirement that the function should be implemented as the specific unitary $U_{f}|x\rangle |y\rangle =|x\rangle |y⊕f(x)\rangle$ lurks an exponential number of constraints: the preservation of an exponential number of complex amplitudes. This is the source of the richness of quantum oracles: the phase constraints enable access to much more information than can be accessed with the standard classical binary-input binary-output function oracle. Only under these phase constraints it is possible to retrieve information otherwise not available from the phase degree of freedom, which in turn is useful for a more efficient solution of the problem under study.

### 6.2. Is the Black Box Black?

The above difference between quantum oracles and classical oracles that only gives us access to function values, puts into question whether they can be used in a justified comparison. The broader non-black-box definition offers a good analogy. In this case, a quantum oracle should be described as a gray-box model; we have partial knowledge about the underlying model generating the statistics, rather than only getting access to the statistics.

Remember the example in Section 2 where a gray-box model is used in relation to cryptography. There, in addition to knowing the function map of the protocol, we also know that it has a physical implementation in some electrical circuit, but we do not know everything that there is to know about the box. In this model, which includes a physical description, side channels become available, e.g., as information leakage through the power consumption of the circuit. In cryptography, a protocol is not considered broken if there is a specific implementation that leaks the secret, which might otherwise be hard to compute. Such a comparison would be unjustified.

In quantum query complexity, the oracle also comes with a physical description, specifically it needs to be implemented as one specific unitary, modulo global phase. Only then will the additional information be available. It is not available from where we usually read out the value of the function, i.e., in the computational basis. Instead, it is available as phase information, in another physical degree of freedom, constituting a side-channel that allows us to access the extra information that enables the speed-up. In quantum computing this side effect is systematically used to efficiently solve computational problems that are normally not efficiently solvable. This gives a direct comparison between solving computational problems in quantum computers, and breaking a cryptographic device through a side channel.

### 6.3. Assumptions in the Use of Oracles

Let us now briefly return to the definitions of Section 2.2, and note some differences to the use of the oracle notion in classical algorithms, quantum algorithms, and QSL algorithms. In query complexity, an oracle is defined by what operation it performs, with very few other properties, and then all statements that are made are relative to that oracle. Informally the question is: if the oracle can be computed in a single time-step, what problems could then be solved with a given amount of resources?

In classical query complexity the oracle computes the function and the standard definition is that of a black box, i.e., there are no other properties to declare. Relative to such an oracle the Deutsch-Jozsa problem cannot be solved deterministically in polynomial time. In reversible classical logic the standard oracle definition is that of a black box that adds the result to a target bit (see Section 2.2). Also relative to such an oracle the Deutsch-Jozsa problem cannot be solved deterministically in polynomial time.

In quantum query complexity the oracle is defined as a unitary transformation computing the function. This gives a not-quite black box, because this adds the property of a specific relation between the relative phases within the output states. Given such an oracle the Deutsch-Jozsa problem can be solved with a single query. The justification for the interest in quantum query complexity is that quantum theory predicts that these oracles exist and can hypothetically be built.

Finally, in QSL the oracle is defined as a reversible transformation computing the function, also adding the property of a specific relation between the relative phases in the phase-basis output. Given such an oracle the Deutsch-Jozsa problem can be solved with a single query. The justification for the interest in QSL query complexity is that QSL predicts that these oracles exist, in fact giving an explicit simulation in a classical probabilistic Turing machine.

In classical versus quantum oracle separations the assumption is that classical oracles cannot encode any other information than the function. QSL serves as an example that if we just slightly modify the classical reversible model, then we do get classically simulatable oracles that systematically encode other information.

### 6.4. Systematic Phase Errors

The additional information that we can choose to retrieve from a quantum circuit, and that sometimes can be used to solve computational tasks, is related to how unitary transformations manage relative phases. The information in the amplitude and in the phase is mutually unbiased, and QSL handles this by storing them independently. A function over the computational bits is always accompanied by a corresponding function on the phase bits.

It would be convenient if, for any Boolean function *f* over the computational bits, we could have a unique expression for the effect on the phase bits. Such an expression would be useful for analyzing the general behavior of QSL. Unfortunately, different implementation of the same function will result in different effects on the phase bits.

As an example, there is another procedure for creating the GHZ-state using two *CNOT*-gates rather than one *CNOT* and one Toffoli gate, as in Figure 17. This construction is shown in Figure 27. It is clear that these two constructions produce different maps. They give the same map on the computational bits, but different on the phase bits. It is not only the maps that are different, also the probability distributions that we relate to quantum states are as well. To see this, consider the marginal probability distributions from measuring the least significant system in the computational basis, and the second system in the phase basis. Doing this with the construction in Figure 17, we end up with a state represented by a probability distribution that has 0 bits of entropy, as we have seen above. On the other hand, doing the same for the construction in Figure 27, using two *CNOT*-gates, the output state will have one bit of entropy.

In other words, simulation of different constructions of quantum circuits that produce the same function in the computational basis, does not produce the same effect on the phase bits. The major inconvenience this phenomenon brings is of course that we cannot analyze the general effect of applying a function, but must resort to specific constructions. If we find a construction that lets us solve a problem efficiently, similarly as in quantum theory, there is no guarantee that other constructions will work. Conversely, if we fail to find a construction that works, we cannot say that the quantum algorithm lacks an efficient simulation in QSL. However, this is similar to what we would expect if we had an implementation of the Toffoli gate with systematic errors, and this will be clarified in what follows.

The presence of systematic phase errors in a quantum gate array will influence the side-channel enabled by the preservation of relative phases. Consider the following Toffoli gate that includes a systematic phase error θ,
(134)$$\left[\begin{array}{cccccccc} 1 & 0 & 0 & 0 & 0 & 0 & 0 & 0\\ 0 & 1 & 0 & 0 & 0 & 0 & 0 & 0\\ 0 & 0 & 1 & 0 & 0 & 0 & 0 & 0\\ 0 & 0 & 0 & 1 & 0 & 0 & 0 & 0\\ 0 & 0 & 0 & 0 & 1 & 0 & 0 & 0\\ 0 & 0 & 0 & 0 & 0 & 1 & 0 & 0\\ 0 & 0 & 0 & 0 & 0 & 0 & 0 & e^{i\theta }\\ 0 & 0 & 0 & 0 & 0 & 0 & e^{i\theta } & 0 \end{array}\right]$$
or equivalently
(135)$$\sum_{x=0}^{5}\left|x\rangle \langle x\right|+e^{i\theta }\left(\left|6\rangle \langle 7\right|+\left|7\rangle \langle 6\right|\right).$$
This unitary will be denoted $\tilde{T}$, or controlled-controlled-$\tilde{X}$. Our aim is to analyze four different constructions of the balanced function computing the majority of three bits; the function with function string (11101000). The four constructions are shown in Figure 28.

The construction in Figure 28A is straightforward, using four $\tilde{T}\left(3\right)$ gates with their target in the answer register. If we construct the $\tilde{T}\left(3\right)$ gates as in Figure 29, then the resulting operator will be
(136)$$\tilde{T}\left(3\right)=\sum_{X=0}^{13}\left|x\rangle \langle x\right|+e^{i\theta }\left(\left|14\rangle \langle 15\right|+\left|15\rangle \langle 14\right|\right).$$
The unitary implementation of Figure 28A becomes $|x\rangle |y\rangle \to {\left(e^{i\theta }\right)}^{f(x)}|0\rangle |y⊕f(x)\rangle$, and the Deutsch-Jozsa algorithm will wrongfully answer constant with probability
(137)$${\left|\frac{1}{8}\sum_{x=0}^{8}{\left(e^{i\theta }\right)}^{f(x)}{\left(-1\right)}^{f(x)}\right|}^{2}={\left|\frac{4-4e^{i\theta }}{8}\right|}^{2}=\sin^{2}\left(\frac{\theta }{2}\right).$$
With θ=π/3 the error probability becomes 1/4.

The construction in Figure 28B uses an intermediate step to compute the majority in place, using the MAJ-gate shown in Figure 24, adding the result to the output, and then uncomputing the intermediate result by applying $M\;AJ^{†}$. With this construction, the Deutsch-Jozsa algorithm will (correctly) answer balanced with unit probability even though there is a phase error in the $\tilde{T}$ gate.

This reproduces the behavior of QSL, where using the QSL–Toffoli gives an error probability 1/4 when the oracle is constructed as in Figure 28A, and zero error probability if constructed as in Figure 28B. The calculations can be found in Appendix B.

With the quantum circuit in Figure 28C the Deutsch-Jozsa algorithm will wrongfully answer constant with probability
(138)$${\left|\frac{1}{8}\left(1+1+1-e^{i\theta }+1-e^{i\theta }-e^{i\theta }-e^{i3\theta }\right)\right|}^{2}={\left|\frac{4-3e^{i\theta }-e^{i3\theta }}{8}\right|}^{2},$$
and for Figure 28D the probability becomes
(139)$${\left|\frac{1}{8}\left(1+e^{i\theta }+1-e^{i2\theta }+1-1-e^{i\theta }-1\right)\right|}^{2}={\left|\frac{1-e^{i2\theta }}{8}\right|}^{2},$$
and again, if θ=π/3 these become 19/64 and 1/64 respectively. In QSL, only the three random bits going into the query register is taking part in the simulation. Therefore, any probability distribution describing the outcome can only be resolved into fractions of 8. The best approximations we can obtain for 19/64 and 1/64 is then 1/4 and 0 respectively. This is exactly the error probability that we see in QSL (see Appendix B).

Perhaps it is important to stress that the QSL–Toffoli is not a quantum Toffoli with a systematic error. It is a simulation of a quantum Toffoli that uses classical bits in the simulation. What we have seen here is that the QSL–Toffoli behaves similarly to a quantum Toffoli gate with systematic error, giving similar variations in output statistics for different constructions of the same function.

### 6.5. Starting with Something Else Than Access to an Oracle

It may also be argued that if we start from something else than access to an oracle, for instance a circuit implementation, it might be hard to translate that into QSL. The question is why this would be considered an argument at all. In the problems under study, there are simply too many possible functions to convey an explicit construction to the solver of the problem. For example, in the case of an explicit function from the Deutsch-Jozsa problem, the number of balanced functions grows as $\sqrt{2^{2^{n}}}$, so that simply *indexing* the explicit constructions would require an exponential amount of information. How would it be possible to convey the explicit construction to the solver in this case? If successful, the solver will have received an exponential amount of information in the transfer.

One possibility to make the problem practical is to restrict the problem to a polynomial-sized subset of all balanced functions, but this would be a severe alteration of the problem formulation, and then there is no reason to believe that the proof of separation still applies. The simplification to a polynomial-sized subset may enable a polynomial time solution even in a classical Turing machine, and it is even possible to argue that this is likely the case.

In addition to this, given an explicit construction, for example in the form of a circuit or a procedure, it may well be that the structure of the explicit construction gives the solution away, the most clear example of this in quantum computation is the standard construction of the Bernstein-Vazirani oracle that can be found in Figure 20. In the case that the solution cannot be found by inspection, the extraction of the solution from the structure would constitute a field of research in its own right—and with the interpretation of the extra information as a side-channel, it could even be argued that quantum computation is part of that field of research.

The above complications provide the basic motivation to use the black-box query model in the first place. In the remainder of the paper, we will use the query model to investigate how QSL performs in two additional quantum algorithms: Grover’s algorithm and Simon’s algorithm. We will also have a look at a real-world algorithm, Shor’s algorithm, where construction of the gate array is polynomial time, and study the behavior of QSL in this situation.

## 7. Grover’s Algorithm

Given a Boolean function $f:{\left\{0,1\right\}}^{n}\to \left\{0,1\right\}$, Grover’s algorithm [59] is a probabilistic algorithm that returns an element of the preimage of f(x)=1, i.e., it returns with high probability a satisfying assignment to *f*, if there is one. This algorithm can be used to solve the decision problem of answering whether *f* has a satisfying assignment or not.

If we have no knowledge about the function, and we are only given oracle access to it, we must resort to using exhaustive search—simply querying the function for all inputs until we find a 1 or not. This will in the worst case require $O(2^{n})$ queries. The best guess that a probabilistic algorithm can use, without any knowledge about the function, is to pick an input uniformly at random. This will also require $O(2^{n})$ queries to solve with a bounded error probability. Both of these estimates rely on the black-box nature of the function provided, because if the function gate array is available, it may be trivial to find the satisfying assignment, compare for example Figure 30 with Figure 31.

The quantum algorithm proceeds as specified in Algorithm 3, and only requires $O(\sqrt{2^{n}})$ queries to solve the problem with a bounded error probability.

**Algorithm 3:** Grover [59]  Perform a Walsh-Hadamard transform on the initial state ${|0\rangle }^{⊗n}|1\rangle$, and then repeat the following steps $\sqrt{2^{n}}$ times. Apply the oraclePerform a Walsh-Hadamard transform of the query registerApply $\left(I-2\left|0\rangle \langle 0\right|\right)$ to the query register. This is an *n*-controlled-*Z* with all controls inverted (including the target *Z*).Perform a Walsh-Hadamard transform of the query register  Then measure the query register, which will give you the satisfying assignment with highprobability (see further [30]). One repetition of the steps 1–4 is known as the Grover operator.

### 7.1. Problem Formulation

Here we will restrict the problem to the worst case scenario, where there is only one satisfying assumption.

**Definition** **4.**
*Assume that you are given access to an oracle for the function $f:{\left\{0,1\right\}}^{n}\to \left\{0,1\right\}$ and that it only has one satisfying assignment $x^{*}:f\left(x^{*}\right)=1$. Find $x^{*}$.*


This is the original problem formulation used by Grover. As usual, for the quantum case the quantum oracle is assumed to be encoded as a unitary function from qubits to qubits, and not as a Boolean function from bits to bits.

### 7.2. One-Shot Grover

We will start with a small example which is sometimes called One-shot Grover. This is the case where there are two qubits in the query register (In general, the requirement for a “one-shot” Grover is that 1/4 of the input states are satisfying assignments.) and the Grover operator only need to be applied once (see Figure 30).

Since $U_{f}|x\rangle |y\rangle =|x\rangle |y⊕f(x)\rangle$, it is straightforward to check that
(140)$$U_{f}|x+\rangle =|x+\rangle ,$$
and
(141)$$U_{f}|x-\rangle =\left\{\begin{array}{cc} -|x-\rangle , & \;\mathrm{if}\;x=x^{*}\\ +|x-\rangle , & \;\mathrm{otherwise}. \end{array}\right.$$
This is another example of phase kickback. From linearity we obtain
(142)$$U_{f}=I⊗|+\rangle \langle +|+\left(I-2|x^{*}\rangle \langle x^{*}|\right)⊗|-\rangle \langle -|.$$
The combination of steps 2, 3, and 4 is sometimes called “inversion over the mean”, and can be written
(143)$$A=H^{⊗2}\left(I-2\left|00\rangle \langle 00\right|\right)H^{⊗2}=\left(I-2\left|++\rangle \langle ++\right|\right)$$
The quantum algorithm proceeds as follows
(144)$$\begin{array}{ccc} |00\rangle |1\rangle & \overset{H^{⊗3}}{\to } & |++\rangle |-\rangle \\ & \overset{U_{f}}{\to } & \left(|++\rangle -|x^{*}\rangle \right)|-\rangle \\ & \overset{A⊗I}{\to } & \left(-|++\rangle -\left(|x^{*}\rangle -|++\rangle \right)\right)|-\rangle \\ & = & -|x^{*}\rangle |-\rangle , \end{array}$$
so that a measurement of the query register will reveal $x^{*}$ with unit probability.

To see how this works in QSL we need to find an expression for the oracle. For n=2 there are four functions with a single satisfying assignment
(145)$$\begin{array}{c} f\left(x\right)=\bar{x_{1}}\;\bar{x_{0}},\\ f\left(x\right)=\bar{x_{1}}x_{0},\\ f\left(x\right)=x_{1}\bar{x_{0}},\;\mathrm{and}\\ f\left(x\right)=x_{1}x_{0}. \end{array}$$
The canonical constructions for these are to use the four different Toffoli gates with different combinations of regular and inverted controls. Let us take the example where $f\left(x\right)=\bar{x_{1}}x_{0}$. Here $x^{*}=\left(01\right)$ and the resulting quantum circuit is shown in Figure 31.

The operation A (“inversion over the mean”) would be
(146)$$\begin{array}{cc} \left(x_{1},p_{1}\right)\;\left(x_{2},p_{2}\right) & \overset{H^{\times 2}}{\to }\left(p_{1},x_{1}\right)\;\left(p_{2},x_{2}\right)\\ & \overset{{\mathrm{CZ}}_{00}}{\to }\left(p_{1},x_{1}⊕\bar{p_{2}}\right)\;\left(p_{2},x_{2}⊕\bar{p_{1}}\right)\\ & \overset{H^{\times 2}}{\to }\left(x_{1}⊕\bar{p_{2}},p_{1}\right)\;\left(x_{2}⊕\bar{p_{1}},p_{2}\right). \end{array}$$
The complete algorithm becomes
(147)$$\begin{array}{ccc} \left(0,r_{1}\right)\;\left(0,r_{0}\right)\;\left(1,r_{t}\right) & \overset{H^{\times 3}}{\to } & \left(r_{1},0\right)\;\left(r_{0},0\right)\;\left(r_{t},1\right)\\ & \overset{O_{f}}{\to } & \left(r_{1},r_{0}\right)\;\left(r_{0},\bar{r_{1}}\right)\;\left(r_{t}⊕f\left(r\right),1\right)\\ & \overset{A\times I}{\to } & \left(r_{1}⊕\bar{\bar{r_{1}}},r_{0}\right)\;\left(r_{0}⊕\bar{r_{0}},\bar{r_{1}}\right)\;\left(r_{t}⊕f\left(r\right),1\right)\\ & = & \left(0,r_{0}\right)\;\left(1,\bar{r_{1}}\right)\;\left(r_{t}⊕f\left(r\right),1\right) \end{array}$$
Measurement of the query register will reveal $x^{*}=\left(01\right)$ (see Figure 32). This will be true also for the other three functions as well, and we will return to show this later on.

### 7.3. The *n*-Toffoli

To extend this to larger systems we need a description of the simulation of an *n*-Toffoli. For a quantum 3-qubit Toffoli we have the identity of Figure 33.

We see that the corresponding QSL map is
(148)$$\begin{array}{ccc} \left(x_{2},p_{2}\right) & \; & \left(x_{2},p_{2}⊕t_{p}x_{0}x_{1}\right)\\ \left(x_{1},p_{1}\right) & \; & \left(x_{1},p_{1}⊕t_{p}x_{0}x_{2}\right)\\ \left(0,a_{1}\right) & \;\to \; & \left(0,a_{1}\right)\\ \left(x_{0},p_{0}\right) & \; & \left(x_{0},p_{0}⊕t_{p}x_{1}x_{2}\right)\\ \left(t_{x},t_{p}\right) & \; & (t_{x}⊕x_{0}x_{1}x_{2},t_{p}), \end{array}$$
which in QSL gives the relation shown in Figure 34.

Now using the same identity but by extending the 3-Toffoli to a 4-Toffoli (Figure 35) we get the map
(149)$$\begin{array}{c} \left(x_{3},p_{3}\right)\\ \left(x_{2},p_{2}\right)\\ \left(0,a_{1}\right)\\ \left(x_{1},p_{1}\right)\\ \left(x_{0},p_{0}\right)\\ \left(t_{x},t_{p}\right) \end{array}\;\to \;\begin{array}{c} \left(x_{3},p_{3}⊕t_{p}x_{0}x_{1}x_{2}\right)\\ \left(x_{2},p_{2}⊕t_{p}x_{0}x_{1}x_{3}\right)\\ \left(0,a_{1}\right)\\ \left(x_{1},p_{1}⊕t_{p}x_{0}x_{2}x_{3}\right)\\ \left(x_{0},p_{0}⊕t_{p}x_{1}x_{2}x_{3}\right)\\ (t_{x}⊕x_{0}x_{1}x_{2}x_{3},t_{p}). \end{array}$$

By induction, we see that in QSL the map corresponding to an *n*-Toffoli is
(150)$$\begin{array}{ccc} \left(x_{n},p_{n}\right) & \; & \left(x_{n},p_{n}⊕t_{p}(\prod_{k\neq n}x_{k})\right)\\ \;\vdots & & \;\vdots \\ \left(x_{1},p_{1}\right) & \;\to \; & \left(x_{1},p_{1}⊕t_{p}(\prod_{k\neq 1}x_{k})\right)\\ \left(x_{0},p_{0}\right) & \; & \left(x_{0},p_{0}⊕t_{p}(\prod_{k\neq 0}x_{k})\right)\\ \left(t_{x},t_{p}\right) & \; & \left(t_{x}⊕(\prod_{k}x_{k}),t_{p}\right) \end{array}$$

Using inverted controls by putting X before and after the control that is to be inverted, we see that this will not influence the value of the computational bits in the query register. The computational bit of the target and the phase bits of the query register will be affected, as the added X will induce the bit-complement for the corresponding $x_{i}$. With *f* as the function with only one satisfying assignment $x^{*}$,
(151)$$f\left(x\right)=\prod_{k}\left(x_{k}⊕\bar{x_{k}^{*}}\right),$$
built using a single *n*-Toffoli with some controls inverted, gives the QSL mapping
(152)$$\left(x,p\right)\;\left(t_{x},t_{p}\right)\to (x,p⊕t_{p}\tilde{f}\left(x\right))\;\left(t_{x}⊕f\left(x\right),t_{p}\right).$$
Here
(153)$${\tilde{f}}_{i}\left(x\right)=\prod_{k\neq i}\left(x_{k}⊕\bar{x_{k}^{*}}\right)=f\left(\sum_{k\neq i}2^{k}x_{k}+2^{i}x_{i}^{*}\right),$$
so to speak, the function with the $i^{th}$ bit of the argument set to the correct value of the satisfying assignment $x_{i}^{*}$.

### 7.4. A Scaling Algorithm

When increasing the number of input-bits the algorithm will have us to apply the Grover operator several times in sequence before measuring. A QSL simulation of this will not behave as the quantum algorithm, and we have found that the behavior is too far from the quantum behavior to give the quadratic speed-up of Grover’s algorithm. The underlying reason for this lack of speed-up needs more study, but we have found that simulating only the first application of the Grover operator gives a logarithmic speed-up, and this would give the following QSL algorithm.

The algorithm applies the Grover operator only once and measures immediately, and then proceeds to repeat this sequence, whereas standard Grover applies the Grover operator repeatedly, followed by a single measurement. Using our description of the *n*-Toffoli, we can now prove the following theorem.

**Theorem** **4.**
*Algorithm 4 solves Grovers problem with a bounded error probability using no more than $O\left(\frac{2^{n}}{n}\right)$ number of queries to the oracle.*


**Algorithm 4:** Repeat the following steps $O\left(\frac{2^{n}}{n}\right)$ timesPrepare an *n*-QSL-bit query register in the (0,r) state, and an answer register containing one QSL-bit in the $\left(1,r_{y}\right)$ state.Apply a Walsh-Hadamard transform on the query registerApply the oracleApply a Walsh-Hadamard transform on the query registerApply $X^{\times n}$ to the query register.Apply an *n*-controlled Z to the query register.Apply $X^{\times n}$ to the query register.Apply a Walsh-Hadamard transform on the query registerMeasure the query register to obtain a candidate *y* for $x^{*}$.  Check the output candidates for $x^{*}$ using the oracle, and return the candidate that outputs 1, if any.

**Proof.** Let us start by calculating the probability distribution from which the measurement samples.The first four steps of the algorithm produce
(154)$$\left(0,r\right)\;\left(1,r_{t}\right)\to \left(\tilde{f}\left(r\right),r\right)\;\left(r_{t}⊕f\left(r\right),1\right).$$
Applying steps five, six, and seven will have the equivalent effect on the query register as applying another *n*-Toffoli with all controls inverted, on a target system with the phase bit set. This corresponds to applying another Boolean function
(155)$$g\left(x\right)=\prod_{i}\bar{x_{i}}.$$
The effect on the query register is
(156)$$\left(\tilde{f}\left(r\right),r\right)\to \left(\tilde{f}\left(r\right),r⊕\tilde{g}(\tilde{f}\left(r\right))\right),$$
where ${\tilde{g}}_{i}$ is similar to ${\tilde{f}}_{i}$, so that ${\tilde{g}}_{i}$ is *g* with the correct bit-value used in position *i* (here 0, the satisfying assignment of *g*). Measuring after the final Walsh-Hadamard transform we sample from the bitwise distributio
(157)$$y_{i}=r_{i}⊕\underset{{\tilde{g}}_{i}\left(\tilde{f}\left(r\right)\right)}{\underset{︸}{\left(\prod_{k\neq i}\bar{{\tilde{f}}_{k}\left(r\right)}\right)}}$$
where *r* is the whole random bit string, *i* the bit-index, and ${\tilde{f}}_{k}$ the different functions described above. Let us split the analysis into three cases
If $r=x^{*}$, all ${\tilde{f}}_{i}\left(r\right)=1$, and all ${\tilde{g}}_{i}=0$, so that the output is $r=x^{*}$.If $r_{k}=x_{k}^{*}$ except for a single $r_{i}\neq x_{i}^{*}$ (*r* has a Hamming distance of 1 to $x^{*}$), then ${\tilde{f}}_{k}\left(r\right)=\delta_{ik}$ since ${\tilde{f}}_{i}\left(r\right)$ uses the correct value $x_{i}^{*}$ in place of $r_{i}$, while ${\tilde{f}}_{k}\left(r\right)=0$ for k≠i since the wrong bit-value $r_{i}$ is being used throughout. This in turn gives ${\tilde{g}}_{k}\left(\tilde{f}\left(r\right)\right)=\delta_{ik}$ by the same mechanism (the satisfying assignment of *g* is all zero values). Then, the output $r⊕\tilde{g}\left(\tilde{f}\left(r\right)\right)=r⊕\delta_{ik}=x^{*}$.When two or more bits are wrong, all ${\tilde{f}}_{k}\left(r\right)=0$, and all ${\tilde{g}}_{i}=1$. This will give as output the bitwise complement $\bar{r}$, which is not equal to $x^{*}$ except for when *all* bits are wrong, in which case $\bar{r}=x^{*}$.In short, we can think of this as a protocol where a random guess *r* is corrected to $x^{*}$ if there is a single-bit error in *r*, or if *r* is the bitwise complement of $x^{*}$. The sampling distribution therefore has a probability $p=\frac{n+2}{2^{n}}$ of obtaining $x^{*}$, probability p=0 for obtaining the bitwise complement $\bar{x^{*}}$ or a bit-string with Hamming distance of 1 to $x^{*}$, and probability $p=\frac{1}{2^{n}}$ for any other value. Repeating the subroutine $\kappa 2^{n}/\left(n+2\right)=O(2^{n}/n)$ times gives a probability of not obtaining $x^{*}$ even once less than $e^{-\kappa }$ (Theorem 5 of [59]). □

An alternative is to repeat the subroutine until the first occurrence of $x^{*}$. This gives a solution to the problem with error probability 0, but instead gives a random number of trials that obeys the shifted geometric distribution with probability $\left(n+2\right)/2^{n}$ of obtaining $x^{*}$. Then, the expected number of subroutine calls to the first success is $2^{n}/\left(n+2\right)$, so that the expected number of calls to the oracle including checking if the output $y=x^{*}$ is $2^{n+1}/\left(n+2\right)=O(2^{n}/n)$.

To connect with the fist example of one-shot Grover’s for when n=2. There is a total of four assignments, one correct, its bitwise complement, and two strings with one bit difference. The QSL simulation will therefore always return the correct assignment for the one-shot Grover’s.

### 7.5. Comparison with a 3-qubit Experiment

For three-bit input, running the protocol once, as in the above proposal, is a simulation of the setup used in a recent 3-qubit Grover experiment on trapped ions by Figgatt et al. [60]. The authors use two measures to characterize this experiment, the success probability of the algorithm and the squared statistical overlap (SSO). The SSO is the square of the Bhattacharyya coefficient (see Equation (Equation 26)) between the estimated probability distribution that the experiment samples from and the theoretical distribution.

While Figgatt et al. [60] obtain an average SSO of 83.2(7)%, our simulation gives 78.5%, which is slightly lower. On the other hand, their average success probability is 38.9(4)%, while our simulation of the same circuitry has a success probability of 5/8=62.5%, much closer to the theoretical 25/32≈78.1%. It is important to note that when we compare quantum and classical bounds for some problem, we compare the *best-known* bounds—of course including simulation algorithms.

### 7.6. Application to Ciphers

An application of Grover’s algorithm gives a quadratic speed-up to breaking most ciphers for which otherwise exhaustive search is the best-known method. Let us have a closer look at how that works.

Let us suppose that Alice has a classical system that uses a bijective map described by the function E(k,m)=c that takes as arguments a plaintext message *m*, a key *k*, and produces a ciphertext *c*. Alice uses this function to encrypt information and its inversion (as a map from *m* to *c*) to decrypt. When the same key *k* is used both for encryption and decryption this is called a symmetric cipher, and when not, an asymmetric cipher.

An attack that uses Grover’s algorithm is a *known-plain-text attack*. In such an attack, Eve possesses (or can obtain) both the plaintext and the corresponding ciphertext produced with Alice’s key. Let us for simplicity assume that the length of the known plaintext is longer than the unicity length, meaning it is long enough so that the key is uniquely determined by *E*, *m*, and *c*.

To retrieve *k*, Eve also needs to know *E*. That she has in her possession a copy of Alice’s system, or a schematic, can be justified with Kerckhoff’s principle stating that a secure cryptographic system should remain secure under the assumption that an adversary knows everything about the cryptographic system, except for the key.

Now assume that it is possible for Eve to efficiently translate Alice’s classical device into a quantum device, and that this process does not give her any information about the key. Then she can set up her system to encrypt $|m\rangle$ and add 1 to a second register, conditioned on seeing $|c\rangle$ (see Figure 36). This unitary can now be used to find *k* in $O(\sqrt{2^{n}})$ time using Grover’s algorithm.

We conjecture that this attack can be done within the QSL framework. However, it remains to be analyzed for which specific cases this does work since the behavior depends on the implementation. Also, note that the advantage from the above algorithm is smaller, only the factor 1/n in $O(2^{n}/n)$.

## 8. Simon’s Algorithm

Simon’s problem [15,16] is the archetype of quantum exponential speed-up, as the first case of a relativized exponential gap between **BPP** and **BQP**. Its method has been reused and generalized many times over, and was the inspiration for Shor’s algorithm [17,18].

### 8.1. Problem Formulation

We will work from the following problem formulation

**Definition** **5.**
*Consider that we are given access to an oracle encoding a function $f:{\left\{0,1\right\}}^{n}\to {\left\{0,1\right\}}^{n}$, and promised that it is either one-to-one or two-to-one and invariant under a non-trivial XOR-mask s.*


A function being invariant under a non-trivial XOR-mask *s* means that the function *f* yield the same output on two different inputs, *x* and $x^{\prime }$, only if these two inputs differ exactly on those bit positions where the bits of the mask *s* are 1. In other words, $f\left(x\right)=f\left(x^{\prime }\right)$ if and only if $x=x^{\prime }⊕s$.

### 8.2. Probabilistic Solution

Assuming that we only have access to an oracle that only gives us the output of the function $f:{\left\{0,1\right\}}^{n}\to {\left\{0,1\right\}}^{n}$. Simon proved the following theorem.

**Theorem** **5**(Simon [15,16]). *Let O be an oracle constructed as follows: for each n, a random n-bit string s(n) and a random bit b(n) are uniformly chosen from ${\left\{0,1\right\}}^{n}$ and {0,1}, respectively. If b(n)=0, then the function $f_{n}:{\left\{0,1\right\}}^{n}\to {\left\{0,1\right\}}^{n}$ chosen for O to compute on n-bit queries is a random function uniformly distributed over permutations on ${\left\{0,1\right\}}^{n}$; otherwise, it is a random function uniformly distributed over two-to-one functions such that $f_{n}\left(x\right)=f_{n}\left(x⊕s\left(n\right)\right)$ for all x, where* ⊕ *denotes bitwise exclusive-or. Then any PTM that queries O no more than $2^{n/4}$ times cannot correctly guess b(n) with probability greater than $\left(1/2\right)+2^{-n/2}$, over choices made in the construction of O.*

The proof can be found in Ref. [16], here we will instead give a simple motivation as to why this will require an exponential number of queries. Consider an algorithm that queries the oracle on random inputs. If the algorithm finds the same function value twice it halts, and we can conclude that the function is two-to-one and invariant under some XOR-mask *s*. Now, if the queries are chosen from a uniform distribution (there is no point in choosing input from another distribution since *s* is chosen uniformly) the range of the function can also be seen as a probability distribution. If the function is one-to-one it will be uniform. If the function is instead two-to-one, half of the sample space will have p=0 and the rest will be uniformly distributed with $p=\frac{2}{2^{n}}$. It is, therefore, unlikely that an algorithm using less than an exponential number of queries will see the same value twice.

### 8.3. Quantum Algorithm

Let us now assume that we are given access to an oracle that encodes the function in the unitary transformation
(158)$$|x\rangle |y\rangle \overset{U_{f}}{\to }|x\rangle |y⊕f(x)\rangle .$$
Simon’s algorithm (Algorithm 5) can then make use of the subroutine shown in Figure 37 (see also Subroutine 6) and proceed as follows
(159)$$\begin{array}{ccc} |0\rangle |0\rangle & \overset{H^{⊗n}⊗I^{⊗n}}{\to } & \frac{1}{2^{n/2}}\sum_{x}|x\rangle |0\rangle \\ & \overset{U_{f}}{\to } & \frac{1}{2^{n/2}}\sum_{x}|x\rangle |f(x)\rangle \\ & \overset{H^{⊗n}⊗I^{⊗n}}{\to } & \frac{1}{2^{n-b/2}}\sum_{x,p}{\left(-1\right)}^{x\cdot p}|p\rangle |f(x)\rangle , \end{array}$$
where the scalar product is defined mod 2. The normalization constant on the last row depends on whether *f* is one-to-one (b=0) or two-to-one (b=1), and the difference is because of subtractive interference in the two-to-one case, see below.

If the function is one-to-one, measuring the query register will yield an outcome *p* drawn from a uniform distribution. If the function is two-to-one there will be destructive interference so that some terms cancel in the sum, and using f(x)=f(x⊕s) the output can be written
(160)$$\begin{array}{cc} \frac{1}{2^{n-1/2}}\sum_{x,p}{\left(-1\right)}^{x\cdot p}|p\rangle |f(x)\rangle & =\frac{1}{2^{n+1/2}}\sum_{x,p}{\left(-1\right)}^{x\cdot p}|p\rangle \left(|f(x)\rangle +|f(x⊕s)\rangle \right)\\ & =\frac{1}{2^{n+1/2}}\sum_{x,p}\left({\left(-1\right)}^{x\cdot p}+{\left(-1\right)}^{(x⊕s)\cdot p}\right)|p\rangle |f(x)\rangle \\ & =\frac{1}{2^{n+1/2}}\sum_{x,p}{\left(-1\right)}^{x\cdot p}\left(1+{\left(-1\right)}^{s\cdot p}\right)|p\rangle |f(x)\rangle \\ & =\frac{1}{2^{n-1/2}}\sum_{x,p:s\cdot p=0}{\left(-1\right)}^{x\cdot p}|p\rangle |f(x)\rangle \end{array}$$
This will yield an outcome drawn from a distribution that is uniform over values *p* such that s·p=0.

Each call to the subroutine gives independent outcomes so after an expected linear number of rounds, we will have collected n-1 linearly independent bit-vectors. Solving the resulting linear system of equations will give a non-trivial solution $s^{*}$ that will be our candidate for *s*. If the function is one-to-one, $s^{*}$ will be a random bit-vector. By using $s^{*}$ to query the function twice, for f(x) and $f(x⊕s^{*})$ for some *x*, we obtain the solution to the problem. If the function is two-to-one and invariant under the XOR-mask $s^{*}$, the two queries will yield the same result, otherwise not.

**Algorithm 5:** Simon [15,16]  Proceed with the following steps. Make n-1 repetitions of Subroutine 6 (also shown in Figure 37). This will with high probability yield n-1 linearly independent bit-vectors orthogonal to *s*.Solve the resulting linear system of equations to obtain the candidate solution $s^{*}$Query the function for f(x) and $f(x⊕s^{*})$, for some *x*.  If the two queries give the same value, then the function is two-to-one and invariant under the XOR-mask $s=s^{*}$, and otherwise not.

**Algorithm 6:** Simon [15,16]  Proceed with the following steps. Prepare two *n*-qubit registers in the state $|0\rangle$.Apply the Walsh-Hadamard transform to the query register.Apply the oracle.Apply the Walsh-Hadamard transform to the query register.Measure the query register to retrieve a bit-vector *p*.  Return *p*.

### 8.4. QSL Simulation

Theorem 5 does of course not apply to the quantum case where the function oracle is given as a unitary map with *n* qubits as input and *n* qubits as target, and not a function from *n* bits to *n* bits. Neither does it apply to the QSL framework since the oracle is a map that uses *n* QSL-bits as input and *n* QSL-bits as target, or if one prefers, from 2n bits to 2n bits in its classical simulation.

The base construct that we will use for Simon’s subroutine is that used in Tame et al. [61] (see their supplementary material) and consists of a network of *CNOT* gates. It is generated by choosing a basis ${\left\{v^{\left(k\right)}\right\}}_{k=0}^{n-2}$ for a (perhaps *the*) subspace of $Z_{2}^{n}$ orthogonal to *s*, and storing the scalar product between the input *x* and $v^{\left(k\right)}$ in the *k*-th bit of the output *y*. This is done by connecting a *CNOT* between the input bit $x_{i}$ and the output bit $y_{k}$ for each entry where $v_{i}^{\left(k\right)}=1$, producing the mapping
(161)$$|x\rangle |y\rangle \to |x\rangle |y⊕f^{\left(1\right)}\left(x\right)\rangle$$
where
(162)$$f_{k}^{\left(1\right)}\left(x\right)=\left\{\begin{array}{cc} 0, & k=n-1\\ x\cdot v^{\left(k\right)}=\left(x⊕s\right)\cdot v^{\left(k\right)}, & \mathrm{otherwise}. \end{array}\right.$$
The resulting function $f^{\left(1\right)}\left(x\right)$ is a two-to-one function invariant under the XOR-mask *s*. We label the resulting gate array $U_{s}$.

As an example, consider the 3-qubit case where s=(101). the bit-vector basis can be chosen to {(101),(010)}. From $v^{\left(0\right)}=\left(101\right)$ we get two *CNOT*s with their controls at $|x_{0}\rangle$ and $|x_{2}\rangle$, and target at $|y_{0}\rangle$. From $v^{\left(1\right)}=\left(010\right)$ a *CNOT* with control at $|x_{1}\rangle$ and target at $|y_{1}\rangle$ (see Figure 38).

For the one-to-one maps, we add a map $V_{s}$ that uses a final vector $v^{\left(n-1\right)}$ needed to complete the basis to a basis for the whole $Z_{2}^{n}$. Since we can choose any vector not orthogonal to *s*, we simply pick a vector with a single-bit set out of the bits set in *s* (which is possible as soon as s≠0). To construct $V_{x}$, connect a *CNOT* from $x_{i}$ to $y_{n-1}$ for the single $v_{i}^{\left(n-1\right)}=1$. This gives the map
(163)$$|x\rangle |f^{\left(1\right)}\left(x\right)\rangle \to |x\rangle |y⊕f^{\left(0\right)}\left(x\right)\rangle$$
where
(164)$$f_{k}^{\left(0\right)}\left(x\right)=x\cdot v^{\left(k\right)}$$
This equals $f_{k}^{\left(1\right)}$ except for the case k=n-1 for which
(165)$$f_{n-1}^{\left(0\right)}\left(x⊕s\right)=\left(x⊕s\right)\cdot v^{\left(n-1\right)}=\bar{f_{n-1}^{\left(0\right)}\left(x\right)},$$
making $f^{\left(0\right)}\left(x\right)$ a one-to-one function.

The complete construction is shown in Figure 39 and consists of three parts: the $U_{s}$ map, the $V_{s}$ map controlled by a bit *b* that decides if *f* is one-to-one or two-to-one, and a permutation such as the one used in Section 5.

Creating a controlled $V_{s}$ consists simply of exchanging the single *CNOT* with a single Toffoli with the second inverted control attached to the control bit *b*, giving the map $f^{\left(1\right)}\left(x\right)\to f^{\left(b\right)}\left(x\right)$. To produce all possible one-to-one and two-to-one functions of this type from $f^{\left(b\right)}\left(x\right)$, the output needs to be fed into an arbitrary permutation $\pi_{f}$ (of the number states) which makes $\pi_{f}\left(f^{\left(b\right)}\left(x\right)\right)=f\left(x\right)$. Please note that if the function is two-to-one, the permutation will preserve the invariance under addition of *s*. Also note that the use of a generic permutation prohibits a general $|y\rangle$ to be used since $\pi_{f}\left(y⊕f^{\prime }\left(x\right)\right)$ does not necessarily equal $y⊕\pi_{f}\left(f^{\prime }\left(x\right)\right)$, thus the use of y=0 in the construction. For a general *y*, see Section 8.5.

The overall implementation of the oracle will give

(166)$$|x\rangle |0\rangle \overset{U_{f}}{\to }|x\rangle |\pi_{f}\left(f^{\left(b\right)}\left(x\right)\right)\rangle =|x\rangle |f(x)\rangle .$$

Next we need to find out how this behaves in QSL. Each CNOT gives a phase kickback to the controlling QSL-bit, and there is a CNOT between input QSL-bit $x_{i}$ and the output QSL-bit $y_{k}$ for each entry where $v_{i}^{\left(k\right)}=1$. Therefore, for each phase bit $c_{k}$ set in the output, the corresponding $v^{\left(k\right)}$ is added to the phase of the input. The total effect of ${}_{s}$ is to add
(167)$$g^{\left(1\right)}\left(c\right)=\sum_{k=0}^{n-2}c_{k}v^{\left(k\right)}\;\left(\mathrm{mod}\;2\right)$$
to the phase of *x*, while $V_{s}$ adds $c_{n-1}v^{\left(n-1\right)}$ if enabled, creating
(168)$$g^{\left(0\right)}\left(c\right)=\sum_{k=0}^{n-1}c_{k}v^{\left(k\right)}\;\left(\mathrm{mod}\;2\right).$$
Omitting the control system (b,a) (but including a non-zero target *y*), the effect of the QSL oracle is
(169)$$\begin{array}{ccc} \left(x,p\right)\;\left(y,c\right) & \overset{U_{s}}{\to } & \left(x,a⊕g^{\left(1\right)}\left(c\right)\right)\;\left(y⊕f^{\left(1\right)}\left(x\right),c\right)\\ & \overset{{\mathrm{CV}}_{s}}{\to } & \left(x,a⊕g^{\left(b\right)}\left(c\right)\right)\;\left(y⊕f^{\left(b\right)}\left(x\right),c\right)\\ & \overset{\pi_{f}}{\to } & \left(x,a⊕g^{\left(b\right)}\left(c\right)\right)\;\left(\pi_{f}\left(y⊕f^{\left(b\right)}\left(x\right)\right),\pi_{f,y⊕f^{\left(b\right)}\left(x\right)}\left(c\right)\right), \end{array}$$
where the phase permutation notation from Equation (Equation 123) becomes the somewhat cumbersome $\pi_{f,y⊕f^{\left(b\right)}\left(x\right)}\left(c\right)$. Using this to simulate the entire subroutine in Figure 37, we get
(170)$$\begin{array}{ccc} \left(0,r\right)\;\left(0,r^{\prime }\right) & \overset{H\times I}{\to } & \left(r,0\right)\;\left(0,r^{\prime }\right)\\ & \overset{O_{f}}{\to } & \left(r,g^{\left(b\right)}\left(r^{\prime }\right)\right)\;\left(f\left(r\right),\pi_{f,f^{\left(b\right)}\left(r\right)}\left(r^{\prime }\right)\right)\\ & \overset{H\times I}{\to } & \left(g^{\left(b\right)}\left(r^{\prime }\right),r\right)\;\left(f\left(r\right),\pi_{f,f^{\left(b\right)}\left(r\right)}\left(r^{\prime }\right)\right) \end{array}$$
Measuring the query register will yield $g^{\left(b\right)}\left(r^{\prime }\right)$, i.e., a linear combination of vectors $v^{\left(k\right)}$ with $r_{k}^{\prime }$ as coefficients. The state preparation in each round of the subroutine creates independent and uniformly distributed random bit-vectors $r^{\prime }$ (and *r*). Therefore, if b=1 the output will be a random vector uniformly distributed over the subspace orthogonal to *s*, and if b=0 the output is a random vector uniformly distributed over the entire $Z_{2}^{n}$. Thus, the simulation samples from the same probability distribution as the quantum algorithm, and will also solve the decision problem with the same expected linear number of queries to the oracle as the quantum algorithm. We have now arrived at

**Theorem** **6.**
*There is an efficient QSL algorithm that solves*
Simon’s
*problem with an expected O(n) number of queries query to the oracle. Relative to the oracle, this QSL algorithm has an efficient classical simulation on a PTM, in accordance with Lemma 1.*


This shows that relative to this oracle, the problem is in **BPP**.

### 8.5. Adding the Function Output to the Target Modulo 2

We noted earlier that the oracle in Figure 37 does not produce a unitary map that adds the function value mod 2 to the output, because of the final number-state permutation. To allow a non-zero-initialized answer register so that the map becomes $|x\rangle |y\rangle \to |x\rangle |y⊕f(x)\rangle$, Bennett’s trick of uncomputation [26] can be used (see Figure 40).

In this case, the first three gates ($U_{s}$, ${\mathrm{CV}}_{s}$, and $\pi_{f}$) performs the map (only writing what happens in the query register and ancilla),
(171)$$\begin{array}{cc} \left(x,p)\;(0,r\right) & \to \;\left(x,p⊕g^{\left(b\right)}\left(r\right)\right)\;\left(f\left(x\right),\pi_{f,f^{\left(b\right)}\left(x\right)}\left(r\right)\right). \end{array}$$
The CNOT array will add the value f(x) to the target register (y,c) modulo 2 and cause a phase kickback from the target register,
(172)$$(x,p⊕g^{\left(b\right)}\left(r\right))\;\left(f\left(x\right),\pi_{f,f^{\left(b\right)}\left(x\right)}\left(r\right)\right)\to \left(x,p⊕g^{\left(b\right)}\left(r\right)\right)\;\left(f\left(x\right),\pi_{f,f^{\left(b\right)}\left(x\right)}\left(r\right)⊕c\right).$$
Finally, letting $r^{\prime }$ equal the output of the inverted permutation,
(173)$$r^{\prime }=\pi_{f,f^{\left(b\right)}\left(r\right)}^{-1}\left(\pi_{f,f^{\left(b\right)}\left(x\right)}\left(r\right)⊕c\right),$$
the final three gates ($\pi_{f}^{†}$, ${\mathrm{CV}}_{s}^{†}$, and $U_{s}^{†}$) will invert the function map and cause another addition of $g^{\left(b\right)}$ in the phase of the input system,
(174)$$(x,p⊕g^{\left(b\right)}\left(r\right))\;\left(f\left(x\right),\pi_{f,f^{\left(b\right)}\left(x\right)}\left(r\right)⊕c\right)\to \left(x,p⊕g^{\left(b\right)}\left(r\right)⊕g^{\left(b\right)}\left(r^{\prime }\right)\right)\;\left(0,r^{\prime }\right).$$
When used in Simon’s algorithm, p=0, and *r* and *c* are independent uniformly distributed random values, which make $r^{\prime }$ independent of *r* and uniformly distributed. The output of the algorithm is the phase of the query register, which equals
(175)$$g^{\left(b\right)}\left(r\right)⊕g^{\left(b\right)}\left(r^{\prime }\right)$$
As the two arguments to *g* are independent and randomly distributed, the sample retrieved from the measurement will be drawn from the uniform distribution over all bit-strings if b=0, or the strings orthogonal to *s* if b=1. This gives the same output distribution as the quantum algorithm, and the theorem holds in this case as well.

### 8.6. A Deterministic Algorithm for Simon’s Problem

An interesting extension of this algorithm is the one devised by Brassard and Hoyer [62], which relative to an oracle decides the problem in exact quantum polynomial time **EQP**, meaning that the number of calls to the oracle is deterministic and polynomial, and that the error probability is zero. This algorithm was considered relativized evidence for an exponential separation between **EQP** and **BPP**. In [14] we developed a similar algorithm for the case when the oracle is of the form used in Section 8.4, or in general, when the bias enters the function evaluation as
(176)$$|x\rangle |y\rangle \to |x\rangle |f_{y}\left(x\right)\rangle =|x\rangle |\pi_{f}\left(y⊕U_{0}\left(x⊕(x\cdot v_{s})s\right)\right)\rangle ,$$
where $U_{0}$ is a binary orthogonal map (i.e., one-to-one, and preserves scalar products), $v_{s}$ is a vector with $v_{s}\cdot v_{s}=v_{s}\cdot s=1$ unless s=0 in which case $v_{s}=0$, and $\pi_{f}$ is a permutation as before. Please note that the construction enables all $f\left(x\right)=f_{0}\left(x\right)$, and that
(177)$$\left(x⊕s\right)⊕\left(\left(x⊕s\right)\cdot v_{s}\right)s=x⊕(x\cdot v_{s})s,$$
so that $f_{y}\left(x⊕s\right)=f_{y}\left(x\right)$. The difference to the standard construction is that bias enters differently, since the standard construction gives $f_{y}^{\prime }\left(x\right)=y⊕f\left(x\right)$.

The algorithm, as compared to the standard algorithm, adds a Walsh-Hadamard transform on the answer register input, and then simply iterates through bit-strings $\delta_{k}$ that have only the *k*th bit set on the answer register input. A quantum oracle that obeys Equation (Equation 176) (see Figure 41) give the following effect in the algorithm,
(178)$$\begin{array}{ccc} |0\rangle |p\rangle & \overset{H^{⊗n}⊗H^{⊗n}}{\to } & \sum_{x,y}{\left(-1\right)}^{y\cdot p}|x\rangle |y\rangle \\ & \overset{y⊕U_{0}\left(\ldots \right)}{\to } & \sum_{x,y}{\left(-1\right)}^{y\cdot p}|x\rangle |y⊕U_{0}\left(x⊕(x\cdot v_{s})s\right)\rangle \\ & = & \sum_{x,y}{\left(-1\right)}^{\left(y⊕U_{0}\left(x⊕\left(x\cdot v_{s}\right)s\right)\right)\cdot p}|x\rangle |y\rangle \\ & = & \sum_{x,y}{\left(-1\right)}^{x\cdot \left(U_{0}^{-1}p⊕\left(s\cdot U_{0}^{-1}p\right)v_{s}\right)⊕\left(y\cdot p\right)}|x\rangle |y\rangle \\ & \overset{H^{⊗n}⊗\pi_{f}}{\to } & |U_{0}^{-1}p⊕\left(s\cdot U_{0}^{-1}p\right)v_{s}\rangle \sum_{y}{\left(-1\right)}^{y\cdot p}|\pi_{f}\left(y\right)\rangle . \end{array}$$
A measurement on the query register will give the output $U_{0}^{-1}p⊕\left(s\cdot U_{0}^{-1}p\right)v_{s}$, which is orthogonal to *s* because
(179)$$s\cdot \left(U_{0}^{-1}p⊕\left(s\cdot U_{0}^{-1}p\right)v_{s}\right)=\left(s\cdot U_{0}^{-1}p\right)⊕\left(s\cdot U_{0}^{-1}p\right)\left(s\cdot v_{s}\right)=0,$$
and the output from letting *p* iterate over a basis for bit-strings of length *n* will span the subspace orthogonal to *s*. The important difference to the previous analysis is that the present form preserves orthogonality in the phase kickback, whereas there is no such guarantee needed in Section 8.5. It is now easy to check if the output subspace has dimension *n* (the *n* outputs are linearly independent) or n-1. The quantum algorithm solves the problem with error probability zero in linear time, so relative to this oracle, this problem is in **EQP**.

The behavior will be exactly the same for the QSL simulation, but here we only analyze the explicit choices
(180)$$U_{0}={\left[v^{\left(1\right)},v^{\left(2\right)},\ldots ,v^{\left(n\right)}\right]}^{T}$$
and if s≠0,
(181)$$v_{s}=v^{\left(n\right)}.$$
This gives us the explicit construction of Section 8.4, for which
(182)$$\begin{array}{ccc} \left(0,x\right)\left(\delta_{k},y\right) & \overset{H^{\times n}\times H^{\times n}}{\to } & \left(x,0\right)\;\left(y,\delta_{k}\right)\\ & \overset{U_{s}\circ {\mathrm{CV}}_{s}}{\to } & \left(x,v^{\left(k\right)}\right)\;\left(y⊕f^{\left(b\right)}\left(x\right),\delta_{k}\right)\\ & \overset{H^{\times n}\times \pi_{f}}{\to } & \left(v^{\left(k\right)},x\right)\;\left(\pi_{f}\left(y⊕f^{\left(b\right)}\left(x\right)\right),\pi_{f,y⊕f^{\left(b\right)}\left(x\right)}\left(\delta_{k}\right)\right), \end{array}$$
for k≤n-2 and
(183)$$\begin{array}{ccc} \left(0,x\right)\;\left(\delta_{n-1},y\right) & \overset{H^{\times n}\times H^{\times n}}{\to } & \left(x,0\right)\left(y,\delta_{n-1}\right)\\ & \overset{U_{s}\circ {\mathrm{CV}}_{s}}{\to } & \left(x,\bar{b}v^{\left(n-1\right)}\right)\;\left(y⊕f^{\left(b\right)}\left(x\right),\delta_{n-1}\right)\\ & \overset{H^{\times n}\times \pi_{f}}{\to } & \left(\bar{b}v^{\left(n-1\right)},x\right)\;\left(\pi_{f}\left(y⊕f^{\left(b\right)}\left(x\right)\right),\pi_{f,y⊕f^{\left(b\right)}\left(x\right)}\left(\delta_{n-1}\right)\right). \end{array}$$
Again, the iteration will produce measurement results from the query register that span the subspace of bit-strings *v* that obey v·bs=0. The output is exactly the same as in the quantum case. Thus, relative to this QSL oracle, this problem is in **P**. By modifying the effect of bias on the target register, we have moved from a problem in a probabilistic class to a problem in the deterministic class of the same complexity.

### 8.7. Application to Symmetric Ciphers

Simon’s algorithm has been used to attack symmetric-key ciphers, retrieving the key in a linear number of calls to the encryption device. There are several results that lead up to this.

Kuwakado and Morii [63] showed that if built as a unitary, a 3-round Feistel network with internal permutations can be distinguished from a random permutation using Simon’s algorithm. If we look at the 3-round Feistel network as a block cipher running in the Electronic Code Book (ECB) mode, it permutes a block of the plaintext into a block of the ciphertext. If this permutation is indistinguishable from a random permutation, and therefore unpredictable, the best an adversary running a known-plain-text attack can do, is to search the whole key space. However, if the permutation can be distinguished from a random permutation, then we get knowledge about the system and the key space that we need to search shrinks. The result of Kuwakado and Morii [63] show us that after the application of Simon’s algorithm there should be a speed-up of the key recovery algorithm, but not how large the speed-up is.

Kuwakado and Morii [64] later showed that Simon’s algorithm can be used to retrieve the key from the Evan-Mansour cipher by employing several queries that are linear in the key size. This gives an exponential speed-up to the best-known classical query bound, and a significant speed-up compared to the method using Grover’s algorithm described in Section 7.

Kaplan et al. [65] extended this to other ciphers, and modes of operations, and showed that it can give an exponential speed-up some classical attacks that require finding collisions.

As always, for this to work, the encrypting device needs to be re-cast as a unitary operation and the key needs to be stored in the device. Please note that here, the key is a classical parameter to the encryption device, and not under control of an attacker or entered as a quantum state in contrast to the known plaintext attack using Grover’s algorithm in Section 7.6. From a practical point of view, this means that the key or the secret needs to be built into the construction, just as in our previous realization of Simon’s oracle, where *s* is used to build an explicit gate array. Furthermore, an attacker can retrieve the built-in secret because it is built into the dynamics of a physical system. Again, when compared to the previous realization, it is clear that the secret leaks through another degree of freedom in the physical system, and not through the computational basis that holds the cleartext or ciphertext.

The conclusion is that this is a side-channel attack, i.e., an attack made available because of a specific implementation—not an attack on the protocol itself. Cryptographic protocols that are vulnerable to side-channel attacks are in general not considered broken, since all are more or less vulnerable depending on how they are implemented.

With that said, the results of [63,64,65] are important examples of new and innovative ways of performing side-channel attacks, in this case using a quantum-computer implementation of the cipher, and having the owner of the secret hide it as a parameter within the quantum computer. We agree with the recommendation of [64] that classical ciphers should not be implemented in quantum circuits, because of potentially introducing side-channel attacks. A large part of modern cryptography is to prevent the creation of unintended side channels in new encryption devices. It is still an open question whether QSL enables a similar side-channel attack on these kinds of systems, but for the same reason as with quantum computation, we would still advise against using QSL systems for encryption or decryption purposes.

## 9. Shor’s Algorithm Factoring 15

Shor’s algorithm [17,18] is one of the quantum algorithms that solve a computational problem with real-world applications: to efficiently find a factor in a composite number *N*. We have seen that Deutsch–Jozsa’s and Simon’s algorithms can be run in polynomial time on a classical Turing machine; the natural continuation is to attempt integer factorization, whose hardness is one of the most widely believed conjectures in computer science.

While large high-fidelity quantum computers are still far away, several experimental realizations of Shor’s algorithm for small numbers have been presented [66,67,68,69,70,71,72]. These are all very impressive demonstration of quantum optimal control, but an experimental realization of Shor’s algorithm with the currently available technology is demanding and this has led to the need for vast simplifications in the algorithm. There are essentially two parameters subject to optimization: bit-depth and circuit-depth (see Ref. [57] and the citations therein). Also, the  approximate quantum Fourier transform [73] is crucial for scalability. In this optimization procedure, one must be careful not to over-simplify, or make explicit (or implicit) use of knowledge about the solution [74]. We will avoid such over-simplifications by using an identical circuit for factoring 15 as in Ref. [66].

Shor’s algorithm uses a quantum subroutine that finds the order (or period) of an element *a* in the multiplicative group of integers modulo *N*. Here, the order is the smallest integer *r* such that $a^{r}=1$ (mod *N*). This is sufficient information to find a factor in *N*. The algorithm makes use of an input-register quantum state, containing an integer *x*, and an output-register in which modular exponentiation $a^{x}$ (mod *N*) is computed. By  creating a superposition in the input-register using the quantum Fourier transform, performing the calculation, and then inverting the transform, one can with high probability retrieve sufficient information to calculate the order (see Figure 42).

More specifically, this procedure let us sample from a probability distribution with peaks at s/r, where *s* is uniformly distributed over the integers between 0 and r-1. Ideally, the peaks are completely localized to s/r but in most cases there is peak broadening due to Fourier leakage, and to ensure that the measurement yields a binary fraction sufficiently close to s/r, the input-register needs to be at least 2n qubits in size, where *n* is the size of the output-register which is large enough to perform calculations mod *N*. The full procedure to retrieve *r* is given in Algorithm 7.    

**Algorithm 7:** Shor [17]  Proceed with the following steps to retrieve a factor of a composite number with highprobability. Pick at random an integer a≠±1 modulo *N*. If GCD(a,N) is a non-trivial factor of *N*, wehave a solution.Otherwise generate, setup, and run the quantum Subroutine (also shown inFigure 42) to find a candidate for s/r.Use the continued fraction expansion to retrieve *r* (or a factor in *r* when *s* and *r* has a common factor).  If *r* is even, one of GCD$\left(a^{r/2}\pm 1,N\right)$ may be a non-trivial factor of *N*. This happens with highprobability.

**Algorithm 8:** Shor [17]  Proceed with the following steps to find a candidate for the quotient s/r where *s* is randomand *r* is the period of $a^{x}$ mod *N*. Prepare a 2n-qubit query register in the state $|0\rangle$ and an *n*-qubit answer register in thestate $|1\rangle$.Apply the quantum Fourier transform to the query register.Apply the specific unitary $U_{a,N}|x\rangle |y\rangle =|x\rangle |y\times a^{x}\;\mathrm{mod}\;N\rangle$.Apply the inverse quantum Fourier transform to the query register.Measure the query register to retrieve a candidate for s/r.  Return that candidate for s/r. 

For N=15 the possible integers that can occur in steps 2–4 are a∈{2,4,7,8,11,13} (see Figure 43). Please note that deliberately choosing *a* to give a short period that is easy to find, would be another example of over-simplifying the algorithm. It is, therefore, important that the element *a* is chosen randomly see e.g., [74]. In what follows, we have used all alternatives; this is of course only possible because of the small *N* used.

Some useful simplifications *are* allowed. The Fourier transform on the input can be exchanged for the Walsh-Hadamard transform (Hadamard gates), while the inverse Fourier transform can be implemented through Hadamards followed by classically controlled single qubit rotations [75]; by advancing the measurement of the controlling qubit and using the outcome as a classical control (see Figure 44). This decouples the 2n qubits of the input-register in the sense that the procedure of preparation, transformation, and measurement can be performed individually on each qubit. It is common to perform these single qubit procedures in sequence on one single qubit, a method known as qubit recycling, which reduces the overall bit-depth from 3n to n+1 at the cost of circuit-depth.

The demonstration of Monz et al. [66] is the most advanced to date. They refrain from “compiling” the circuitry, and use non-Clifford group operations which is necessary for a scalable version. One simplification Monz et al. do use is restricting the resolution of the input-register from 2n qubits to three qubits. This is possible because all elements in the multiplicative group mod 15 have power-of-two periods, and they only verify the behavior of the exponentiation until and including the first that is equivalent to the identity map (see Figure 43).

It has been argued [76] that the three-bit precision of Monz et al. is insufficient since 2n bits are required for the algorithm to overcome Fourier leakage in general, and to succeed with a bounded error rate required for scalability. However, for N=15 there is no Fourier leakage because of the power-of-two periods and this is clear when building the quantum gate array. Therefore, measuring more qubits will only add noise and not precision. The distribution is completely described at two bits of precision. On the other hand, this also means that the process of generating the circuitry solves the factoring problem, since it is enough to know at which point the identity emerges from exponentiation. Thus, the experiment [66] is not so much about factoring 15 through Shor’s algorithm, but more a verification that their quantum (ion-trap) realization is good enough to approximate the ideal quantum statistics even when not using a compiled circuit.

We now present an experimental realization of Shor’s algorithm, factoring 15, using QSL. Please note that the motivation for doing this is not to factor 15, but to compare our completely classical construction with the current state-of-the-art quantum implementations, and to show that QSL is applicable outside the oracle paradigm. We use basically the same algorithmic setup, but employ the semiclassical Fourier transform without qubit recycling, and the emerging identity operations are of course all omitted. Our setup is similar to that of Monz et al. (see Figure 44) and uses the multiplication operators from Markov et al. [57], but avoids precomputing their effect on the initial state (see Figure 43).

We have chosen a hardware implementation in 2-complementary reversible pass-transistor logic, specifically using transmission gates [77]. These are built using currently available semiconductor technology and operate in the digital regime using TTL levels, see Figure 45. The random number generators of the source and measurement devices use high-gain analogue amplifiers to sample random bits from noise at a reverse-biased PN junction.

The output probability distributions are estimated from $10^{6}$ samples for each element a∈{2,4,7,8,11,13} (see Figure 46). We see that the distributions for a=7,8,13 are not uniform over their support, as predicted by quantum theory, but they are closer to the ideal distributions than any other implementation that we know of. To analyze their performance, Monz et al. used the *square statistical overlap* (SSO) as a fidelity measure. Our implementation reaches an SSO of {0.9999(1), 0.9999(1), 0.933(3), 0.984(2), 0.9999(1), 0.984(2)} for a∈{2,4,7,8,11,13} respectively (statistical errors as one standard deviation). Notably, our implementation gives the same probability (0.5) of returning a good candidate for *r* as the ideal quantum subroutine.

Like any other physical implementation of quantum gates, our simulation suffers from systematic errors. This will result in an error propagation that suppresses the amount of useful information that we can retrieve when scaling the algorithm to larger numbers—even though there is no practical restriction for doing so. Further work is needed, beyond the scope of this paper, to reduce these systematic errors, perhaps by altering the framework or using error correcting techniques, so that it becomes useful for larger instances.

## 10. Conclusions

In this paper, we have looked at the QSL simulation framework, which is efficiently simulatable on a classical Turing machine and can at the same time reproduce many quantum phenomena. These phenomena include results from quantum query complexity in which the query complexity is exactly the same in QSL as in quantum theory. In the circuit model there is only a constant overhead in classical resources, which in turn gives a polynomial time simulation in a classical probabilistic Turing machine.

Using this approach there is no quantum advantage with respect to the QSL implementation for the Deutsch-Jozsa, Bernstein-Vazirani, or Simon’s problem. For Grover search QSL has a slight advantage over the classical case, reaching $O(2^{n}/n)$ queries compared to $O(2^{n})$ in the classical case, while the quantum algorithm retains an advantage using $O(2^{n/2})$ queries. Outside the oracle paradigm, the choices made in the QSL construction gives less of an advantage, but does enable small examples, which we have illustrated by simulating the setup of two recent experimental realizations of Shor’s and Grover’s algorithms.

Regarding Deutsch-Jozsa, Bernstein-Vazirani, and Simon’s problem we can also conclude that no genuinely quantum resources are required, i.e., that the resources used can be efficiently simulated on a classical Turing machine. The reason is that the resources needed are also available in QSL. The inclusion of phase information in QSL gives rise to phenomena that have previously been suggested resources for the quantum advantage, including superposition, interference, and some aspects of entanglement, among others. We believe that it is more appropriate to assign the speed-up in these cases to the ability to store, process, and retrieve information in another information-carrying degree of freedom, in this case the phase degree of freedom.

Functions in quantum computation are realized as dynamics from physical systems to physical systems, and not as mere function maps from bits to bits. Tracing the full dynamics of such a physical system may require keeping track of an exponential amount of information. The available efforts to efficiently trace the dynamics used in quantum computation usually adopt a top-down approach, restricting the available dynamics until the necessary information can be efficiently traced. For instance, the stabilizer sub-theory is the restriction to Pauli measurements and Clifford gates, and this can be traced efficiently. The approach presented here instead uses a bottom-up approach that extends the standard function maps to also encompass the phase information within each individual qubit. Even though simulating each qubit with a single classical bit is not enough, we have found that two classical bits per qubit gets us very far.

This clearly shows that comparing bounds from classical query complexity with those from quantum query complexity does not provide conclusive evidence for a quantum advantage. A quantum oracle is so vastly different from a classical oracle that it is questionable whether they are comparable at all. The comparison between quantum oracle and QSL oracle is much more well-founded, since both enable a choice between retrieving function value or additional information related to the function, stored in the phase degree of freedom. All it takes to enable the same behavior for a QSL oracle as a quantum oracle, in the above examples, is a single bit of extra (phase) information, i.e., a constant overhead. This leaves a lot of headroom for improvement, as allowing for a polynomial overhead will enable a more accurate simulation.

In conclusion, the enabling root cause (resource) for the quantum speed-up in the mentioned examples is not superposition, interference, entanglement, contextuality, the continuity of quantum state space, or quantum coherence. It is the ability to store, process, and retrieve information in an additional information-carrying degree of freedom in the physical system being used as information carrier.

## Figures and Tables

**Figure 1 entropy-21-00800-f001:**
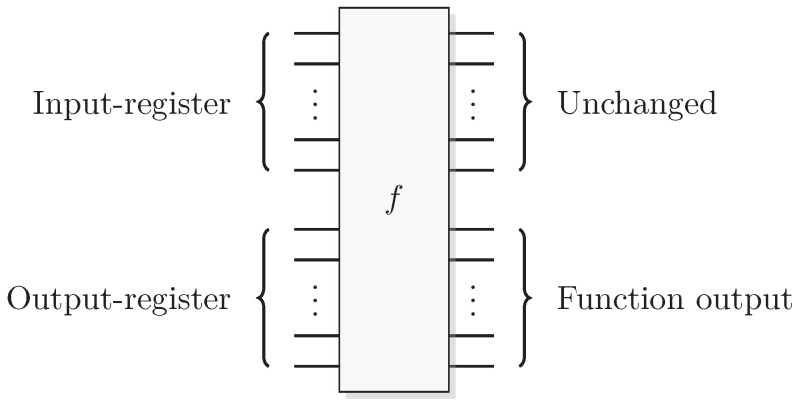
A classical reversible logical function construction for a function *f*.

**Figure 2 entropy-21-00800-f002:**
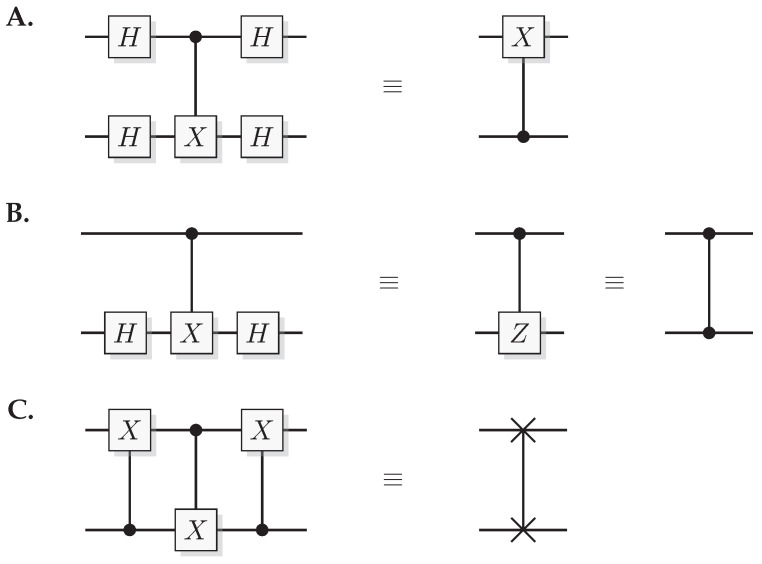
Identities for two-qubit quantum gates. (**A**) The effect of a quantum CNOT in the Hadamard basis. (**B**) relation between CNOT and CZ. (**C**) construction of a SWAP gate from three CNOT gates.

**Figure 3 entropy-21-00800-f003:**
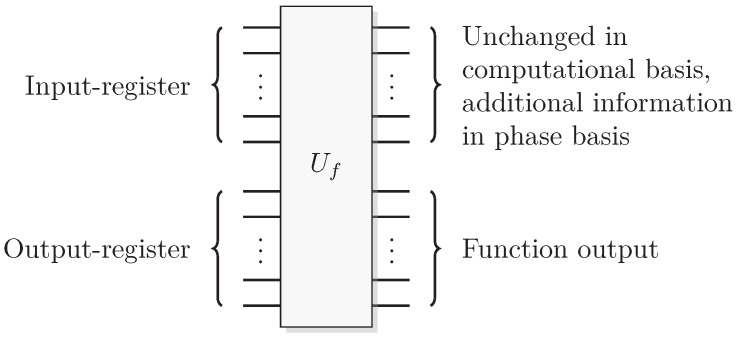
A reversible logical function. In the quantum case we can choose to retrieve function output or some additional information from the output of the query register.

**Figure 4 entropy-21-00800-f004:**
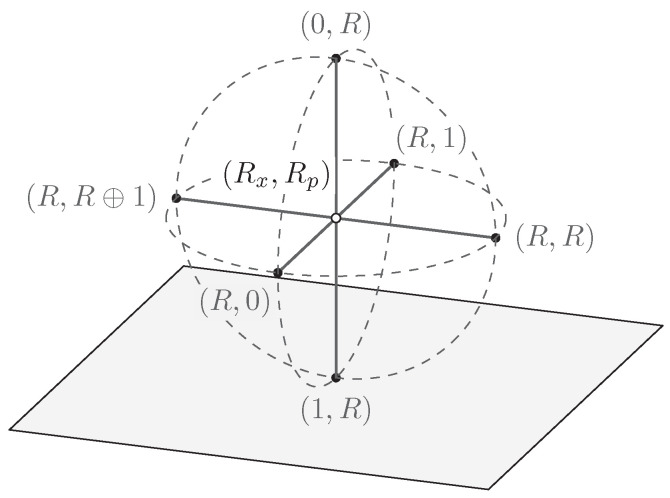
Geometric representation of the seven states of an elementary QSL system and their position relative to the Bloch sphere.

**Figure 5 entropy-21-00800-f005:**
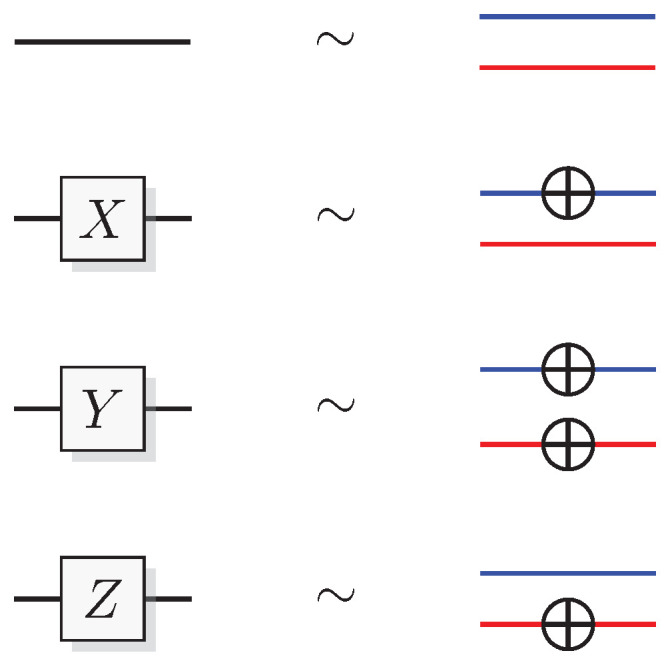
Graphical representation of the simulation of the Pauli gates I,X,Y and *Z*. Blue line segments represent the computational bit and red represents the phase bit.

**Figure 6 entropy-21-00800-f006:**
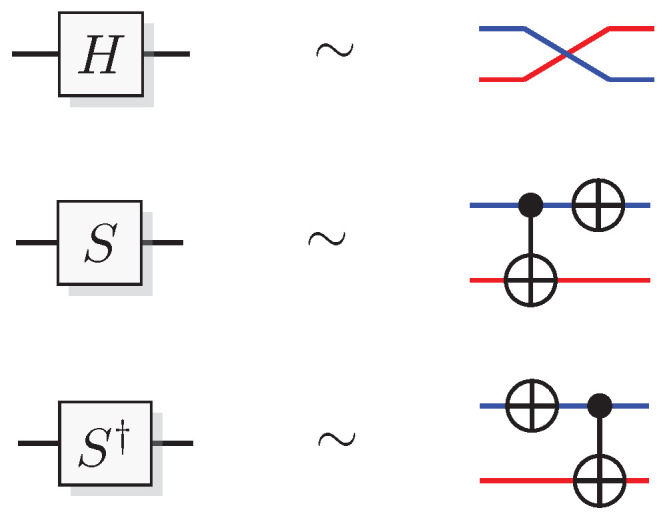
Graphical representation for the simulation of Hadamard, *S*, and $S^{†}$ gates.

**Figure 7 entropy-21-00800-f007:**
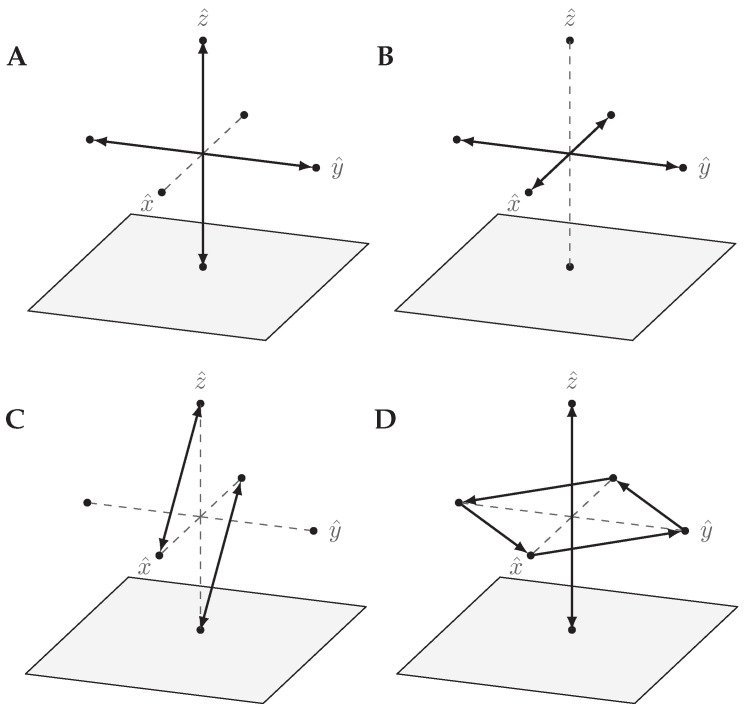
Effect of the transformations on the six QSL states in the Bloch representation. (**A**–**D**) shows the X, Z, H, and S gates, respectively.

**Figure 8 entropy-21-00800-f008:**
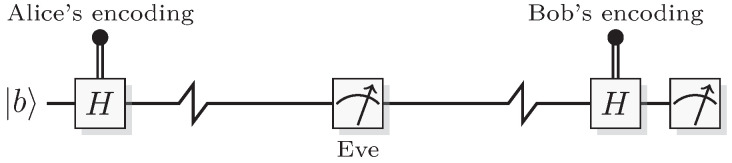
The BB84 protocol with an eavesdropper present.

**Figure 9 entropy-21-00800-f009:**
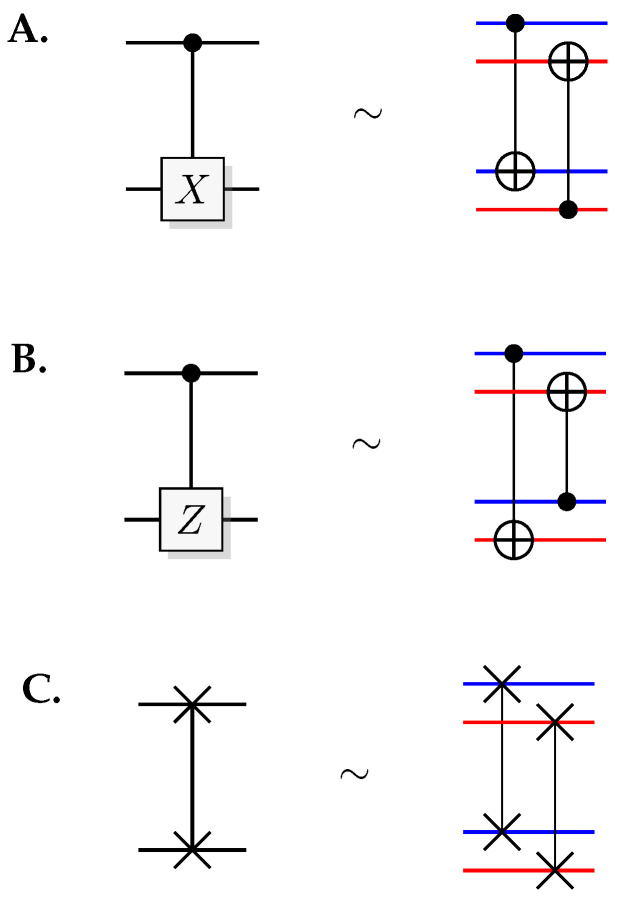
Graphical representation of the QSL analogue of two-qubit quantum gates. (**A**) *CNOT*, (**B**) controlled-*Z*, and (**C**) *SWAP*, all constructed to uphold the identities in Figure 2.

**Figure 10 entropy-21-00800-f010:**
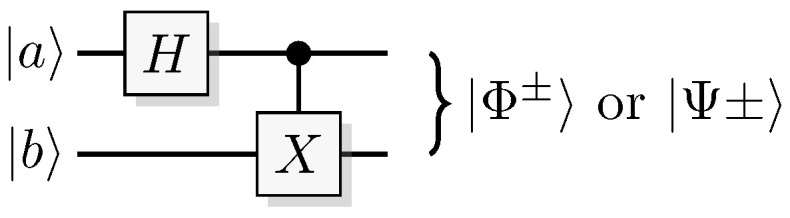
Circuit for generating the four Bell states.

**Figure 11 entropy-21-00800-f011:**
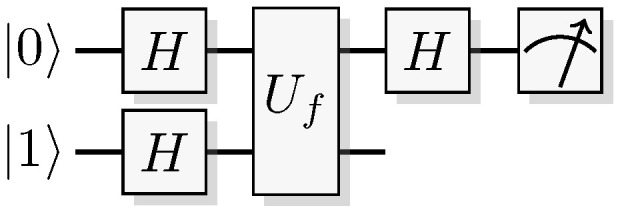
Circuit for the Deutsch algorithm used to illustrate interference.

**Figure 12 entropy-21-00800-f012:**
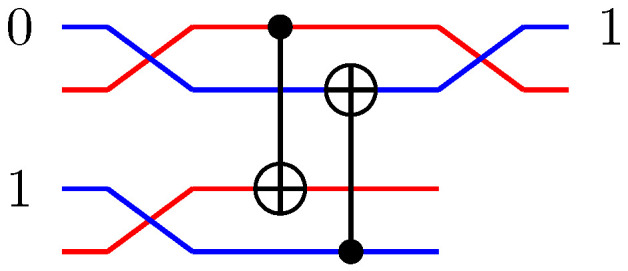
Example of QSL simulating interference in Deutsch algorithm when f(x)=x.

**Figure 13 entropy-21-00800-f013:**
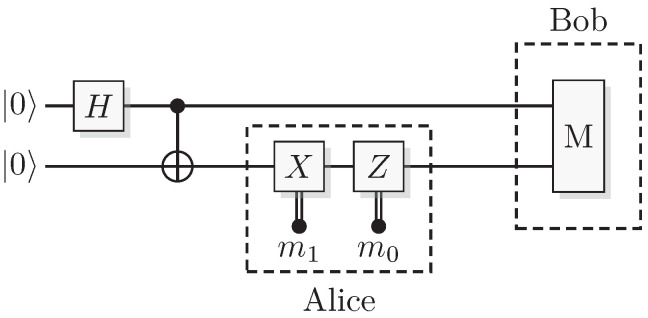
Protocol for superdense coding. Alice can convey two bits of information $m_{1}$ and $m_{0}$ to Bob by only interacting with one qubit of a correlated pair.

**Figure 14 entropy-21-00800-f014:**
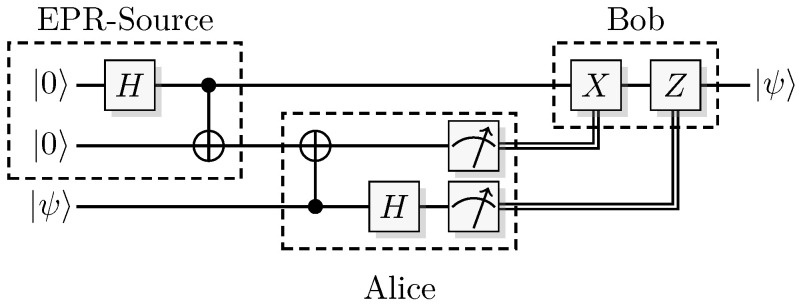
The teleportation protocol.

**Figure 15 entropy-21-00800-f015:**
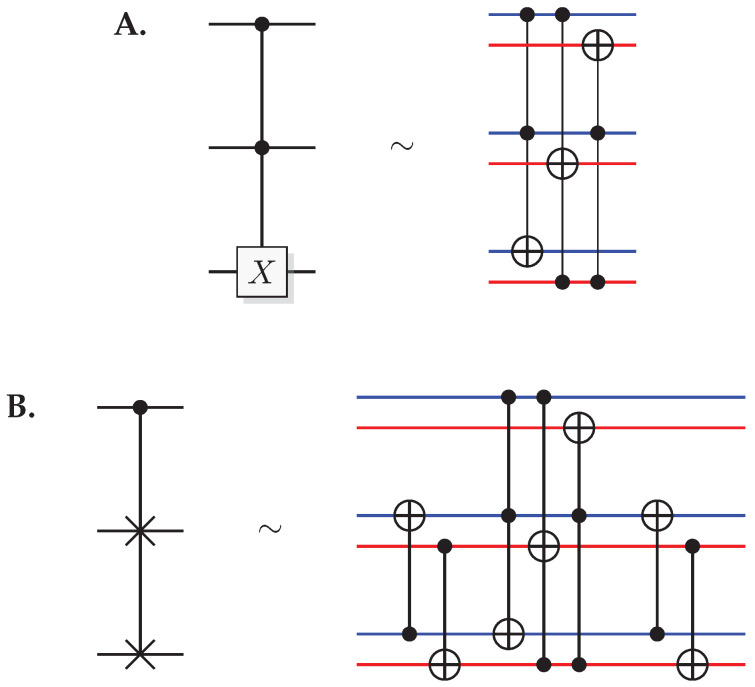
QSL construction of the Toffoli gate (**A**) and Fredkin (**B**), constructed to uphold the identities in Figure 16.

**Figure 16 entropy-21-00800-f016:**
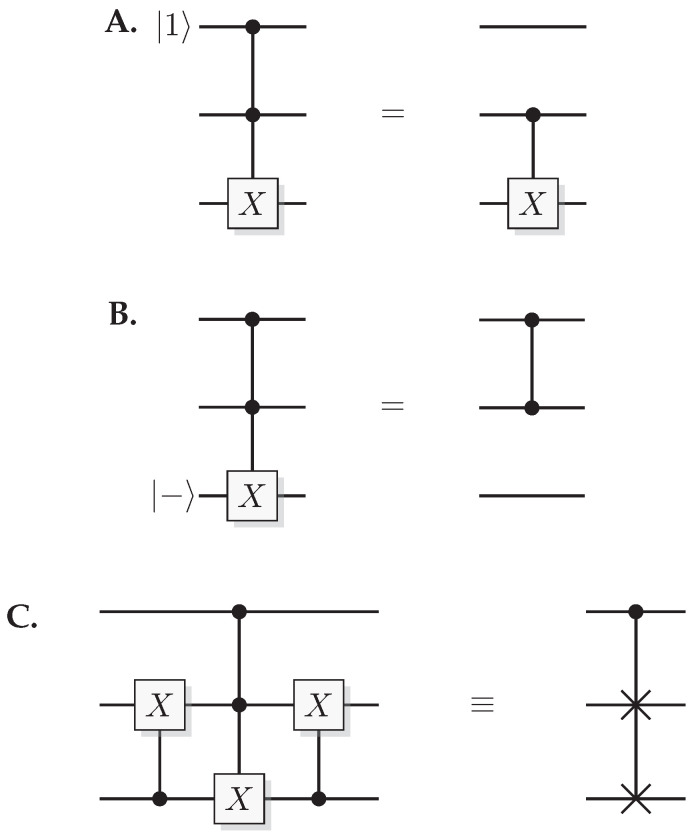
Quantum gate identities. (**A**) Toffoli gate with one of the controlling qubits initiated in $|1\rangle$ results in a *CNOT* over the other two qubits. (**B**) Toffoli gate with the target qubit initiated in $|-\rangle$ results in a CZ over the control qubits. (**C**) Identity connecting the Toffoli and Fredkin gate.

**Figure 17 entropy-21-00800-f017:**
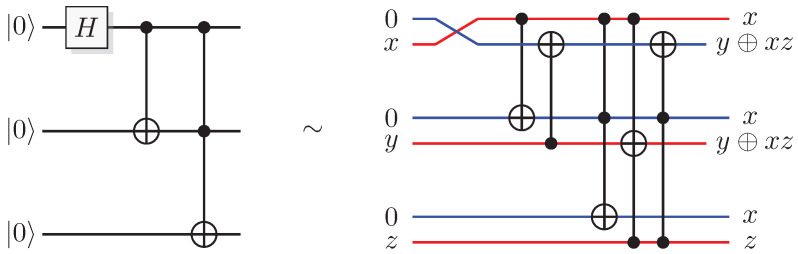
One construction that produces a GHZ-state, and the matching QSL construction.

**Figure 18 entropy-21-00800-f018:**
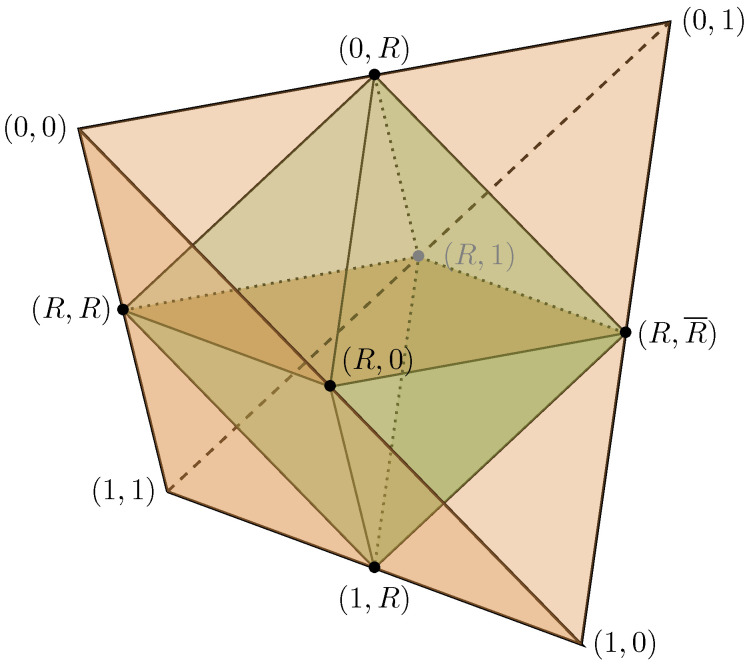
Representation of the state space of a single elementary system, with a geometry of a tetrahedron inscribing an octahedron. To aid with the visualization: if we take the vertices of the tetrahedron and fold them to one of the closest vertices on the octahedron, we recover the octahedron.

**Figure 19 entropy-21-00800-f019:**
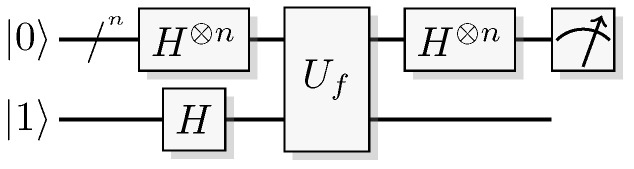
Quantum circuit for the Bernstein-Vazirani algorithm.

**Figure 20 entropy-21-00800-f020:**
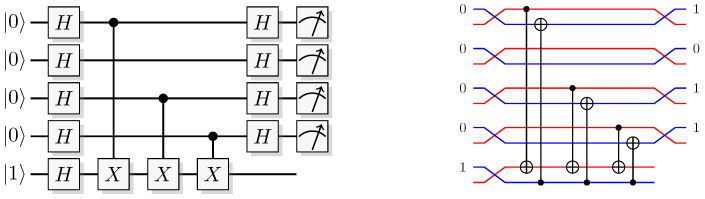
Example of a quantum and the corresponding QSL algorithm for solving the Bernstein-Vazirani problem when the secret string is s=(1011). The unsigned red inputs and outputs of the QSL circuit are all uniformly distributed random bits.

**Figure 21 entropy-21-00800-f021:**
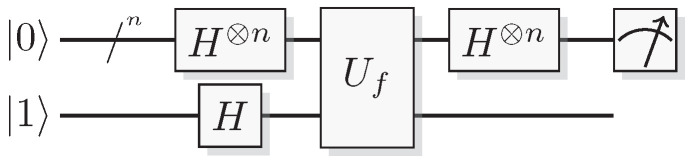
Circuit representation of the Deutsch-Jozsa algorithm.

**Figure 22 entropy-21-00800-f022:**
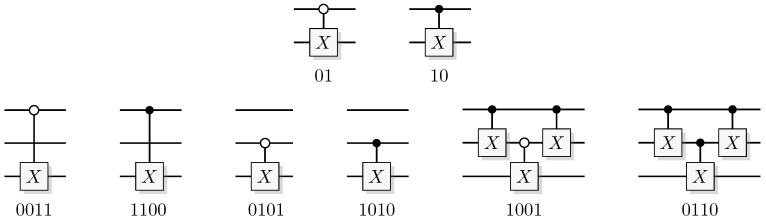
Explicit implementations of all balanced oracles for the Deutsch-Jozsa algorithm for one and two-qubit input. White controls are inverted.

**Figure 23 entropy-21-00800-f023:**
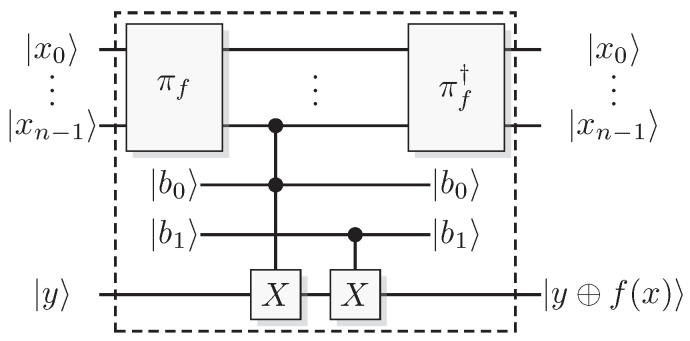
A specific implementation of a quantum oracle that can be used to solve the Deutsch-Jozsa problem in Definition 2. The internal parameter $b_{0}$ chooses between a constant or balanced function, and if the function is constant, the value is $b_{1}$. For balanced functions, the idea is that the first Toffoli acting as a *CNOT* to produce one specific balanced function (if the permutation $\pi_{f}$ is the identity), adding the most significant bit to the output, $f\left(x\right)=x_{n-1}$. Access to all balanced functions is then obtained by an arbitrary permutation of the output.

**Figure 24 entropy-21-00800-f024:**
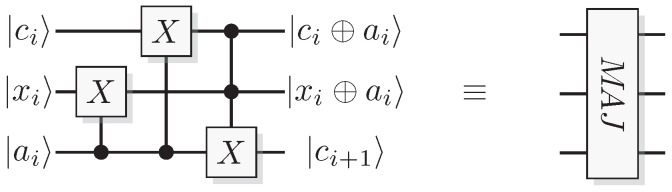
A majority-function MAJ module where the output carry $c_{i+1}=a_{i}⊕\left(c_{i}⊕a_{i}\right)\left(x_{i}⊕a_{i}\right)=a_{i}c_{i}⊕a_{i}x_{i}⊕c_{i}x_{i}$ contains the majority value of the three input bits.

**Figure 25 entropy-21-00800-f025:**
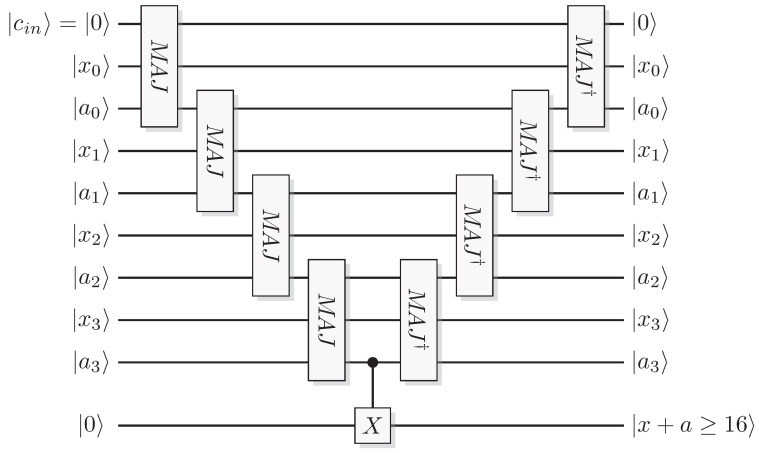
Quantum circuit of a reversible comparator for comparing the sum of two 4-bit integers *x* and *a* to the number $2^{4}$, signaling if $x+a\geq 2^{4}$ in the answer register.

**Figure 26 entropy-21-00800-f026:**
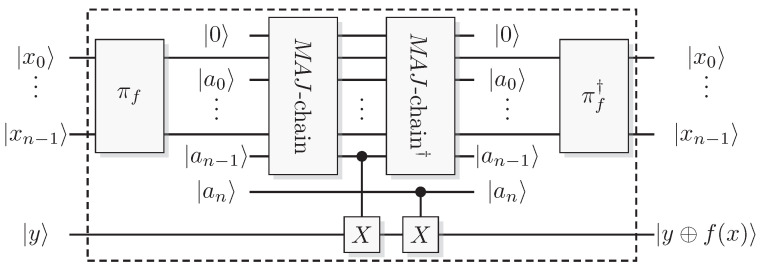
Construction of a circuit that can give any Boolean function. The idea is to construct one specific Boolean function for each number $0\leq a\leq 2^{n}$ that gives the output 1 on *a* inputs by using a comparator. Then produce the other functions by using a permutation of the input states. The comparator is built through two chains of MAJ and $M\;AJ^{†}$ gates as in Figure 25.

**Figure 27 entropy-21-00800-f027:**
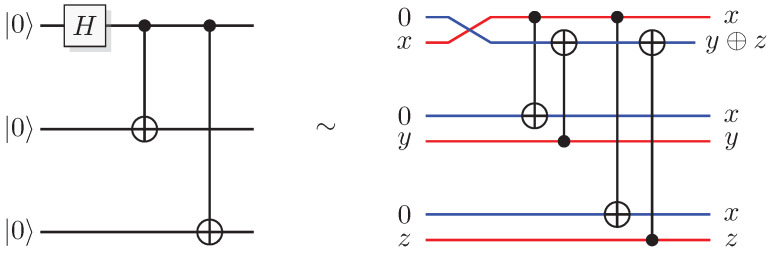
Another construction that produces a GHZ-state, and the analogue construction in QSL.

**Figure 28 entropy-21-00800-f028:**
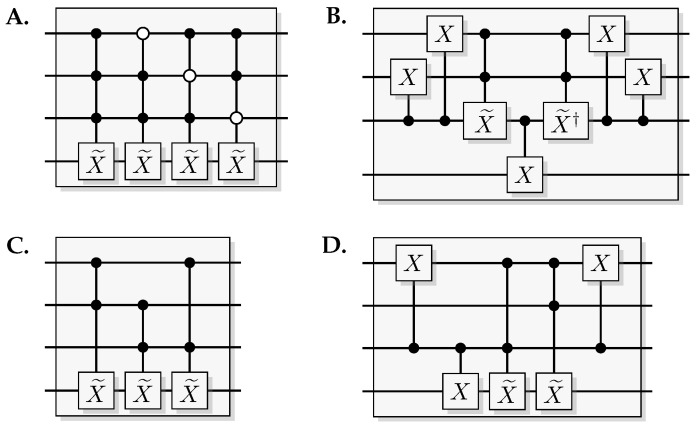
Four different quantum circuit implementations of the same balanced function, computing the majority MAJ (see Figure 24) of three bits. (**A**) Straightforward construction using four $\tilde{T}\left(3\right)$ gates. (**B**) The majority is computed in place in the query register, the answer is copied out to the answer register, and the intermediate step is then uncomputed by applying $M\;AJ^{†}$ to the query register. (**C**) is an optimization of A, tuned for as few gates as possible, and (**D**) is another optimized for few Toffolis followed by few CNOTs, generated as part of the list of all 72 possible functions in Appendix A.

**Figure 29 entropy-21-00800-f029:**
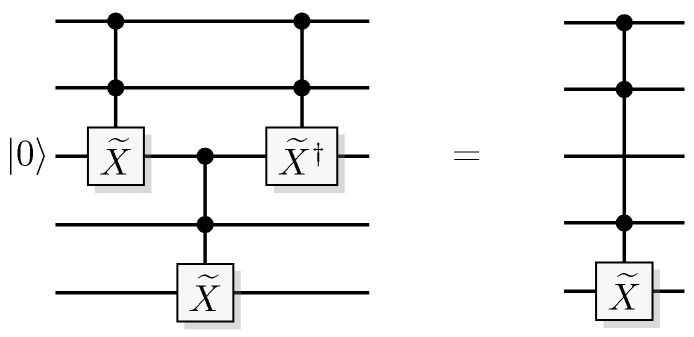
Scheme for constructing a $\tilde{T}\left(3\right)$ from three $\tilde{T}$ gates with a systematic error and an ancillary qubit initiated in $|0\rangle$.

**Figure 30 entropy-21-00800-f030:**
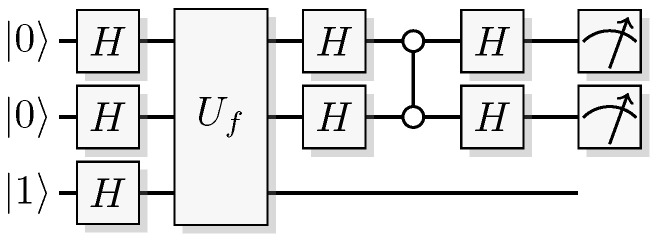
Circuit for the one-shot Grover instance. The second-to-last gate is a two-qubit controlled phase flip (controlled-*Z*) with both controls inverted.

**Figure 31 entropy-21-00800-f031:**
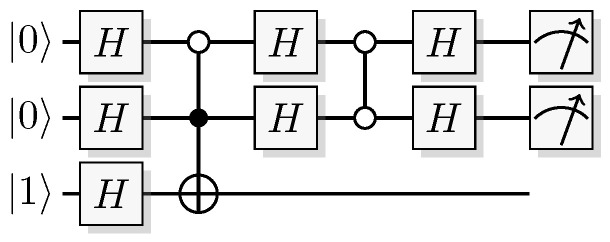
Example circuit of the one-shot Grover where $f\left(x\right)=\bar{x_{1}}x_{0}$.

**Figure 32 entropy-21-00800-f032:**
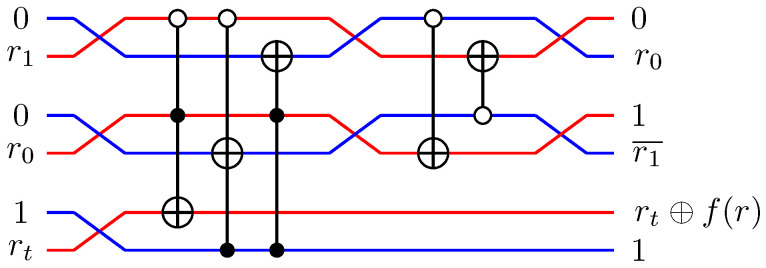
QSL simulation of the one-shot Grover instance from Figure 31.

**Figure 33 entropy-21-00800-f033:**
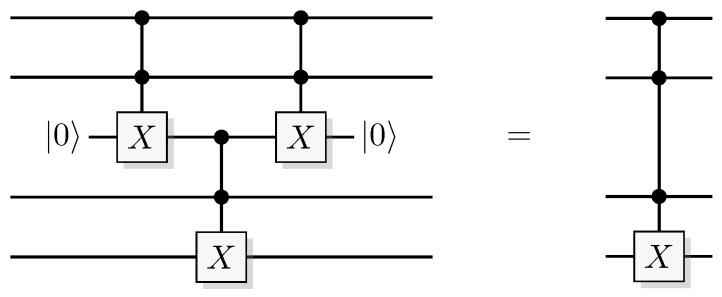
Scheme for construction a 3-Toffoli from three regular Toffoli gates and an ancillary qubit initiated in $|0\rangle$.

**Figure 34 entropy-21-00800-f034:**
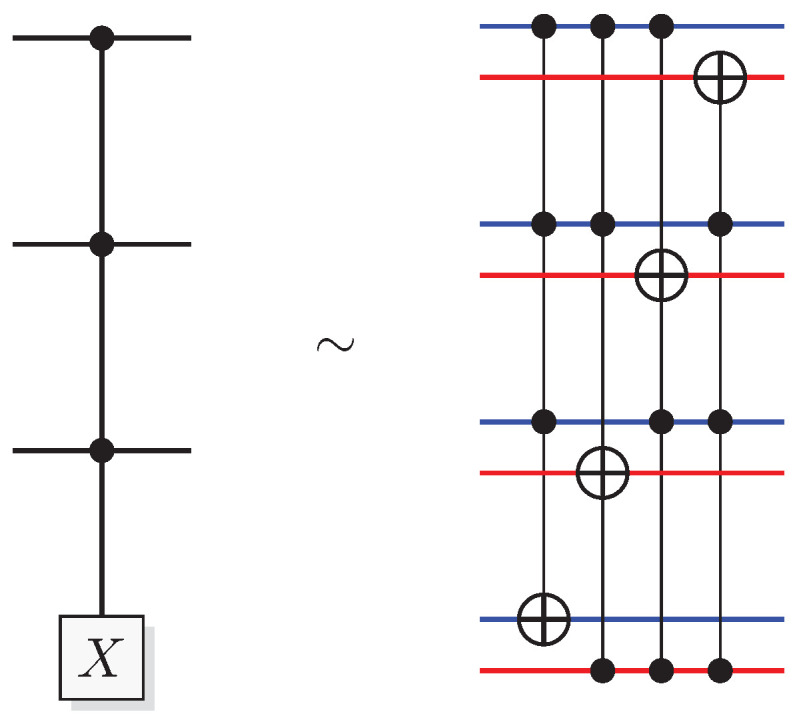
QSL identity for the 3-Toffoli.

**Figure 35 entropy-21-00800-f035:**
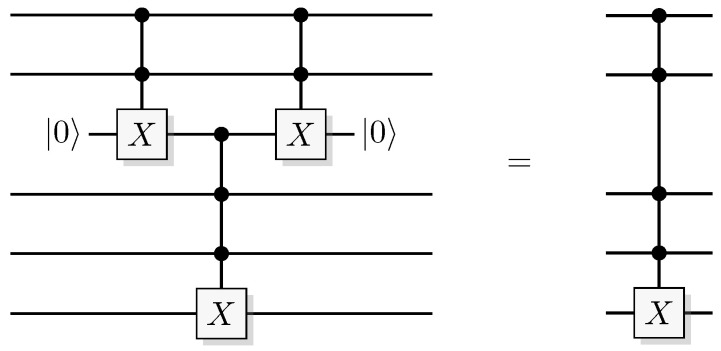
Scheme for construction a 4-Toffoli from one 3-Toffoli, two regular Toffoli gates, and an ancillary qubit initiated in $|0\rangle$.

**Figure 36 entropy-21-00800-f036:**
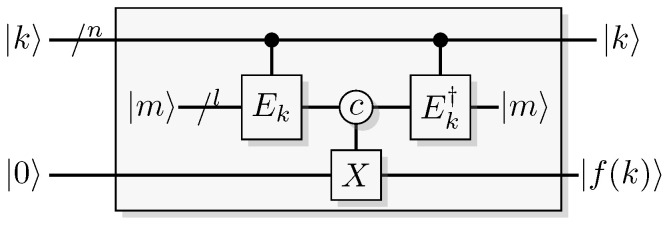
Circuit construction for using Grover’s algorithm to perform a known plaintext attack on a symmetric cipher $E_{k}$, where the key *k* has a bitsize of *n* and the known plaintext a bitsize of *l*. The control labeled *c* expresses that *X* is applied only if the output of $E_{k}\;|m\rangle$ is $|c\rangle$, inverting the target bit. This can be constructed with an *l*-Toffoli with regular and inverted controls.

**Figure 37 entropy-21-00800-f037:**
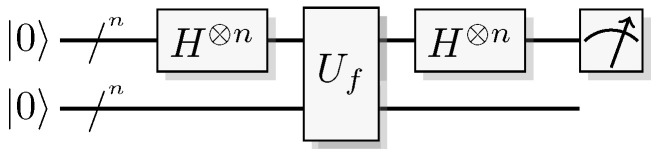
Quantum circuit used as a subroutine in Simon’s algorithm. The subroutine assumes oracle access to $U_{f}$.

**Figure 38 entropy-21-00800-f038:**
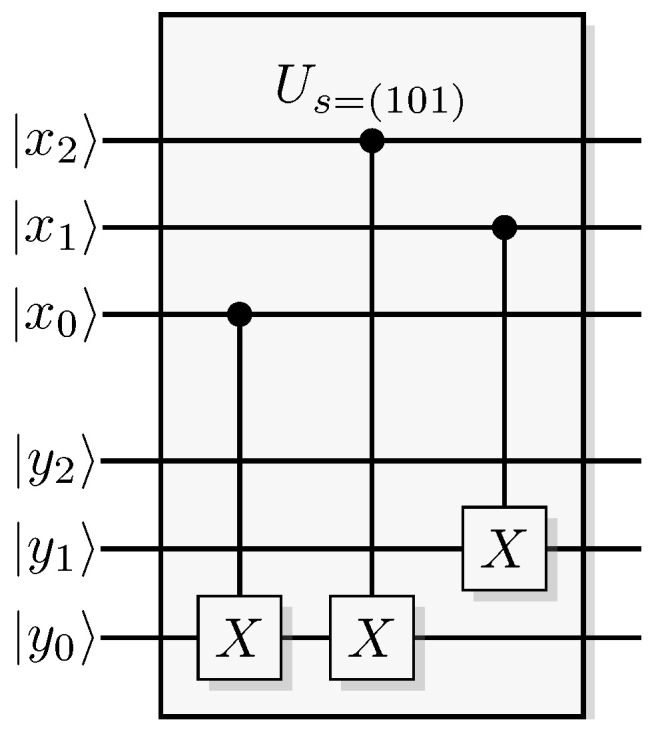
Example construction for $U_{s}$, where s=(101) and ${\left\{v^{\left(k\right)}\right\}}_{k=0}^{1}=\left\{\left(101\right),\left(010\right)\right\}$.

**Figure 39 entropy-21-00800-f039:**
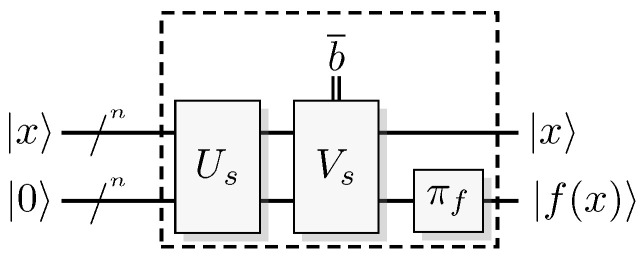
Oracle construction for Simon’s Subroutine.

**Figure 40 entropy-21-00800-f040:**
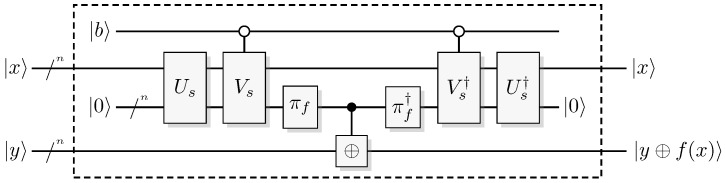
Oracle construction for Simons algorithm where the output can be initiated in a non-zero state. The boxed modulo 2 addition denotes an array of *CNOT*s that adds each ancilla bit to the corresponding target bit.

**Figure 41 entropy-21-00800-f041:**
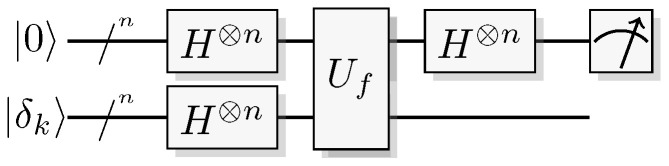
Proposed quantum circuit as subroutine in an algorithm solving Simon’s problem in polynomial time, both in quantum theory, and in QSL.

**Figure 42 entropy-21-00800-f042:**
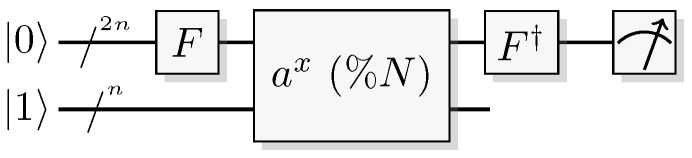
Circuit diagram of the quantum subroutine used in Shor’s algorithm. A 2n-qubit register is initiated in the zero-state $|0\rangle$, and an *n*-qubit register in $|1\rangle$. Basis change of the input-register part of the controlled modular exponentiation operator allow for sampling a probability distribution with peaks at s/r.

**Figure 43 entropy-21-00800-f043:**
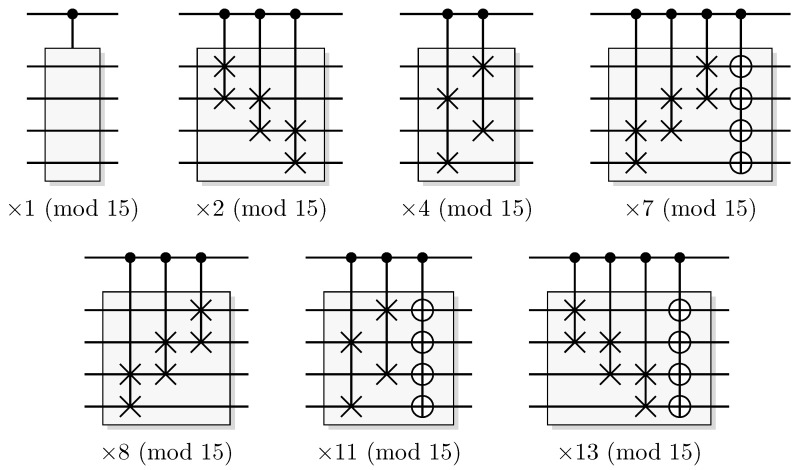
Controlled modular multipliers that occur in Shor’s algorithm.

**Figure 44 entropy-21-00800-f044:**
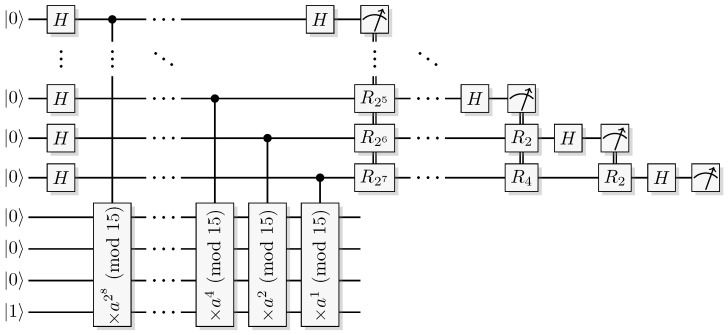
Shor’s algorithm with semiclassical inverse Fourier transform. Please note that $\times 2^{2}\equiv \times 7^{2}\equiv \times 8^{2}\equiv \times 13^{2}\equiv \times 4^{1}$ and $\times 4^{2}\equiv \times 11^{2}\equiv \times 1$ (mod 15), so that many of the controlled multiplications will be identities. Therefore, most rotations $R_{2^{k}}$ will never be applied (in the ideal situation), in fact only the very last $R_{2}$ operation can ever occur.

**Figure 45 entropy-21-00800-f045:**
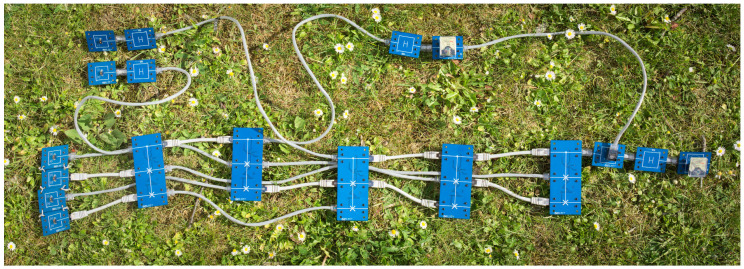
Hardware realization QSL simulation of Shor’s algorithm. For the case a=8 so that the modular multipliers used are, $\times 8^{2}\equiv \times 4$ (mod 15), and ×8 (mod 15).

**Figure 46 entropy-21-00800-f046:**
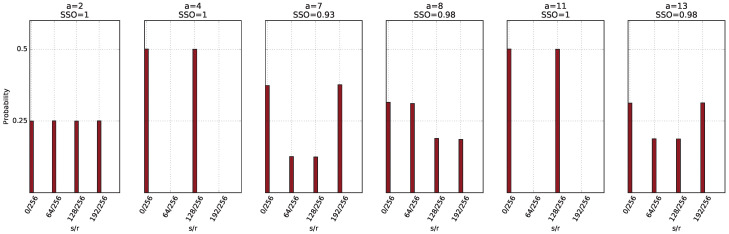
Estimated output probability. Point estimates of the output probability distributions of the subroutine, for the non-trivial elements in the multiplicative group of integers mod 15. Each distribution is estimated from $10^{6}$ samples.

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
