# Peer review of "Quantum Simulation Logic, Oracles, and the Quantum Advantage"

_entropy, 2019, doi:10.3390/e21080800_

Round 1

Reviewer 1 Report

"Quantum Simulation Logic, Oracles, and the Quantum Advantage" by  Niklas Johansson and Jan-Ake Larsson                    A referee report       The paper aims to shed some more light onto the subtle difference between classical and quantum algorithms. To mimic truly quantum information processing the authors introduce a scheme called quantum simulation logic, which allows them to simulate quantum circuits with classical resources. In this model, inspired by the theoretical toy model of Spekkens and easily realizable in the lab, each quantum bit is replaced by two classical bits, which cannot be measured simultaneously. On one hand the authors show, how to execute some known quantum algorithms, including Deutsch-Jozsa algorithm with simple classical, electronic devices, without direct usage of quantum entanglement. On the other hand, the paper contributes to  our understanding, to what extend the complexity of certain algorithms, can be reduced by allowing ne to make use of quantum resources. The work contains valuable review of the field of quantum oracles and provides a readable introduction to quantum algorithms, and a useful Appendix with a catalogue of circuits, so its considerable length can be justified. As the paper presents results interesting from both theoretical and experimental point of view it can be eventually suitable for publication. My minor editorial remarks include 1.   The meaning of horizontal arrows in Eqs. (10)-(12) is not explained so these simple formulae could be misleading. (perhaps one could add operations applied above the arrows, as in Eq. (63)?  Furthermore,  I would ask the authors to verify, whether all signs in Eq. (11) are correct, (or whether a misprint occurred there - state on the left is separable while this on the right is entangled..) 2.  the authors introduce the notion of fidelity between two mixed states and quote the seminal 1994 paper of Jozsa.      However, their formula does not agree with the original definition of Jozsa in which F is equal to the square of trace (of the expression given in Eq. (25)), as for two pure states F has a natural interpretation of probability. 3. It is not very clear to me where formulae (138) comes from. A short explanation would be welcome.  4. p. 50; caption to Fig. 30,  the meaning of two white circles  between the Hadamard gates should be explained  5. Reference [56]  of Cuccaro seems not to be complete.  6. A minor remark addressed also to editors:      I am not entirely sure, whether the current vestion of the article fits well the scope of the journal...Perhaps the authors could add some new material with direct usage of the entropy function to make their paper more related to the key topic of the journal?

Author Response

We would like to start by thanking the reviewer for his or her careful reading of our manuscript. The points 1-5 are all valid (point 6 is for the editor to handle), and we have addressed them as follows.

1. We have added an explanation of what the arrows mean (and standardized to one type of arrows). There was indeed a sign error in both sides of eqn (11), this has been corrected.

2. The formula for fidelity does indeed not agree with the original definition of Jozsa. However, we have chosen to use the definition found in Nielsen and Chuang (Ref. [30]). The ref. to Jozsa was intended to point at the derivation of how the expression simplifies when the states are pure. We have now clarified this in full.

3. A derivation of formulae (138) and (139) has been added.

4. We have clarified which gate is meant by two white connected circles (a controlled-Z with inverted controls), and also added reminders in two more figure captions that white circles denote inverted controls.

5. For some reason, arxiv references were incomplete, these have now been fixed.

Reviewer 2 Report

This long article gives a comprehensive introduction about quantum logic, quantum algorithm and quantum simulation logic. Many of the materials are known. I do not know what are the new findings in this work. What worries me most is the conclusion of this paper:

“These phenomena include results from quantum query complexity in which the query complexity is exactly the same in QSL as in quantum theory. In the circuit model there is only a constant overhead in classical resources, which in turn gives a polynomial time simulation in a classical probabilistic Turing machine.”

If this were true, then there would be no need to build quantum computers. I seriously doubt the validity of this conclusion.

As I do not see the novelty of this paper. The conclusion seems contradicting the current known results about quantum computing. The paper is more like a tutorial material for QSL quantum simulation training course, with faulty conclusions. Therefore I advise reject of the manuscript.

Author Response

-

Reviewer 3 Report

This is a very original and very substantial contribution to the field of quantum computation. The results for quite a bombshell since the authors show that some of the "promise" of quantum computing is illusory. I have to admit that I have not had enough time to check all the mathematical details but I am confident that the results are correct. I have been at presentations of the work at two conferences, with many top experts present, so I have seen how it is received by the experts. The mathematical arguments are relatively clear and straightforward. The paper is of necessity rather long since the arguments which show that various celebrated algorithms from "theoretical quantum computer science" can be implemented on classical computers without loss of speed are certainly delicate and technical. I compliment the authors that they have given a full account, so that true experts in the field (I am not) will be able to check every last detail.

I did find the paper hard to read. The paper seems to start with quite technical material. The first section "introduction" is very short and hard to read, except no doubt for the real specialists. Then there are some extensive (and necessary!) sections with background theory. But what I miss is a real and more extensive introduction for non-specialists, sketching the broad outlines of the work and requiring as little prior technical knowledge as possible. After all, the paper makes quite dramatic and perhaps even controversial claims. It is already very long. I think the world deserves a few more pages of more leisurely introduction.

Author Response

We would like to thank the reviewer for the kind words.

We understand the concern of the reviewer, and have added half a page at the start of the introduction, attempting to sketch the broad outlines of the work while requiring as little prior technical knowledge as possible. Making such an intro a few pages long, as the reviewer suggests, carries the risk of going into too many details. But please let us know if we did not get the balance right.